# Real-World Adverse Weather Image Restoration via Dual-Level Reinforcement Learning with High-Quality Cold Start

**Fuyang Liu[1], Jiaqi Xu[3], Xiaowei Hu[2]***
[1]Nanjing University of Science and Technology
[2]South China University of Technology
[3]Huawei Noah's Ark Lab

## Abstract

Adverse weather severely impairs real-world visual perception, while existing vision models trained on synthetic data with fixed parameters struggle to generalize to complex degradations. To address this, we first construct HFLS-Weather, a physics-driven, high-fidelity dataset that simulates diverse weather phenomena, and then design a dual-level reinforcement learning framework initialized with HFLS-Weather for cold-start training. Within this framework, at the local level, weather-specific restoration models are refined through perturbation-driven image quality optimization, enabling reward-based learning without paired supervision; at the global level, a meta-controller dynamically orchestrates model selection and execution order according to scene degradation. This framework enables continuous adaptation to real-world conditions and achieves state-of-the-art performance across a wide range of adverse weather scenarios. Code is available at https://github.com/xxclfy/AgentRL-Real-Weather

## 1 Introduction

Adverse weather conditions present a persistent challenge for computer vision systems operating in real-world environments. Rain, snow, haze, and their interactions degrade image quality through intricate physical processes, including light scattering by atmospheric particles, dynamic sensor noise, and surface-level phenomena such as water film reflections and ice crystal refraction. Various deep-learning-based methods are developed from adverse weather image restoration, such as weather-specific models for deraining [15, 61, 20, 68, 21, 54, 14], dehazing [19, 4, 13, 44, 48], desnowing [32], and all-in-one models for multiple weather types [27, 46, 11, 35, 69, 49, 63, 36, 58, 8].

Despite these advances, existing methods often struggle in real-world applications due to a fundamental limitation: *models trained on synthetic data fail to generalize effectively to unpredictable real-world degradations.* This performance gap arises from three limitations in current approaches: (i) existing multi-weather synthetic datasets fail to capture the high-precision representation and the intricate physics underlying weather phenomena, (ii) conventional static models lack the capacity to adapt to novel degradation patterns encountered during real-world deployment, and (iii) single-model architectures cannot leverage dynamic coordination strategies to optimally handle diverse and multiple degradation types.

To address these, we develop a self-evolving approach that contains the physics-driven synthetic data generation with a dual-level reinforcement learning architecture. First, we create the High-Fidelity Large-Scale Weather dataset (HFLS-Weather), which simulates weather artifacts like rain, fog, and

---

*Corresponding author: huxiaowei@scut.edu.cn

snow based on their physical formation. This dataset contains one million images, using depth information predicted by a robust depth estimation model [59, 60] to generate realistic weather effects in any scene. On this foundation, we train specialized restoration models for various weather conditions, including rain, snow, haze, and mixed weather types, providing high-quality supervised cold starts. Similarly to LLMs [17], a high-quality "cold start" is essential for the effectiveness of subsequent reinforcement learning.

Second, we design a Dual-level Reinforcement Learning framework (**DRL**) for continuous refinement and real-world adaptation. At the local level, multiple specialized restoration models, such as derain, dehaze, and desnow, are first trained on our HFLS-Weather dataset and then continuously refined based on real-world feedback through reinforcement learning. At the global level, a meta-controller dynamically coordinates the collaboration of individual restoration models (agents) by analyzing degradation patterns and historical execution data. This dual-level synergy establishes a closed-loop learning ecosystem: *at the local level, individual restoration models continuously refine their capabilities based on real-world feedback, while at the global level, the meta-controller dynamically optimizes model coordination for enhanced overall performance.*

The key challenge for this framework is training the dual-level system on real data without paired ground-truth images. Reinforcement learning offers a potential solution, as it doesn't require pixel-level supervision, making it suitable for scenarios where real data, particularly in adverse weather, lacks paired ground-truth images. However, unlike recent successful reinforcement learning applications in large language models (*e.g.*, Group Relative Policy Optimization, GRPO [42]), where (i) multiple responses can be generated for a single prompt, enabling result comparison [17], and (ii) rule-based reward designs work well for tasks with deterministic answers [43], image restoration models typically output a fixed result for each input and it is difficult to derive deterministic rewards without paired ground-truth images.

To train each individual restoration model, we develop Perturbation-driven Image Quality Optimization (**PIQO**), which modifies GRPO [42] in two aspects to make it suitable for image restoration tasks. First, we introduce perturbations to network parameters, enabling the model to generate different results for a single input image, facilitating effective comparison during the learning process. Second, to assess performance and provide learning rewards for unlabeled real-world images, we design a reward assessment strategy for image quality, which integrates various evaluation metrics for restored images. To train the global meta-controller, we take the image quality assessment score as reward to autonomously determine the optimal execution sequence for input images and dynamically select the most suitable model to maximize performance. The scheduling policy is continuously refined through real-world interactions, enabling adaptability to changing conditions.

Lastly, we conduct various experiments under complex real-world conditions across diverse weather scenarios by comparing with various methods for removing weather-related artifacts. The results demonstrate that *our model outperforms the previous methods by a large margin*, both quantitatively and visually. To our knowledge, this is the first work to successfully apply GRPO concepts to image restoration, demonstrating that a high-quality cold start and effective reward design are key to success.

## 2 Related Work

### 2.1 Adverse Weather Image Restoration

Earlier research primarily focused on restoring images degraded by specific weather conditions, such as rain [15, 61, 20, 68, 21, 54, 14], haze [19, 4, 13, 44, 48], and snow [32]. More recent efforts aim to develop unified frameworks for general adverse weather removal [27, 46, 11, 35, 69, 49, 63, 36]. All-in-One [27] first unified weather restoration via joint training; TransWeather [46] introduced transformer-based adaptive queries; and Chen *et al.* [11] used contrastive learning with knowledge distillation. WeatherDiff [35] proposed a diffusion-based model, while WGWS [69] employed a two-stage pipeline for general-to-specific refinement. WeatherStream [64] introduced real degraded–clean pairs but suffered from compression noise. Although these approaches have shown impressive results on synthetic benchmarks [15, 38, 32, 25, 26, 20, 55, 57], their real-world performance is hindered by the domain gap between controlled synthetic data and the complexity of actual environmental conditions. This gap often limits their ability to handle the unpredictable and diverse nature of real-world weather scenarios.

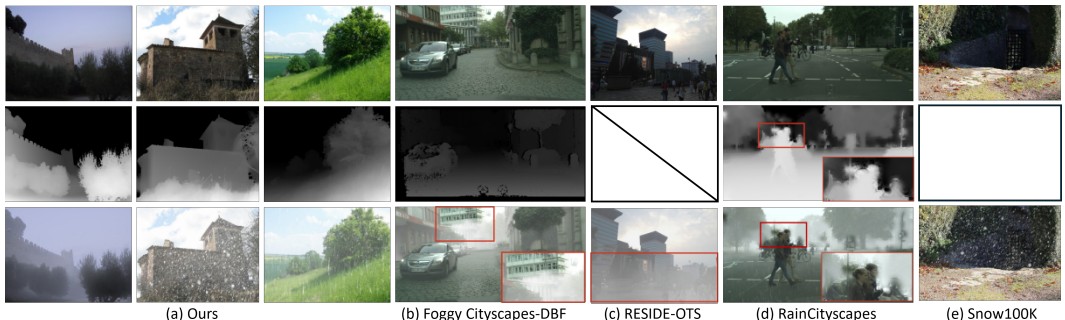

| (a) Ours | (b) Foggy Cityscapes-DBF | (c) RESIDE-OTS | (d) RainCityscapes | (e) Snow100K |

Figure 1: Weather-degraded images from Foggy Cityscapes-DBF [40], RESIDE-OTS [25], RainCityscapes [20], and Snow100K [32] showcase artifacts such as ghosting and uneven weather effects resulting from depth estimation errors. Note that RESIDE-OTS does not provide public depth maps, and Snow100K lacks depth data. The three rows represent clean images, depth maps, and weather-degraded images, respectively.

## 2.2 Large-Scale, Agent-Based, and Perturbation Methods

Recent advancements leveraged large-scale models and multi-agent systems to address image restoration challenges across various adverse weather conditions. DA-CLIP [33] extends CLIP [39] via a dynamic controller for robust embeddings. Other methods integrate external knowledge: *e.g.*, distilling semantics from SAM [65, 24], using prompt learning and depth priors [8], or leveraging VLMs for semi-supervised enhancement [56]. RestoreAgent [6] uses multimodal LLMs to assess, sequence, and apply restoration tools autonomously. AgenticIR [67] coordinates multiple expert agents via LLMs for toolbox-based restoration, including synthetic degradation generation. Very recently, JarvisIR [29] also adopts multi-agent RL strategies for weather-degraded image restoration.

Some works explore *perturbation-based* mechanisms for improving restoration diversity and robustness. DFPIR [45] introduces degradation-aware feature perturbation, while earlier works exploit latent-space or parameter perturbations through learned priors, such as deep mean-shift priors [3] and autoencoding priors [2]. Unlike these, our method employs RL-guided parameter perturbations with IQA-based reward filtering, and further introduces a dual-level structure where a global meta-controller dynamically coordinates local restoration agents, enhancing adaptability in real-world scenarios. Although GPT-4o[2] shows potential in visual editing (*e.g.*, removing weather artifacts), it often produces visually appealing but physically unauthentic outputs (*e.g.*, hallucinated objects, distorted structure) [9], limiting its utility in tasks requiring geometric and photometric fidelity.

## 3 High-Fidelity Large-Scale Weather Dataset

### 3.1 Dataset Overview

Existing weather-related datasets for fog, rain, and snow [41, 20, 21] primarily rely on synthetic images generated via atmospheric scattering models [16], with depth maps sourced from LiDAR [41, 20] or monocular depth estimation [30]. While widely used for weather artifact removal, these datasets suffer from key limitations: (i) LiDAR-based collection is expensive and limited in scale and scene diversity; (ii) depth maps often lack granularity, introducing unrealistic artifacts such as ghosting (see Fig. 1); and (iii) depth estimation models generalize poorly, further degrading realism.

To overcome these issues, we develop **HFLS-Weather**, a large-scale, high-fidelity dataset for realistic weather synthesis. Leveraging an advanced depth prediction model [60], we generate precise and scalable simulations of rain, haze, and snow. The rain and snow simulations include both pure artifacts (*i.e.*, rain-only or snow-only) and mixed conditions combining rain or snow with haze. This design improves both the diversity and physical plausibility of training data (Fig. 1a).

HFLS-Weather offers two core advantages: (i) *High-fidelity depth at scale.* We generate accurate depth maps from clear-weather images using a state-of-the-art model, eliminating the cost and limitations of LiDAR while enabling realistic synthesis across diverse scenes. (ii) *Depth-consistent multi-weather simulation.* A unified framework applies depth-driven attenuation not only to haze,

---

[2]https://openai.com/index/hello-gpt-4o/

Table 1: Comparison of datasets for image restoration under adverse weather.

| Dataset | Year | Weather | Depth | Depth Source | #Clean | #Pairs | Real/Syn |
|---|---|---|---|---|---|---|---|
| Snow100K [32] | 2017 | Snow | No | - | 50,000 | 50,000 | Syn |
| Rain14000 [15] | 2017 | Rain | No | - | 650 | 9,100 | Syn |
| RESIDE-OTS [25] | 2018 | Haze | Yes | DCNF | 2,061 | 72,135 | Syn |
| NTURain [7] | 2018 | Rain | No | - | - | 3,123 | Syn |
| Foggy Cityscapes [41] | 2018 | Haze | Yes | Stereo Vision | 2,975 | 8,925 | Syn |
| Foggy City.-DBF [40] | 2018 | Haze | Yes | Stereo Vision | 2,975 | 8,925 | Syn |
| RESIDE-RTTS [25] | 2018 | Haze | - | - | - | 4,322 | Real |
| RainHeavy25 [62] | 2019 | Rain | No | - | - | 1,710 | Syn |
| RainCityscapes [20, 21] | 2019 | Rain | Yes | Stereo Vision | 262 | 9,432 | Syn |
| Outdoor-Rain [26] | 2019 | Rain | No | - | - | 13,500 | Syn |
| CSD [10] | 2021 | Snow | No | - | - | 10,000 | Syn |
| WeatherStream [64] | 2023 | Rain/Haze/Snow | No | - | «188,000 | 188,000 | Real (diff. time) |
| CDD11 [18] | 2024 | Rain/Haze/Snow | Yes | MegaDepth [28] | 1,383 | 13,013 | Syn |
| Weather30K [57] | 2025 | Rain/Haze/Snow | No | - | 30,000 | 30,000 | Syn |
| **HFLS-Weather** | **2025** | **Rain/Haze/Snow** | **Yes** | **DepthAnything v2 [60]** | **1,000,000** | **1,000,000** | **Syn** |

but also to rain and snow, ensuring that all weather effects decay realistically with distance. This yields physically consistent degradations aligned with scene geometry, supporting robust training for multi-weather restoration models.

## 3.2 Weather-Related Artifact Simulation

We simulate realistic fog, rain, snow, and hybrid conditions using an atmospheric scattering model [5, 16], incorporating depth-dependent interactions between weather phenomena. In real-world scenarios, distant objects are often obscured by fog, while proximate regions exhibit rain streaks or snowflakes [20, 21]. To model this, we define the weather-affected image $I_{\text{weather}}(x)$ as

$$I_{\text{weather}}(x) = J(x)\left(1 - M(x) - F(x)\right) + M(x) + A(x)F(x) ,$$

where $J(x)$ represents the clean image, $M(x) \in [0, 1]$ denotes the rain/snow layer, and $F(x) = e^{-\beta d(x)}$ corresponds to the fog layer, with $\beta$ as the atmospheric scattering coefficient and $d(x)$ representing high-quality depth information. $A(x)$ is the global atmospheric light. Fog is simulated using a transmission map $F(x) = e^{-\beta d(x)}$, where $\beta$ controls fog density and $d(x)$ provides depth information, allowing fog to naturally obscure distant objects while leaving closer objects clearer. Rain and snow are represented by the rain/snow layer $M(x)$, a semi-transparent mask created through procedural generation. Rain streaks appear more intense on closer objects, while snowflakes accumulate in a scattered pattern, adding realism through variable opacity based on depth. In both rain and snow conditions, fog effects can be applied to objects at greater distances from the camera.

## 3.3 Dataset Comparison

To construct HFLS-Weather, we collected one million clean images from diverse sources including Snow100K [32], RESIDE-OTS [25], Google Landmark V2 [51], and OSV5M [1]. Each image was randomly augmented with one weather type, *i.e.*, haze, rain (rain-only & rain+haze), and snow (snow-only & snow+haze), using our physically grounded synthesis pipeline, resulting in one million high-quality degraded images.

As summarized in Table 1, HFLS-Weather provides balanced coverage across rain, haze, and snow, unlike prior datasets that target single weather types. Its use of high-fidelity depth enables accurate simulation of weather effects, improving realism and consistency. With one million diverse backgrounds and generated pairs, it surpasses existing datasets in both scale and diversity, facilitating robust generalization. By combining large-scale synthesis with physically consistent depth cues, HFLS-Weather addresses key limitations of prior benchmarks and supports inter-condition learning for advanced weather artifact removal.

## 4 Dual-Level Reinforcement Learning Framework

In this work, we present a dual-level reinforcement learning framework for real-world adverse weather image restoration, integrating both Perturbation-Driven Image Quality Optimization (PIQO) and a Multi-Agent System to continuously refine restoration models through real-world feedback.

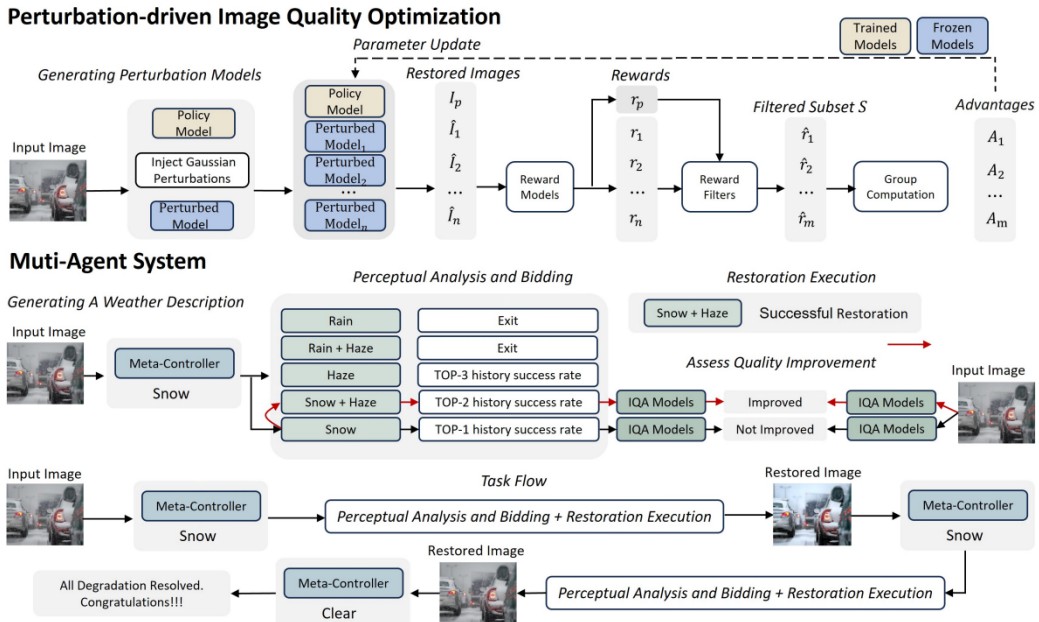

Figure 2: Architecture of Perturbation-Driven Image Quality Optimization and Multi-Agent System.

Figure 2 illustrates our framework, where PIQO applies Gaussian perturbations to model parameters, generating multiple restored images that are evaluated using quality assessment models to guide learning and improve adaptation to various weather conditions. The Multi-Agent System uses a meta-controller to generate weather descriptions and dynamically select the most suitable restoration model based on historical success rates, thereby enhancing performance across diverse weathers.

## 4.1 Perturbation-driven Image Quality Optimization

While reinforcement learning has advanced LLM alignment through techniques like Group Relative Policy Optimization (GRPO) [42], its application to image restoration is less explored. Unlike text generation, which naturally supports one-to-many outputs, image restoration typically follows a one-to-one mapping from degraded inputs to plausible outputs, limiting diversity and complicating reward design, especially without paired ground truth. Rule-based rewards, effective in deterministic settings [43], often underperform in such underconstrained scenarios.

To address this, we present Perturbation-driven Image Quality Optimization (PIQO), a GRPO-inspired framework tailored for image restoration. PIQO injects small Gaussian perturbations into model parameters during inference to produce diverse outputs for the same input. Let $\theta$ be the current model parameters and the $i$-th perturbed version $\theta'_i = \theta + \Delta$. For each degraded input image, we generate multiple outputs $\hat{I}_i$ using perturbed models $f(\theta'_i)$.

To evaluate output quality without paired supervision, we define a composite no-reference reward function combining four metrics: LIQE [66], CLIP-IQA [47], and Q-Align [52]. The reward for the $i$-th output is:

$$r_i = w_1 \times \text{LIQE}(\hat{I}_i) + w_2 \times \text{CLIP-IQA}(\hat{I}_i) + w_3 \times \text{Q-Align}(\hat{I}_i), \tag{1}$$

where $w_*$ are the weights for each metric. This reward function produces a scalar score $r_i$ for each candidate output, reflecting its predicted perceptual quality. By design, higher indicates better overall visual quality as judged by the ensemble of metrics.

Not all perturbed outputs are beneficial for learning, and some may produce degraded images with low rewards, introducing high variance or even harmful gradients. To mitigate this, we apply a reward filtering step that discards outputs whose rewards fall below that of the unperturbed model, ensuring the optimization focuses only on advantageous directions. Let $I_p$ denote the output from the current

model and MUSIQ [23] be the filtering criterion; we retain only the indices with:

$$\mathcal{S} = i \mid \text{MUSIQ}(\hat{I}_i) > \text{MUSIQ}(I_p). \tag{2}$$

Next, we compute the normalized advantage $A_i$ for each retained sample $i \in \mathcal{S}$, which reflects how much its reward deviates from the group mean:

$$A_i = \frac{r_i - \bar{r}}{\sigma_r + \varepsilon}, \quad \bar{r} = \frac{1}{N}\sum_{j=1}^{N} r_j, \quad \sigma_r = \sqrt{\frac{1}{N}\sum_{j=1}^{N}(r_j - \bar{r})^2}, \tag{3}$$

where $\bar{r}$ and $\sigma_r$ denote the mean and standard deviation of the rewards $\{r_j\}_{j=1}^{N}$ within the retained group $\mathcal{S}$. This relative advantage indicates the quality of a perturbed output compared to the group average: positive $A_i$ suggests a beneficial perturbation, while negative $A_i$ implies a less favorable one (note that low-quality samples have mostly been filtered out by the $r_{\text{p}}$ baseline). Normalizing the advantages reduces variance in the gradient estimate and serves as a built-in baseline, akin to standard policy gradient methods.

Given the filtered set of perturbations and their advantages, PIQO updates the model parameters in the direction that increases expected image quality. We estimate the cumulative policy gradient $g$ over the filtered subset $\mathcal{S}$:

$$g = -\frac{1}{|\mathcal{S}|}\sum_{i \in \mathcal{S}} A_i(\theta_i' - \theta), \tag{4}$$

where the negative sign reflects gradient ascent on reward.

To stabilize updates, we apply implicit KL regularization by approximating divergence in parameter space. This is analogous to the trust-region constraint in PPO, which limits how much the policy can change in a single step.

$$\text{KL}_{\text{approx}} = \frac{1}{|\mathcal{S}|}\sum_{i \in \mathcal{S}} \frac{1}{|\theta|}\sum_{j=1}^{|\theta|}\left(\theta_{i,j}' - \theta_j\right)^2, \tag{5}$$

where $|\theta|$ is the number of parameters and $\theta_{i,j}'$ the $j$-th parameter of the $i$-th perturbed model.

We compute a scaling factor based on a KL threshold $\tau$:

$$\text{scale} = \begin{cases} \sqrt{\tau/\text{KL}_{\text{approx}}}, & \text{if } \text{KL}_{\text{approx}} > \tau \\ 1, & \text{otherwise} \end{cases} \tag{6}$$

The final parameter update is:

$$\theta \leftarrow \theta + \eta \cdot \text{scale} \cdot g, \tag{7}$$

where $\eta$ is the learning rate. This update ensures stability by preventing large shifts when parameter divergence is high.

PIQO extends GRPO to image restoration by enabling learning from unlabeled real-world data through perturbation-induced diversity, reward-based filtering, and variance-reduced gradient updates.

### 4.2 Muti-Agent System

To further deal with the complex adverse weather conditions in real world, we present a multi-agent system for image restoration that can handle one or multiple adverse weather types by learning from real data. After training individual restoration models for specific weather conditions, the system utilizes specialized agents, each focusing on a particular type of degradation. These agents collaborate autonomously through a bidding mechanism driven by perceptual analysis.

As shown in Figure 2, the process begins by analyzing the input image using a meta-controller (CLIP [39]), which identifies the dominant weather-related degradation and generates a corresponding "weather description." This semantic description guides the selection of agents for subsequent restoration stages. The system broadcasts this weather description to all registered agents, who assess the compatibility of their specialization with the identified degradation. Each agent decides whether to participate in the bidding process based on its historical success rate with similar degradations.

Table 2: Performance comparison in real-world scenarios, evaluated by IQA metrics.

| Method | Snow | | | | Haze | | | | Rain | | | |
|---|---|---|---|---|---|---|---|---|---|---|---|---|
| | Q-Align | CLIP-IQA | LIQE | MUSIQ | Q-Align | CLIP-IQA | LIQE | MUSIQ | Q-Align | CLIP-IQA | LIQE | MUSIQ |
| Chen et al.[11] | 3.5898 | 0.4959 | 3.1256 | 60.2062 | 3.1109 | 0.3373 | 2.0729 | 54.0597 | 3.7629 | 0.4201 | 2.5429 | 54.2367 |
| WGWS [69] | 3.5901 | 0.5026 | 3.1042 | 60.4800 | 3.1137 | 0.3643 | 2.1464 | 54.0680 | 3.7986 | 0.4428 | 2.5310 | 54.5487 |
| PromptIR [37] | 3.6492 | 0.5291 | 3.2397 | 61.1700 | 3.0906 | 0.3757 | 2.0673 | 53.8121 | 3.8074 | 0.4466 | 2.5622 | 54.6686 |
| OneRestore [18] | 3.5884 | 0.5089 | 3.1478 | 61.3300 | 2.9825 | 0.3293 | 2.0571 | 53.9140 | 3.7019 | 0.4167 | 2.4556 | 55.0806 |
| DA-CLIP [33] | 3.6261 | 0.5219 | 3.2410 | 61.1583 | 3.1301 | 0.3687 | 2.0797 | 54.4134 | 3.8144 | 0.4656 | 2.5810 | 54.9613 |
| **Ours** | **3.9569** | **0.5918** | **3.9458** | **67.7990** | **3.5608** | **0.4561** | **3.0267** | **63.3000** | **4.0283** | **0.5623** | **3.2945** | **64.1187** |

Table 3: Performance comparison in real-world scenarios, evaluated by GPT-4o.

| Weather | Metric | Chen et al. | WGWS | PromptIR | OneRestore | DA-CLIP | DFPIR | JarvisIR | **Ours** |
|---|---|---|---|---|---|---|---|---|---|
| Snow | Artifact Removal ↑ | 2.949 | 2.664 | 3.057 | 3.116 | 3.047 | – | 3.570 | **4.421** |
| | Weather Resilience ↑ | 3.014 | 3.045 | 3.172 | 3.440 | 3.128 | – | 3.610 | **4.355** |
| | Overall Visual Quality ↑ | 3.012 | 2.936 | 3.232 | 3.464 | 3.003 | – | 3.730 | **4.393** |
| Haze | Artifact Removal ↑ | 3.142 | 3.015 | 3.083 | 3.415 | 3.071 | 3.170 | 3.650 | **4.074** |
| | Weather Resilience ↑ | 3.056 | 2.935 | 3.014 | 3.322 | 3.016 | 3.140 | 3.450 | **4.015** |
| | Overall Visual Quality ↑ | 3.070 | 3.098 | 2.978 | 3.415 | 3.211 | 3.240 | 3.580 | **3.948** |
| Rain | Artifact Removal ↑ | 2.965 | 2.978 | 3.371 | 3.359 | 3.275 | 3.340 | 3.710 | **4.254** |
| | Weather Resilience ↑ | 2.841 | 2.923 | 3.323 | 3.222 | 3.159 | 3.150 | 3.690 | **4.007** |
| | Overall Visual Quality ↑ | 2.984 | 3.014 | 3.314 | 3.201 | 3.163 | 3.180 | 3.870 | **3.896** |

Only agents with a high likelihood of success, based on past performance, submit bids. The system ranks the bidding agents by their historical success rates and selects the top-ranked agent to handle the restoration task. Once the restoration is completed, the system evaluates the result through a two-step process: (i) The CLIP model re-analyzes the restored image to check for the targeted degradation. (ii) An objective Image Quality Assessment (IQA) score is calculated to assess the quality improvement. Specifically, we adhere to the PIQO reward configuration; see Eq. (1).

If the IQA score decreases compared to the previous round, the restoration is considered a failure. In this case, the system reverts the image to its previous state, removes the failed agent from the candidate list, and selects the next highest-ranked agent for another attempt. This process continues until a successful restoration is achieved or three consecutive failures occur, at which point the image with the highest IQA score is returned.

If the restoration does not result in a decrease in IQA score, the system checks whether further degradation is present using the CLIP model. If degradation is still detected, the restoration is considered partially successful, and the system enters the next round. The bidding process is re-initiated, excluding the agent from the previous round to avoid redundancy. If no further degradation is detected, the restoration is considered complete, and the image is returned as the final output.

To avoid computational overload, the system limits the number of agents involved in a single restoration to three. If three consecutive restoration attempts result in failure (*i.e.*, successive IQA drops), the process terminates, and the image with the highest IQA score is returned.

### 4.3 Training Strategy

We adopt the dual-level reinforcement learning strategy to train a DSANet [12]-based multi-agent system for real-world weather adaptation, utilizing eight NVIDIA RTX 4090 GPUs. The training process begins with a cold start on the HFLS-Weather dataset, using the Adam optimizer with a batch size of eight and a learning rate of 0.0001 for up to 100 epochs. Early stopping is applied based on the validation loss. At this stage, we also fine-tune the rain sub-model on the SPA+ dataset[69]to further improve rain removal. At the local level, task-specific restoration agents (*e.g.*, deraining, dehazing, desnowing) are enhanced via Perturbation-driven Image Quality Optimization (PIQO), guided by weighted image quality assessment (IQA) rewards with weights $w_1 = 0.2$, $w_2 = 1$, and $w_3 = 0.2$. The training leverages real-world data, including 2,318 hazy images from the URHI dataset [25], and 2,433 rainy images and 2,018 snowy images from the WReal dataset [56], with a learning rate of 0.0001 and a batch size of 16. At the global level, the multi-agent system is further optimized using a batch size of 16 and a learning rate of 0.0001.

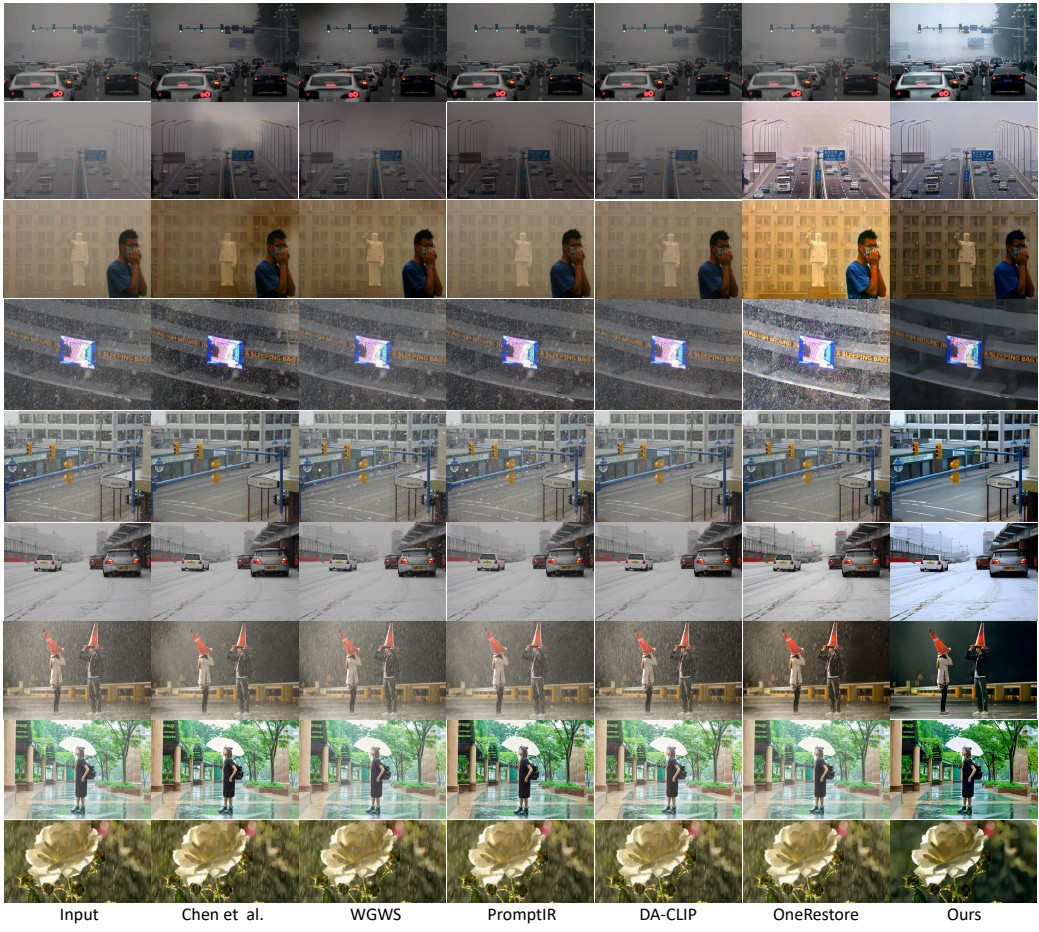

| Input | Chen et al. | WGWS | PromptIR | DA-CLIP | OneRestore | Ours |

Figure 3: Visual comparisons of real images under haze, snow, and rain, with [11, 18, 33, 37, 69].

## 5 Experimental Results

To assess model performance in removing weather artifacts, we use the WReal dataset [56], which includes 4,322 real haze images from RTTS [25], 2,320 real rain images from DDN-SIRR [50] and Real3000 [31] (excluding synthetic scenes), and 1,329 real snow images from Snow100K [32].

### 5.1 Performance Comparison with State-of-the-Art Methods

**Quantitative comparison.** In real-world scenarios, the lack of labeled images affected by haze, snow, or rain complicates the evaluation of image restoration methods. Metrics such as CLIP-IQA [47] and Q-Align [53], originally designed for general datasets, often fail to capture subtle noise and residual artifacts, resulting in inflated scores that misrepresent visual quality. To address these issues, besides IQA metrics, we use GPT-4o [34] for additional evaluation across artifact removal, weather resilience, and overall visual quality. Artifact removal assesses artifact suppression and color accuracy, while weather resilience evaluates model robustness in more challenging cases. Overall visual quality measures the aesthetic coherence of the image.

Table 2 and Table 3 show the results of common image quality metrics and GPT-4o evaluation, comparing with recent image restoration methods under adverse weather conditions, including Chen et al.[11], WGWS[69], PromptIR[37], OneRestore[18], DA-CLIP[33], DFPIR [45], and JarvisIR [29]. From the results, we observe the following: (i) *Our method outperforms all competitors under snow, haze, and rain*, demonstrating strong generalization across diverse real-world degradations. (ii) *Our method achieves the highest scores in all IQA metrics*, reflecting superior visual fidelity and semantic relevance. (iii) *GPT-4o's perceptual evaluation highlights our method's excellence* in artifact removal,

Table 4: Comparison of models that are pre-trained on various synthetic and real datasets as different "cold starts", followed by further finetuning with our PIQO approach.

| Metric | Snow | | | | Haze | | | Rain | | | |
|---|---|---|---|---|---|---|---|---|---|---|---|
| | RealSnow | Snow100K | Our Snow | Our Snow+Haze | OTS | ITS | Our Haze | SPA+ | Rain1300 | Our Rain | Our Rain+Haze |
| Q-Align | 3.6974 | 3.7490 | 3.8482 | **3.8693** | 3.1220 | 3.1014 | **3.5329** | 3.7805 | 3.6974 | **3.9318** | 3.9205 |
| CLIP-IQA | 0.5315 | 0.5149 | **0.5653** | 0.5625 | 0.3846 | 0.3697 | **0.4496** | 0.4612 | 0.4205 | 0.5340 | **0.5615** |
| LIQE | 3.3386 | 3.4609 | 3.7094 | **3.7741** | 2.1239 | 2.1412 | **3.0120** | 2.6766 | 2.5681 | 2.9805 | **3.0485** |
| MUSIQ | 63.5571 | 64.2048 | **69.86** | 69.86 | 54.8760 | 54.5783 | **63.6006** | 56.5147 | 55.2493 | 60.4364 | **61.8700** |

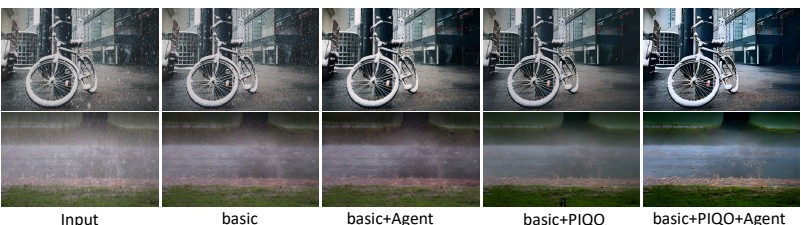

Input     basic     basic+Agent     basic+PIQO     basic+PIQO+Agent

Figure 4: Visual ablation study of the framework components.

weather resilience, and overall visual quality. (iv) While competing methods fluctuate across different weather conditions, our approach maintains stable and superior performance, demonstrating that *real-world refinement via dual-level reinforcement learning significantly boosts generalization and robustness* beyond the models trained solely on synthetic data.

**Visual comparison.** Figure 3 shows the visual comparison results, where the effectiveness of our approach is demonstrated across different weather conditions, including haze, snow, and rain. As shown, our method consistently delivers superior restoration quality, better preserving fine details, colors, and structural integrity of the scene, while others may fail to remove weather degradations, recover important features or introduce artifacts, particularly in challenging real weather conditions.

**Latency comparison.** Our Multi-Agent System incurs higher latency (570 ms) due to running multiple specialized models, yet it achieves superior restoration quality under complex weather. Compared with single-model baselines such as OneRestore (17 ms), Chen et al. (18 ms), WGWS (95 ms), and PromptIR (208 ms), our framework is slower but more robust. At the same time, it remains far more efficient than other multi-agent systems like DA-CLIP (6543 ms) and JarvisIR (15250 ms), striking a favorable balance between efficiency and performance.

## 5.2 Ablation Study

**Evaluation on the "cold starts."** To evaluate the effectiveness of our high-fidelity synthetic dataset for cold-start pretraining, we conduct studies on both single-degradation and mixed-weather datasets, comparing models initialized on different synthetic datasets and then refined using our PIQO.

As the results in Table 4, we have the following observations. (i) *Superiority of High-Quality Cold Start*: Models initialized with HFLS-Weather show significant improvements in image quality over those trained on prior public datasets for snow, haze, and rain. (ii) *Cross-Weather Pretraining Helps*: Incorporating weather diversity in pretraining, such as combining snow and haze, enhances performance, suggesting that exposure to multiple degradation types improves generalization during PIQO finetuning. *Additional comparisons of HFLS-Weather are in the Appendix.*

**Evaluation of the framework design.** To assess the contribution of each component, we conduct ablation studies under real-world weather conditions using three quantitative metrics: CLIP-IQA, LIQE, and Q-Align. The *Basic* model refers to the baseline image restoration network pretrained on our HFLS-Weather synthetic dataset, while *Agent* denotes the proposed multi-agent coordination system. As shown in Table 5 and Figure 4: (i) adding the PIQO training significantly enhances perceptual quality, especially under challenging conditions like rain and snow; (ii) the Agent framework improves adaptive restoration by dynamically dispatching specialized agents; and (iii) combining PIQO with Agent yields the best performance across all metrics, highlighting the effectiveness of joint local optimization and global coordination.

Table 5: Ablation study of the framework components under snow, haze, and rain.

| Method | Snow | | | Haze | | | Rain | | |
|---|---|---|---|---|---|---|---|---|---|
| | CLIP-IQA | Q-Align | LIQE | CLIP-IQA | Q-Align | Q-Align | CLIP-IQA | Q-Align | Q-Align |
| Basic | 0.4774 | 3.6649 | 3.1794 | 0.3661 | 3.2673 | 2.0232 | 0.4392 | 3.7678 | 2.5349 |
| Basic + Agent | 0.5242 | 3.7415 | 3.4893 | 0.3814 | 3.2977 | 2.2302 | 0.4721 | 3.8785 | 2.6871 |
| Basic + PIQO | 0.5653 | 3.8482 | 3.7094 | 0.4496 | 3.5329 | 3.0120 | 0.5340 | 3.9318 | 2.9805 |
| Basic + PIQO + Agent | **0.5918** | **3.9458** | **3.9569** | **0.4561** | **3.5608** | **3.0267** | **0.5623** | **4.0283** | **3.2945** |

# 6 Conclusion

We develop a dual-level reinforcement learning framework for real-world adverse weather image restoration, combining a physics-driven synthetic dataset (HFLS-Weather) with a two-tier adaptive learning system. At the local level, weather-specific models are refined using perturbation-driven optimization without paired supervision. At the global level, a meta-controller dynamically schedules model execution based on degradation patterns. Nevertheless, the multi-agent system introduces extra inference-time overhead as a result of its multi-round interactions.

**Potential negative societal impacts.** While our method improves visual robustness in adverse conditions, it may be misused for surveillance or deepfake generation, and poses risks in safety-critical applications without proper validation. Responsible use and safeguards are necessary.

# Acknowledgment

This work was supported by the Research Start-up Fund for Prof. Xiaowei Hu at the Guangzhou International Campus, South China University of Technology (Grant No. K3250310).

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

# A  Additional Experimental Results

## A.1  Training on Different Synthetic Datasets

Table below compares the performance of models trained on different synthetic datasets for snow, haze, and rain conditions, using metrics such as Q-Align [53], CLIP-IQA [47], LIQE [66], and MUSIQ [23]. For snow, models trained on "Our Snow" and "Our Snow+Haze" datasets outperform others, with "Our Snow+Haze" achieving the highest Q-Align score of 3.7179 and the best CLIP-IQA score of 0.4964, indicating superior quality and alignment. Haze models also favor "Our Haze," which scores highest in Q-Align (3.2673) and LIQE (2.2555), suggesting better restoration of haze-affected images compared to others like OTS and ITS. In the rain category, the "Our Rain+Haze" model excels across all metrics, particularly in Q-Align (3.8148) and MUSIQ (56.1142), outperforming models trained on Rain13k and "Our Rain."

These results highlight the benefits of combining multiple weather conditions (rain and haze) in training, as it leads to more realistic and effective restoration. Overall, our dataset consistently outperforms other synthetic datasets, proving the effectiveness of HFLS-Weather in generating realistic and high-fidelity weather conditions for image restoration tasks.

Table 6: Performance comparison of models trained on different synthetic datasets.

| Metric | Snow | | | | Haze | | | Rain | | | |
|---|---|---|---|---|---|---|---|---|---|---|---|
| | RealSnow[69] | Snow100K[32] | Our Snow | Our Snow+Haze | OTS[25] | ITS[25] | Our Haze | SPA+[68] | Rain13k[22] | Our Rain | Our Rain+Haze |
| Q-Align | 3.6037 | 3.6357 | 3.6649 | **3.7179** | 3.0942 | 3.0309 | **3.2673** | 3.7707 | 3.6357 | 3.7678 | **3.8148** |
| CLIP-IQA | 0.4727 | 0.5025 | 0.4774 | **0.4964** | **0.3994** | 0.3661 | 0.3786 | 0.4335 | 0.3839 | 0.4392 | **0.4439** |
| LIQE | 3.0512 | 3.2642 | 3.1794 | **3.3741** | 2.0663 | 2.0232 | **2.2555** | 2.4695 | 2.3275 | 2.5349 | **2.5779** |
| MUSIQ | 61.3510 | 61.6517 | **62.3326** | 62.3262 | 53.9367 | 53.9367 | **55.8514** | 55.1913 | 53.6375 | 55.4581 | **56.1142** |

## A.2  Re-training Other Methods on the HFLS-Weather

A potential concern is that performance improvements may primarily come from the scale of the HFLS-Weather dataset rather than the proposed framework. To clarify this, we first report in the main paper (Table 5) the results of our model trained solely on HFLS-Weather without reinforcement learning (PIQO) or the agent framework, which already demonstrates the additional benefit of our algorithmic components.

To further disentangle the contributions, we re-trained two representative baselines, WGWS (CVPR 2023) and OneRestore (ECCV 2024), on HFLS-Weather using their released code. The quantitative results are presented in Table 7. Both baselines improve notably when trained on HFLS-Weather, confirming that the dataset provides strong supervision. Nevertheless, our full framework consistently outperforms them across all metrics and weather conditions, indicating that dataset scale alone does not account for the gains.

These results demonstrate two key points: (i) HFLS-Weather is indeed a valuable contribution that benefits existing methods, and (ii) the proposed PIQO and dual-level agent framework provide substantial additional improvements, establishing state-of-the-art performance under diverse weather degradations.

Table 7: Performance of re-trained baselines on HFLS-Weather. Metrics include LIQE, CLIP-IQA, Q-Align, and MUSIQ.

| Method | Snow | | | | Haze | | | | Rain | | | |
|---|---|---|---|---|---|---|---|---|---|---|---|---|
| | LIQE | CLIP | Q-Align | MUSIQ | LIQE | CLIP | Q-Align | MUSIQ | LIQE | CLIP | Q-Align | MUSIQ |
| WGWS | 3.71 | 0.54 | 3.40 | 65.67 | 3.31 | 0.40 | 2.57 | 56.76 | 3.81 | 0.47 | 2.70 | 56.62 |
| OneRestore | 3.73 | 0.55 | 3.56 | 65.32 | 3.38 | 0.41 | 2.67 | 57.13 | 3.83 | 0.48 | 2.68 | 58.67 |
| **Ours** | **3.84** | **0.56** | **3.70** | **66.30** | **3.53** | **0.44** | **3.01** | **63.60** | **3.93** | **0.53** | **2.98** | **60.43** |

# B  More Comparisons with the State-of-the-Art Methods

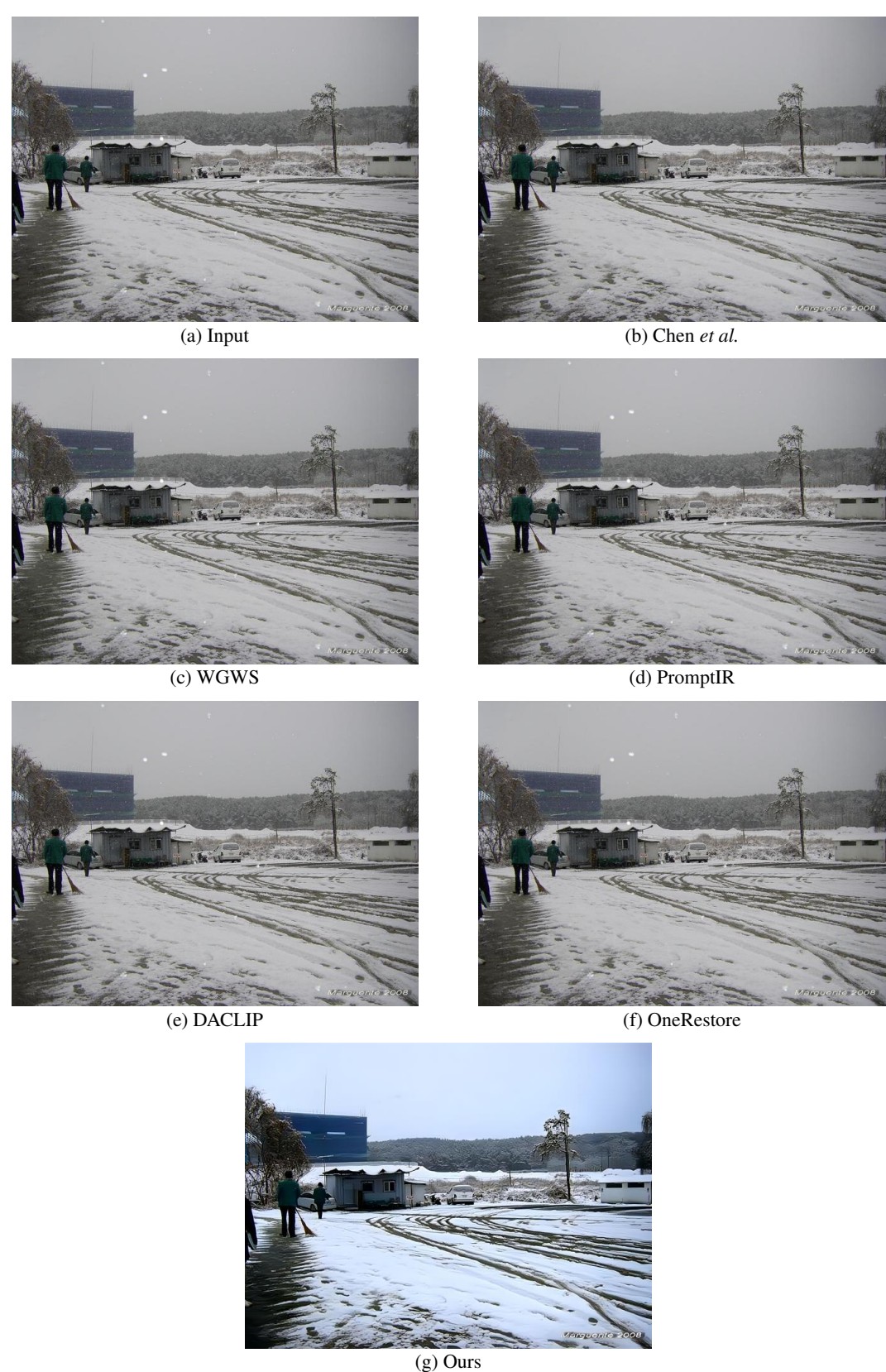

Figure 5: Visual comparison of real-world images under snow with [11, 18, 33, 37, 69].

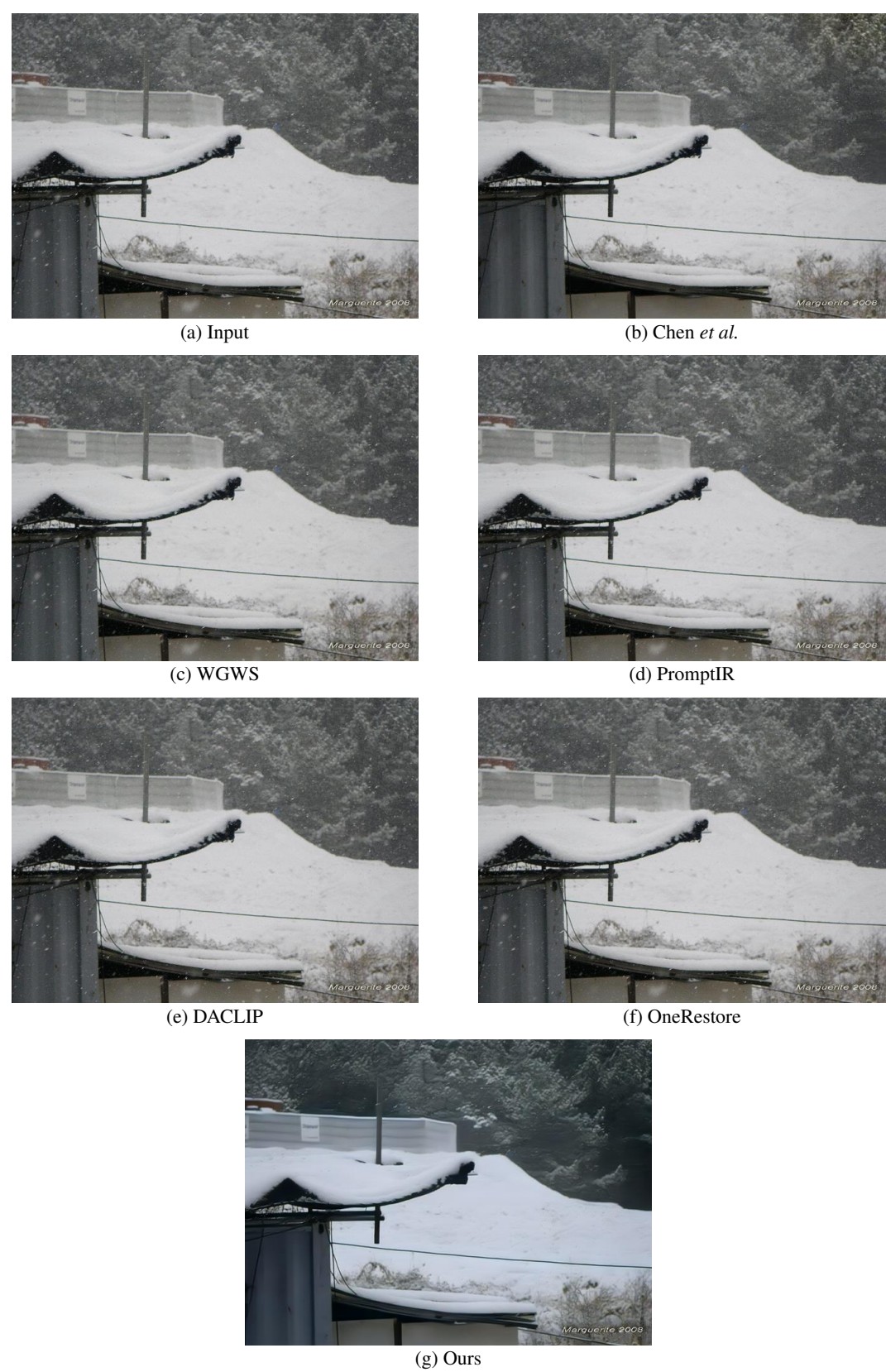

Figure 6: Visual comparison of real-world images under snow with [11, 18, 33, 37, 69].

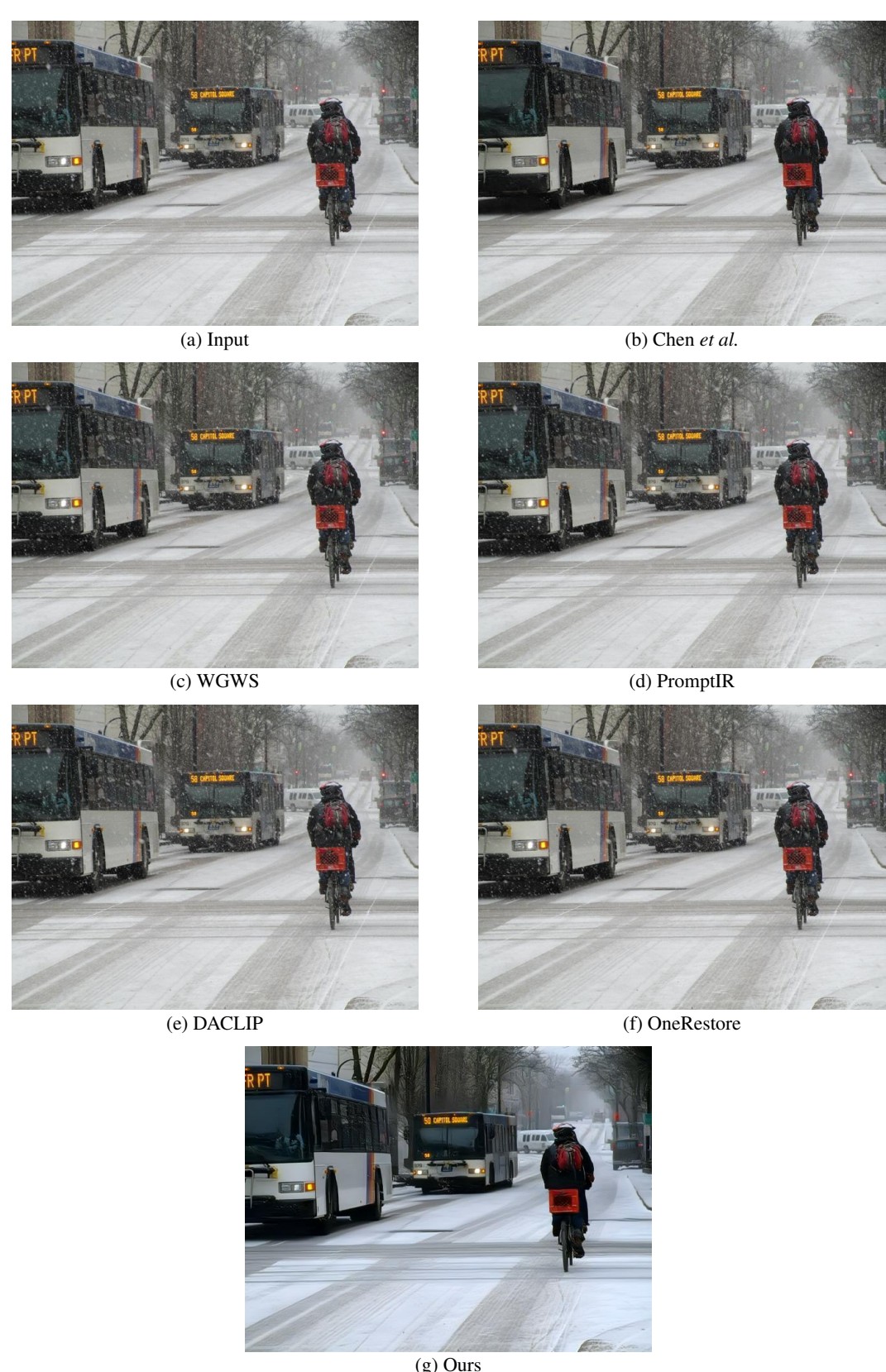

(a) Input

(b) Chen *et al.*

(c) WGWS

(d) PromptIR

(e) DACLIP

(f) OneRestore

(g) Ours

Figure 7: Visual comparison of real-world images under snow with [11, 18, 33, 37, 69].

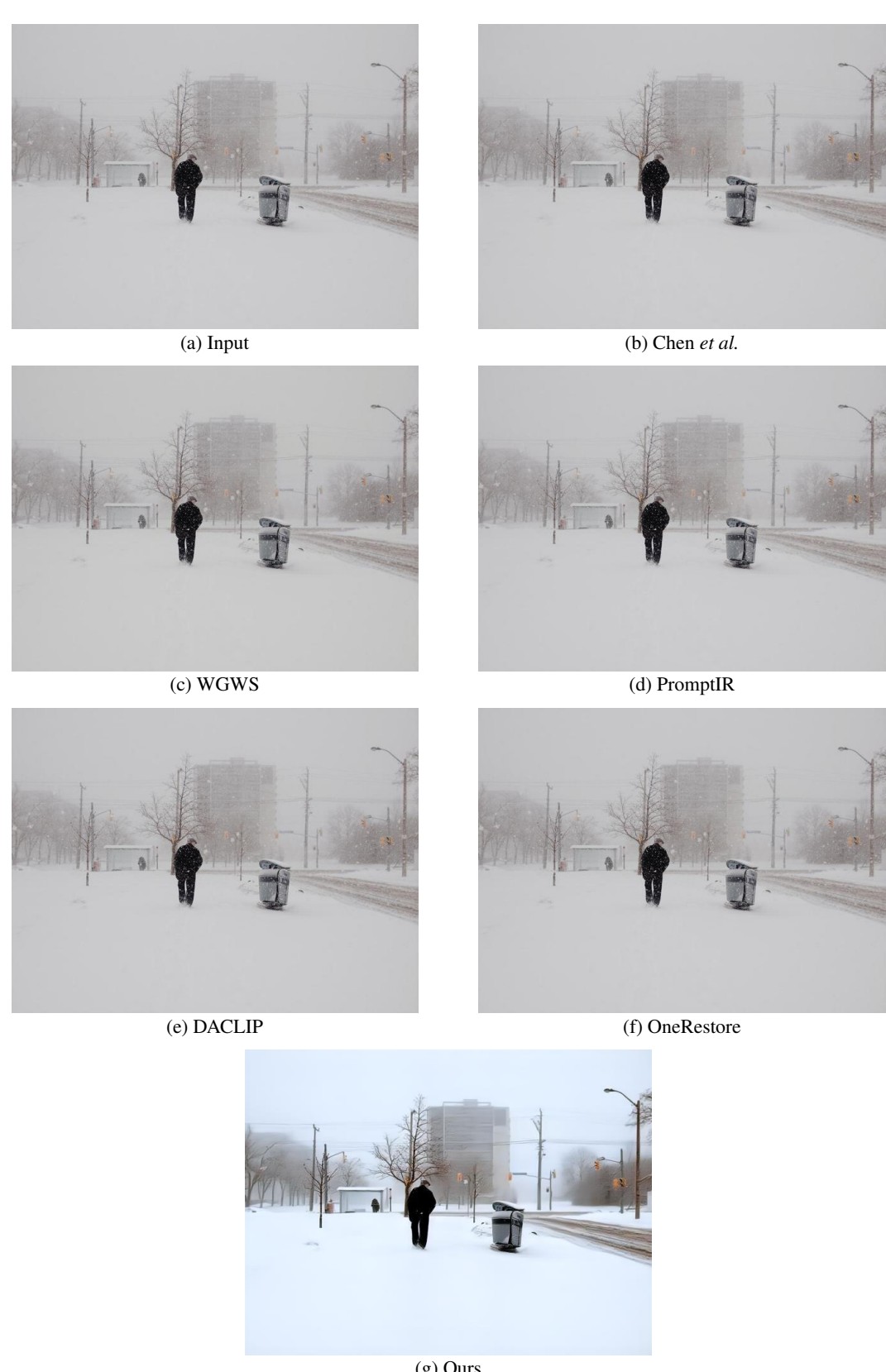

Figure 8: Visual comparison of real-world images under snow with [11, 18, 33, 37, 69].

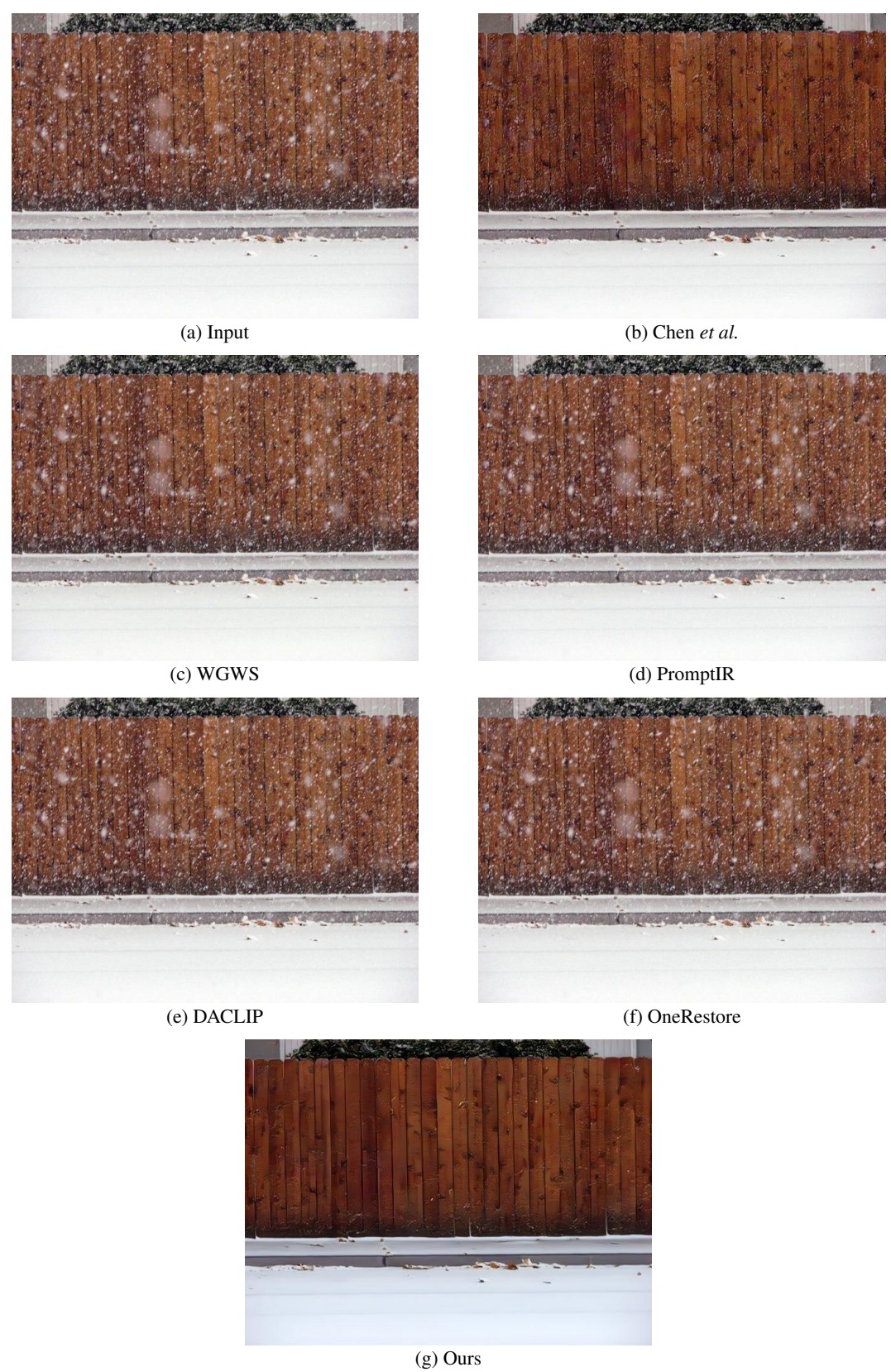

(a) Input

(b) Chen *et al.*

(c) WGWS

(d) PromptIR

(e) DACLIP

(f) OneRestore

(g) Ours

Figure 9: Visual comparison of real-world images under snow with [11, 18, 33, 37, 69].

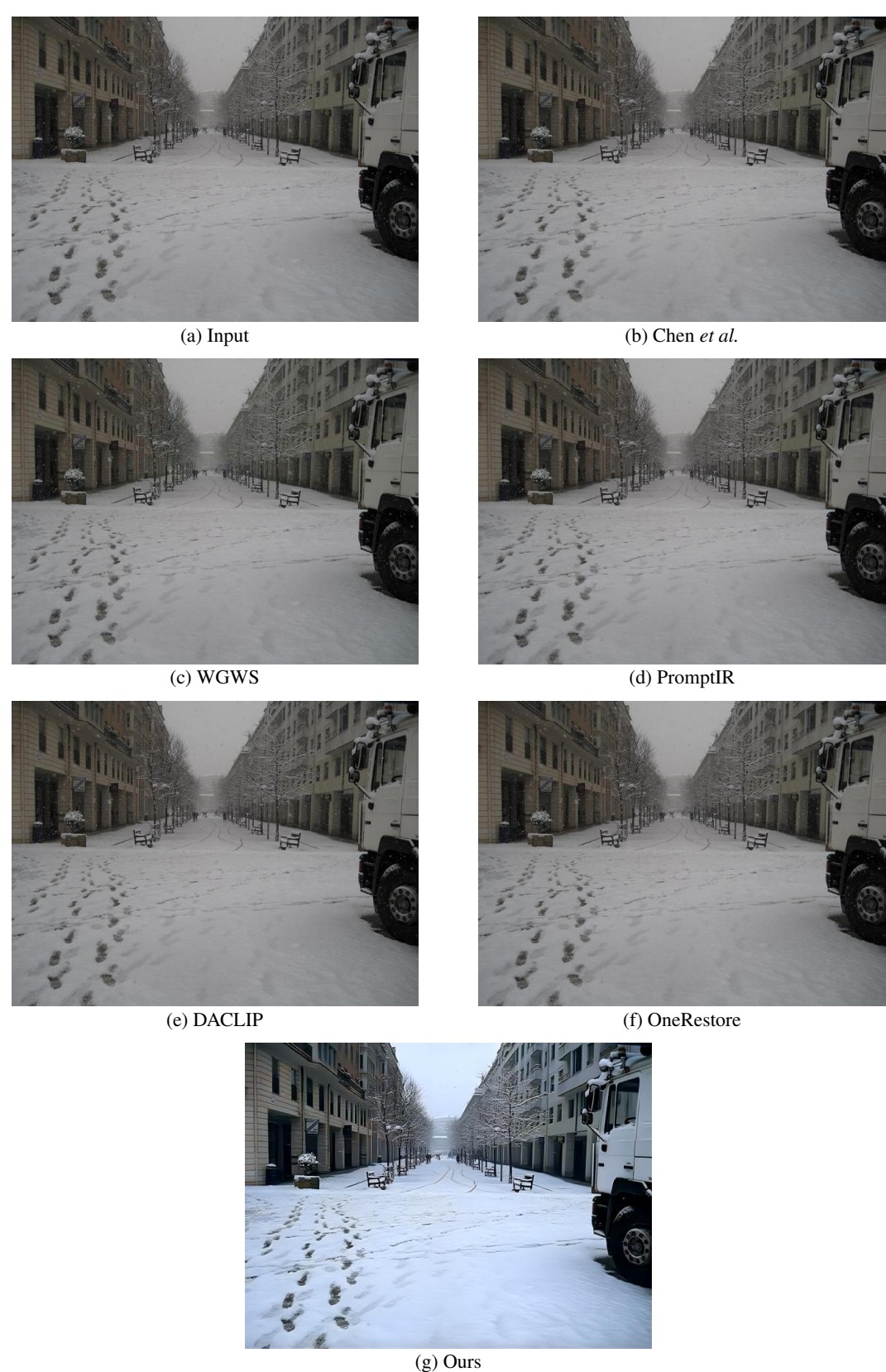

Figure 10: Visual comparison of real-world images under snow with [11, 18, 33, 37, 69].

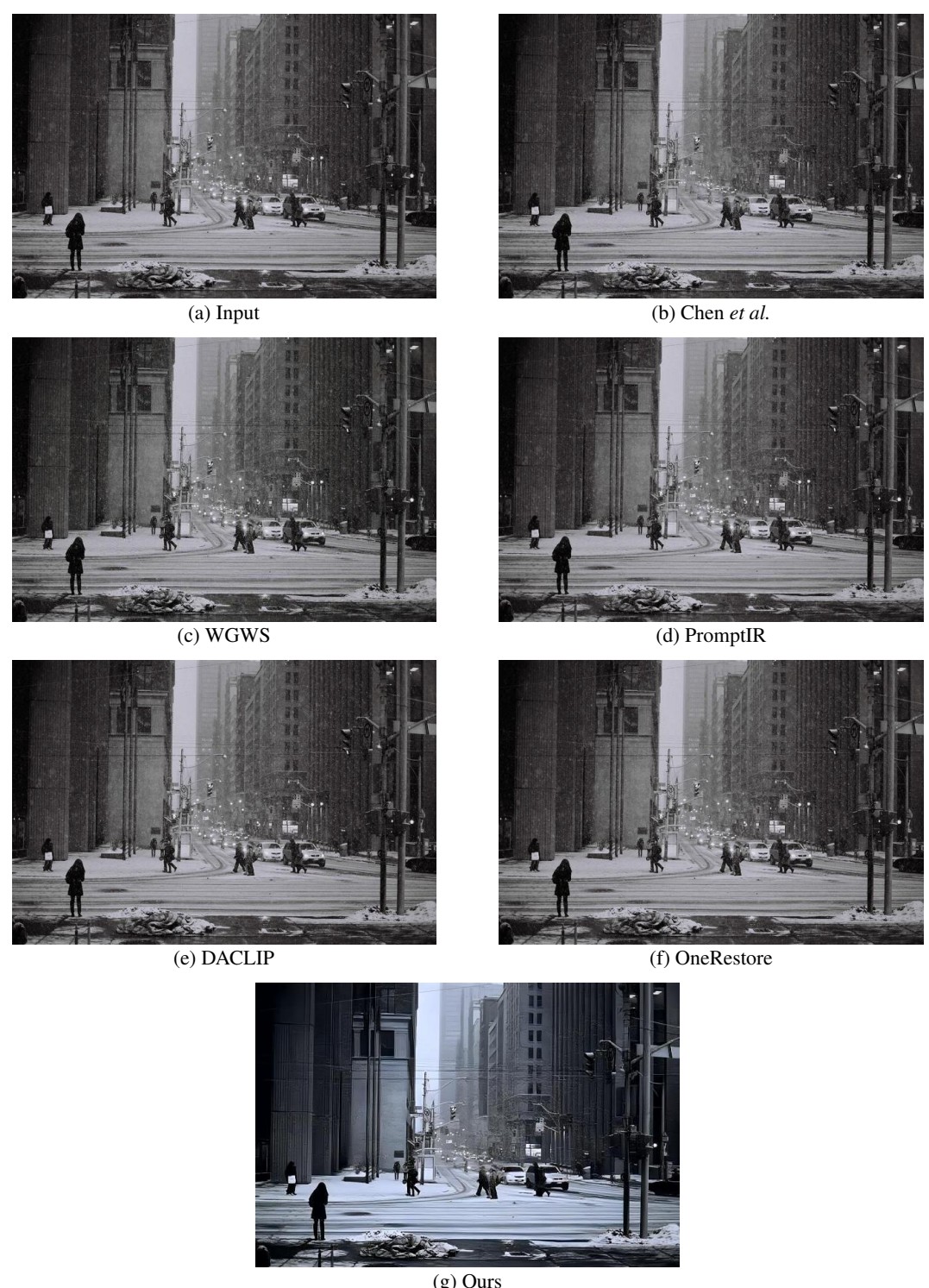

(a) Input

(b) Chen *et al.*

(c) WGWS

(d) PromptIR

(e) DACLIP

(f) OneRestore

(g) Ours

Figure 11: Visual comparison of real-world images under snow with [11, 18, 33, 37, 69].

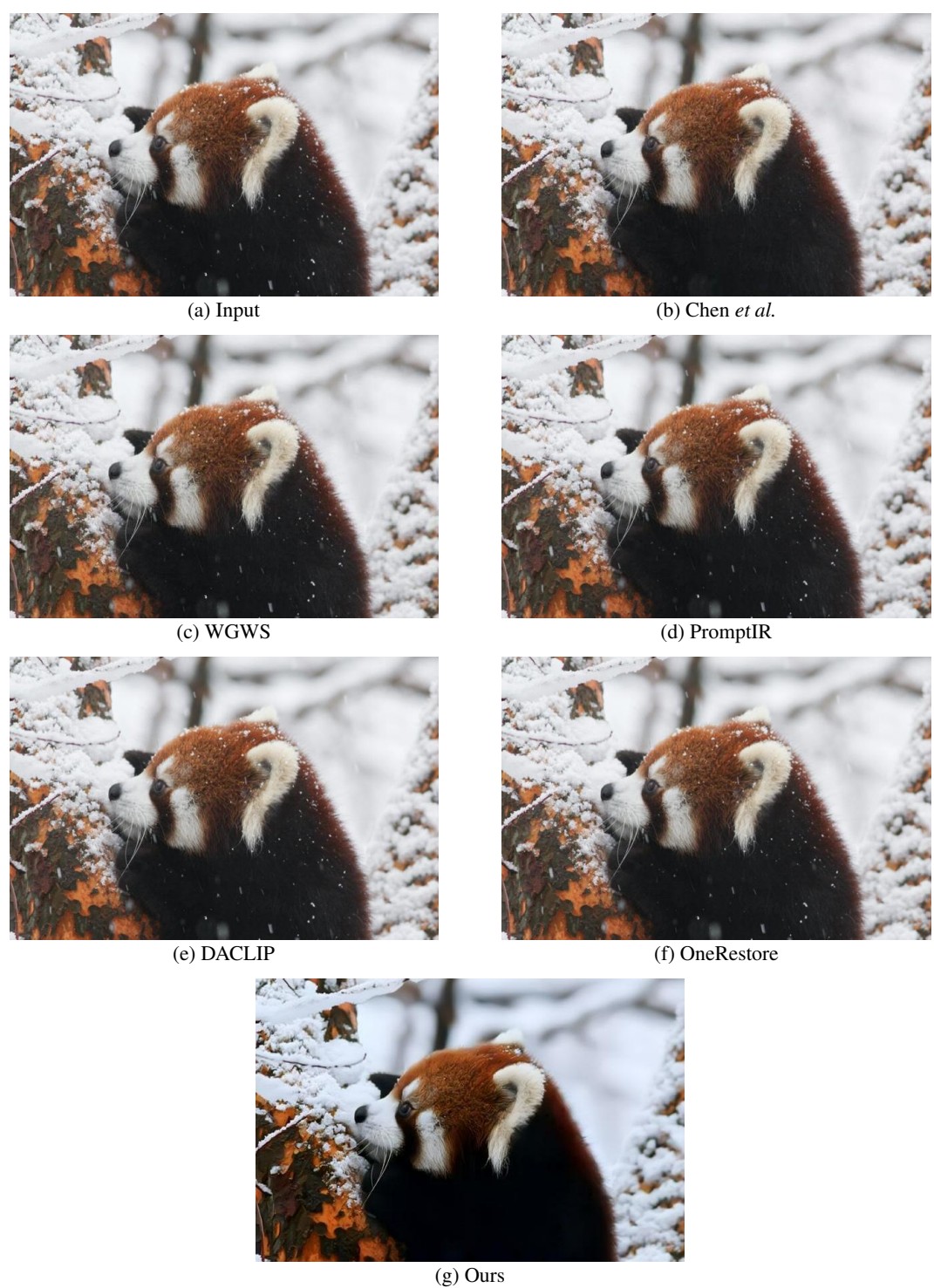

Figure 12: Visual comparison of real-world images under snow with [11, 18, 33, 37, 69].

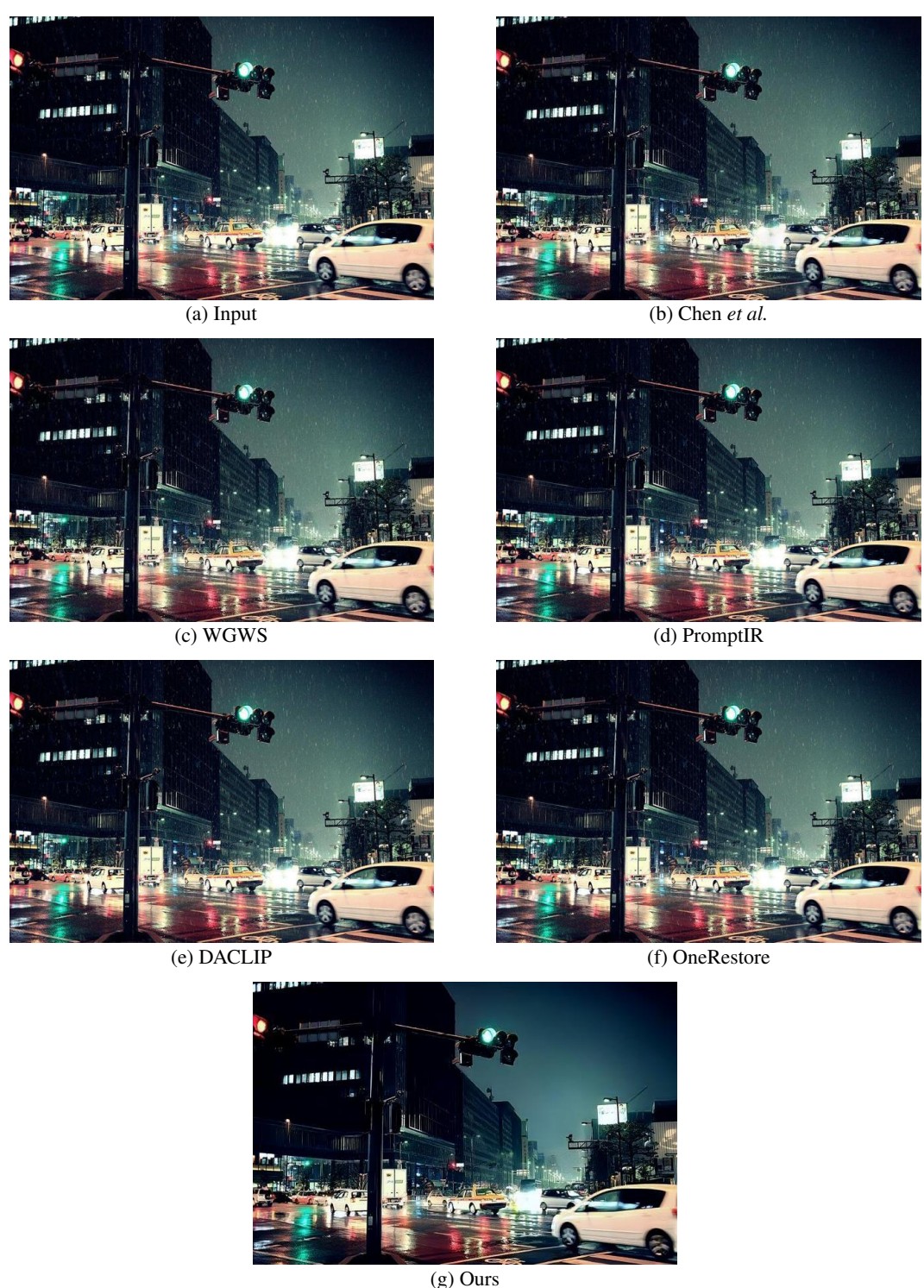

Figure 13: Visual comparison of real-world images under snow with [11, 18, 33, 37, 69].

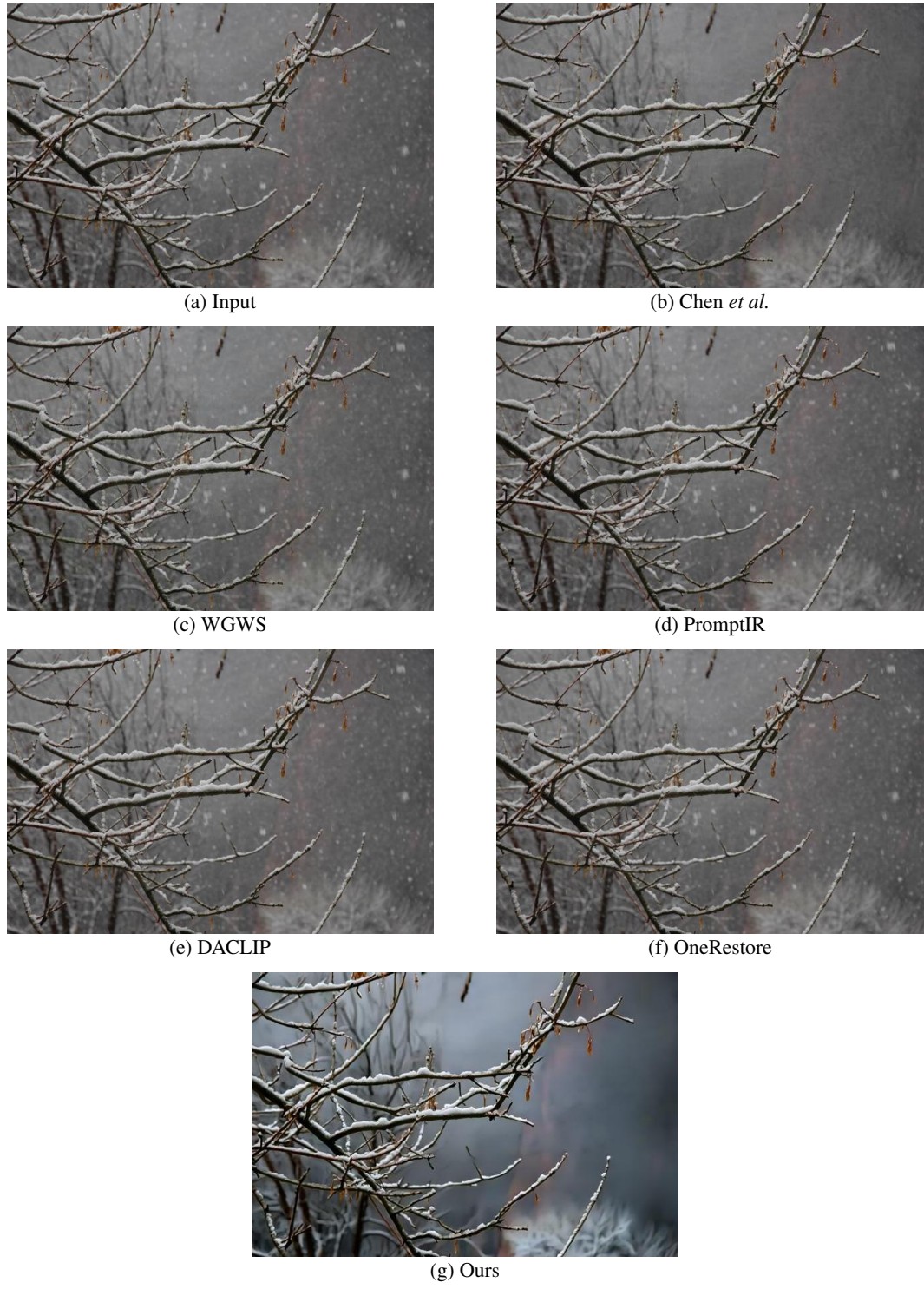

Figure 14: Visual comparison of real-world images under snow with [11, 18, 33, 37, 69].

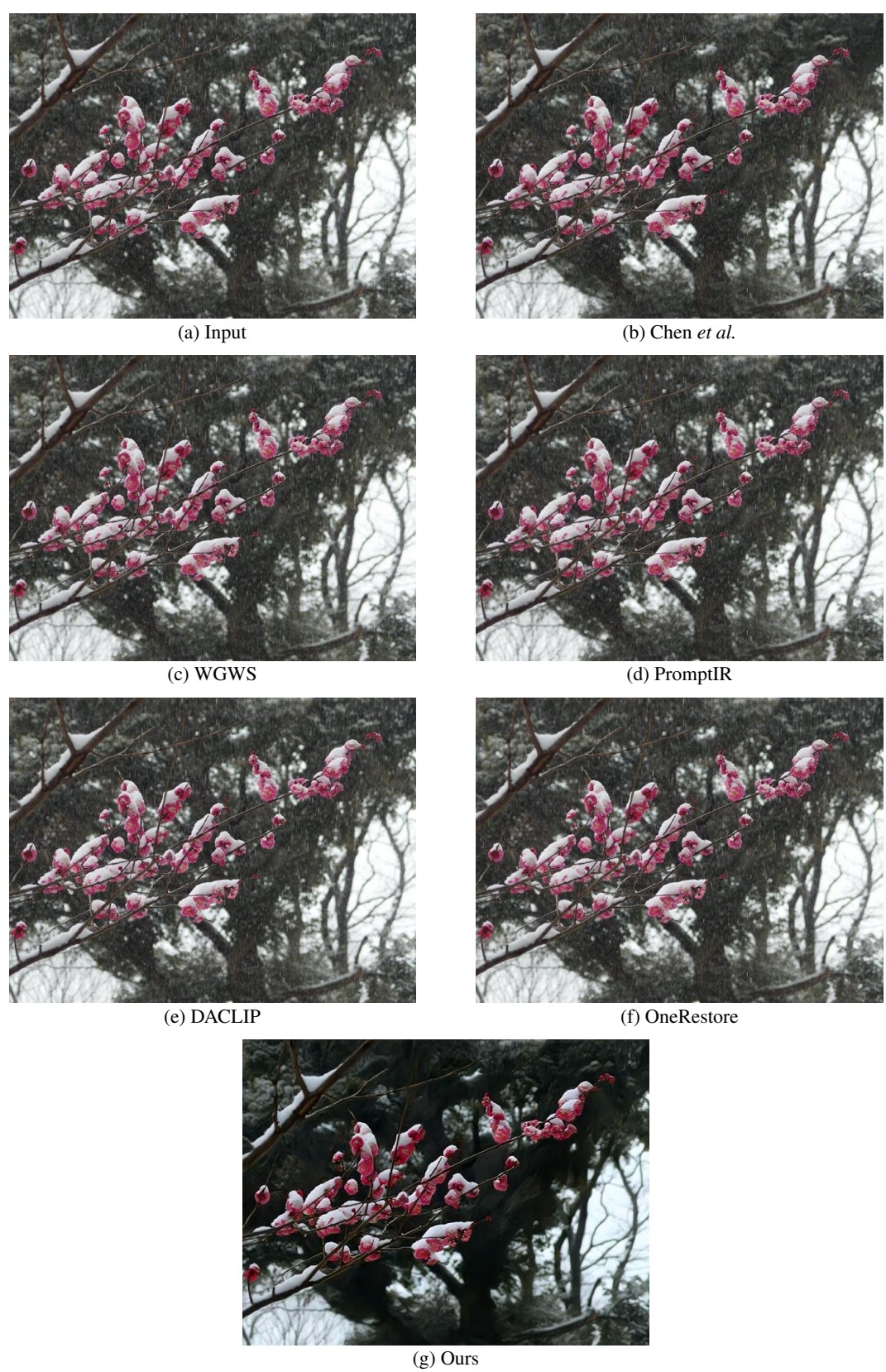

(a) Input

(b) Chen *et al.*

(c) WGWS

(d) PromptIR

(e) DACLIP

(f) OneRestore

(g) Ours

Figure 15: Visual comparison of real-world images under snow with [11, 18, 33, 37, 69].

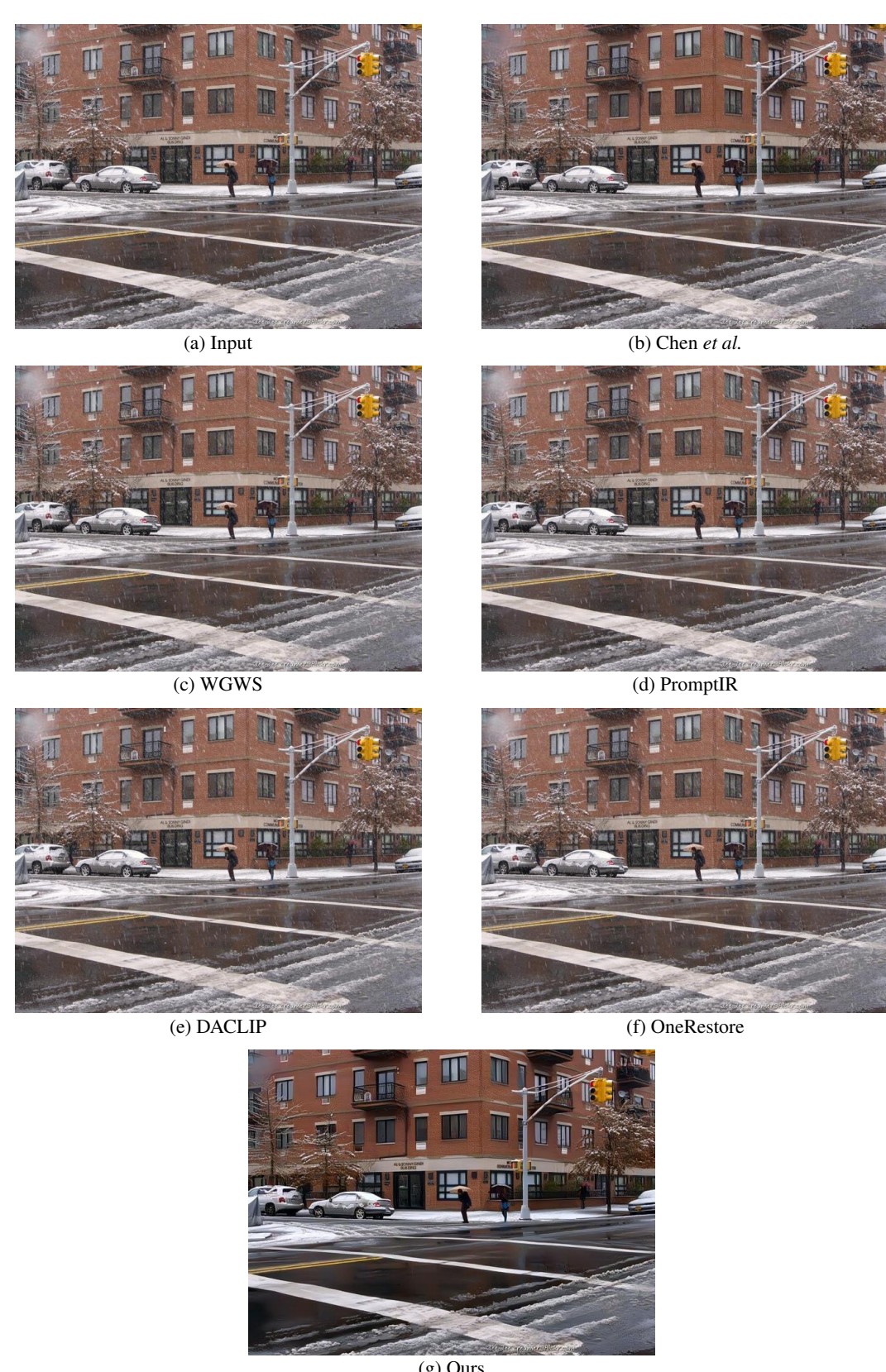

(a) Input

(b) Chen *et al.*

(c) WGWS

(d) PromptIR

(e) DACLIP

(f) OneRestore

(g) Ours

Figure 16: Visual comparison of real-world images under snow with [11, 18, 33, 37, 69].

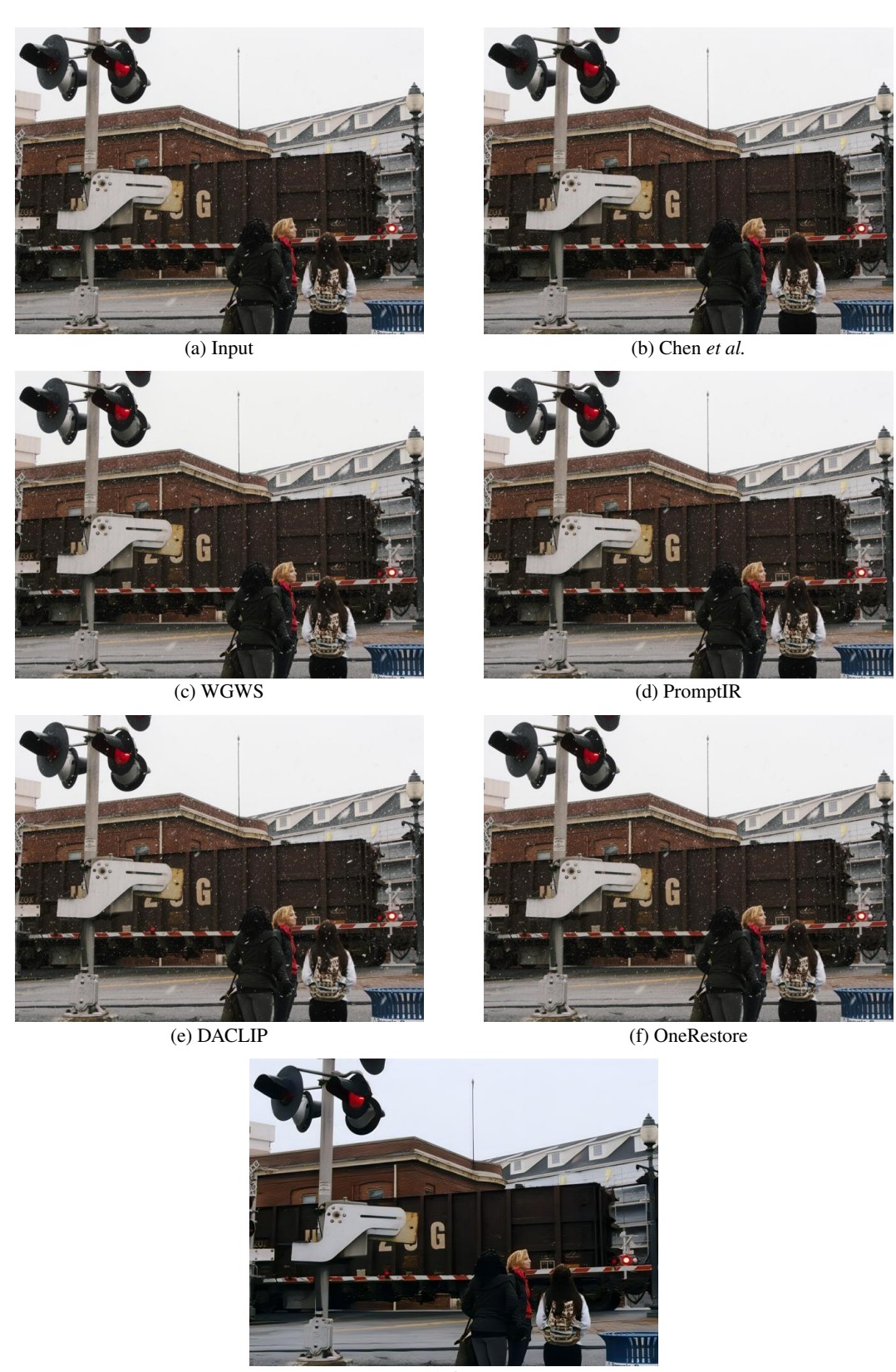

(a) Input

(b) Chen *et al.*

(c) WGWS

(d) PromptIR

(e) DACLIP

(f) OneRestore

(g) Ours

Figure 17: Visual comparison of real-world images under snow with [11, 18, 33, 37, 69].

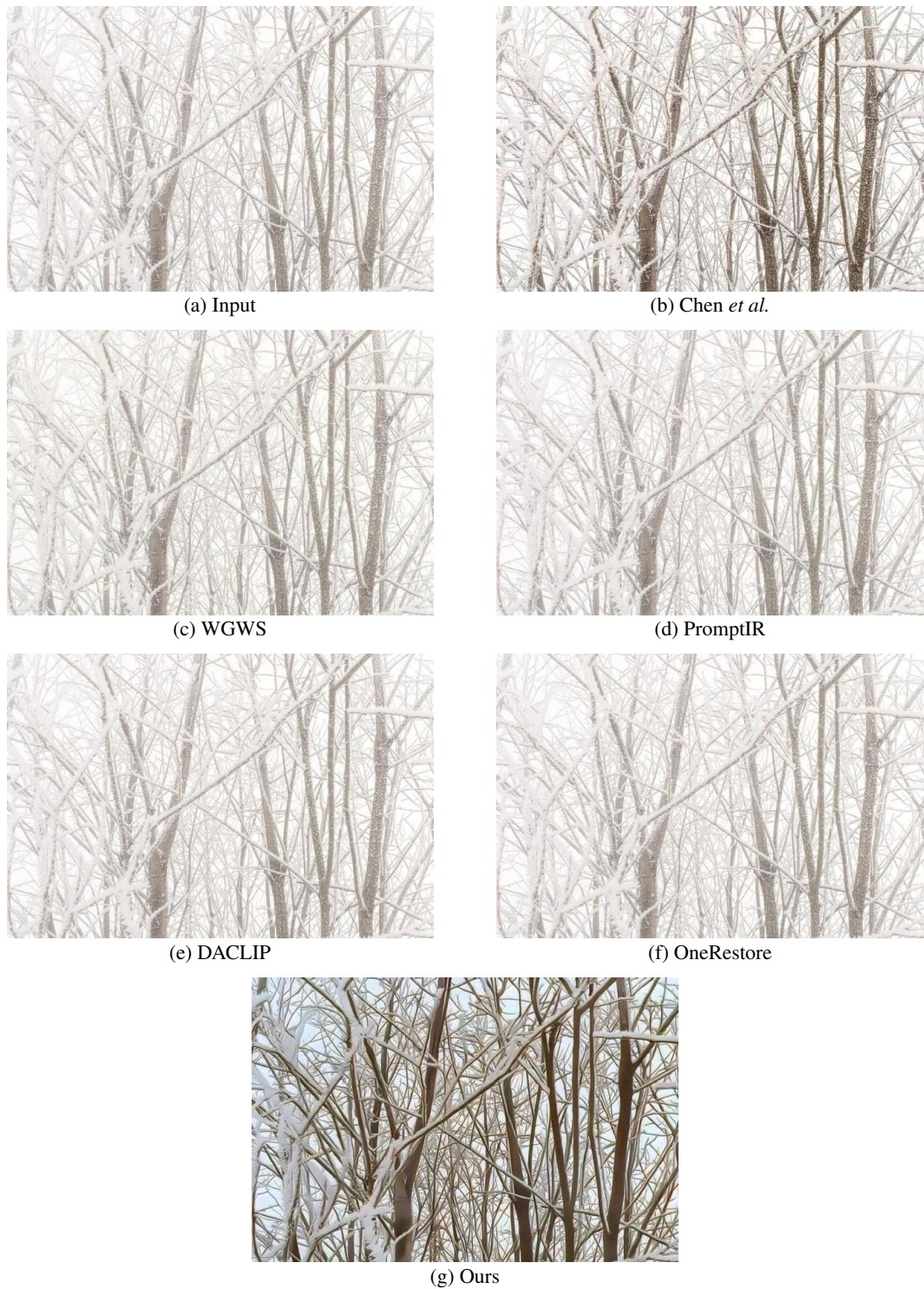

(a) Input        (b) Chen *et al.*

(c) WGWS        (d) PromptIR

(e) DACLIP        (f) OneRestore

(g) Ours

Figure 18: Visual comparison of real-world images under snow with [11, 18, 33, 37, 69].

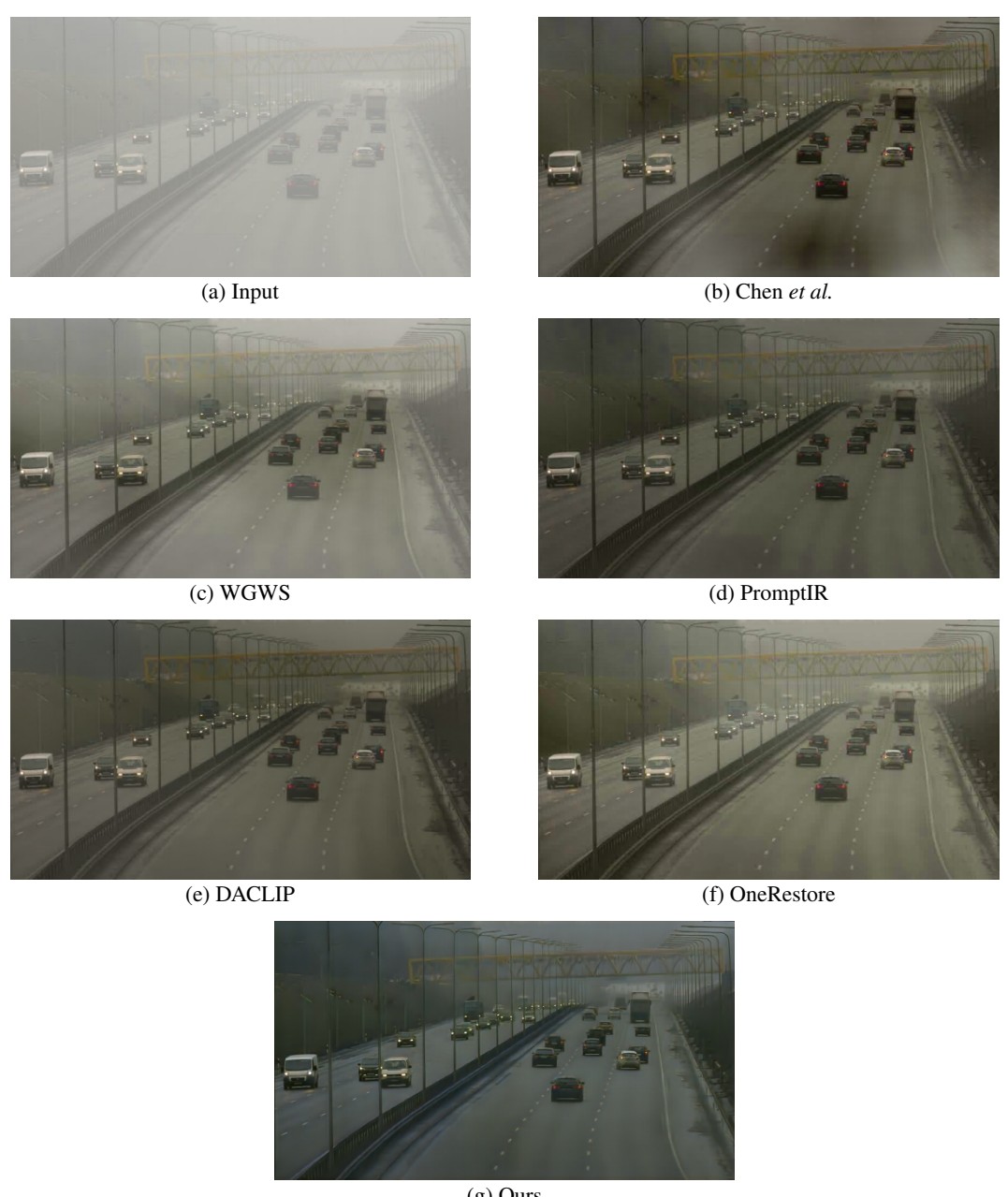

(a) Input

(b) Chen *et al.*

(c) WGWS

(d) PromptIR

(e) DACLIP

(f) OneRestore

(g) Ours

Figure 19: Visual comparison of real-world images under haze with [11, 18, 33, 37, 69].

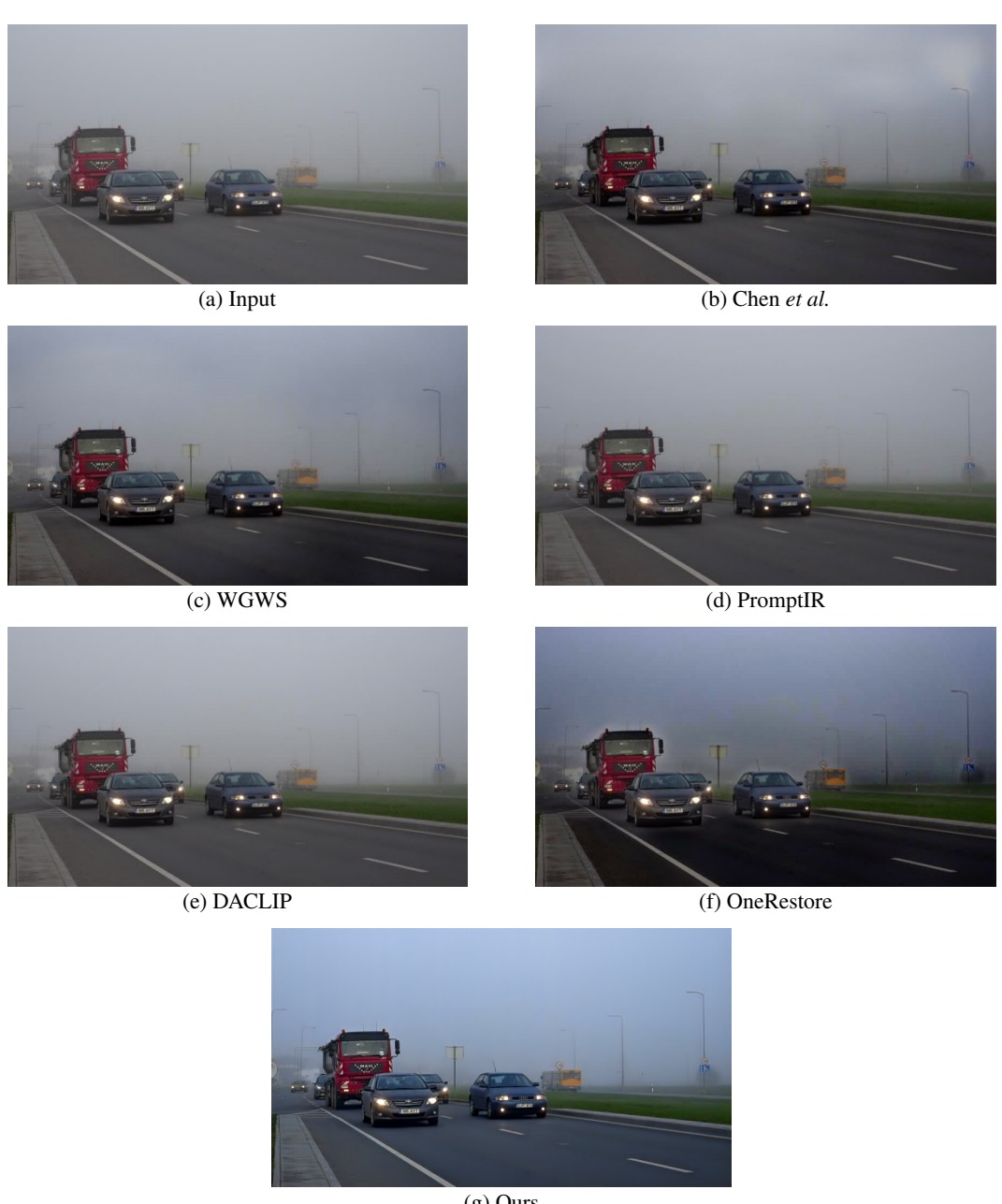

(a) Input

(b) Chen *et al.*

(c) WGWS

(d) PromptIR

(e) DACLIP

(f) OneRestore

(g) Ours

Figure 20: Visual comparison of real-world images under haze with [11, 18, 33, 37, 69].

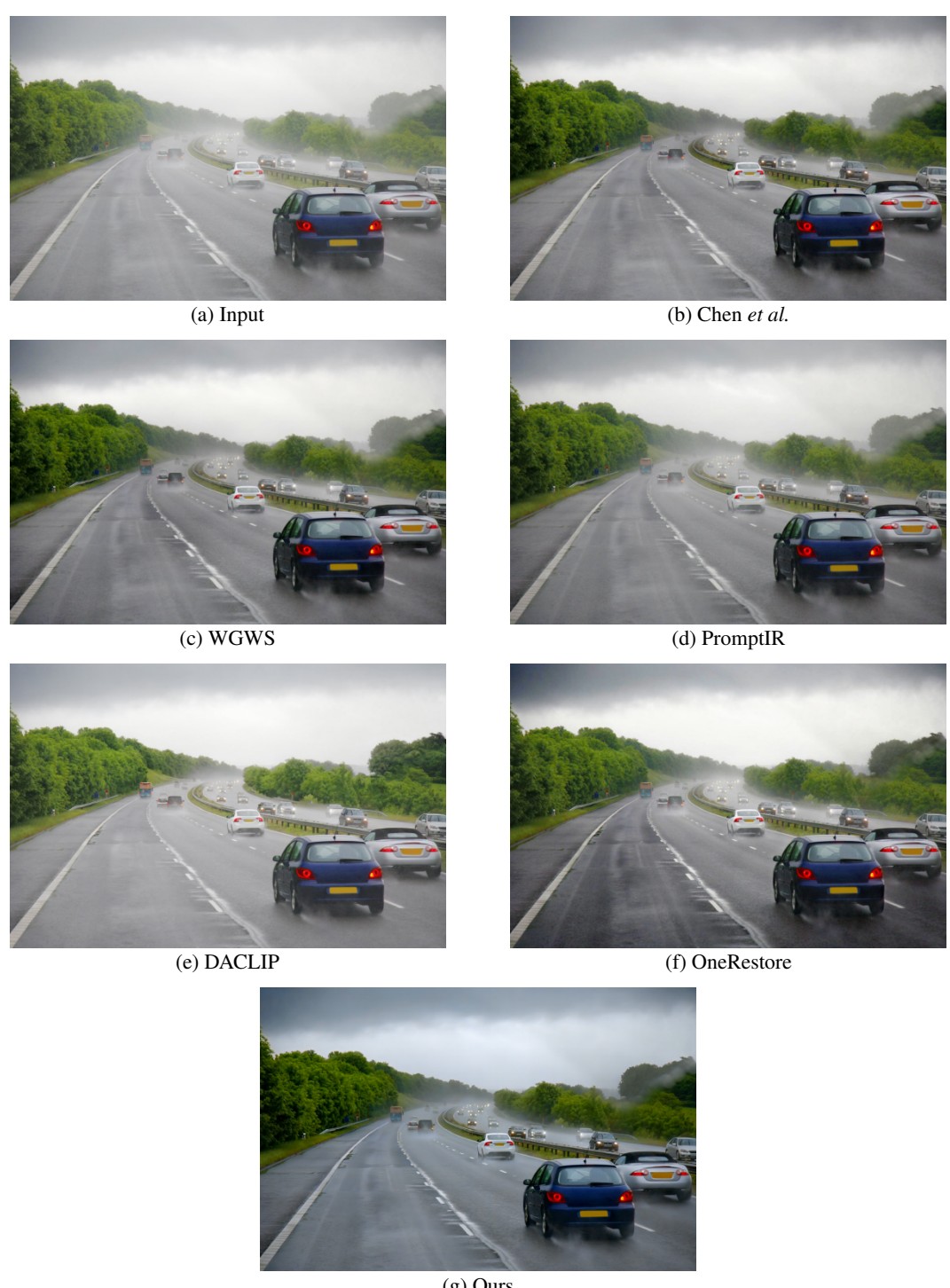

(a) Input          (b) Chen *et al.*

(c) WGWS          (d) PromptIR

(e) DACLIP          (f) OneRestore

(g) Ours

Figure 21: Visual comparison of real-world images under haze with [11, 18, 33, 37, 69].

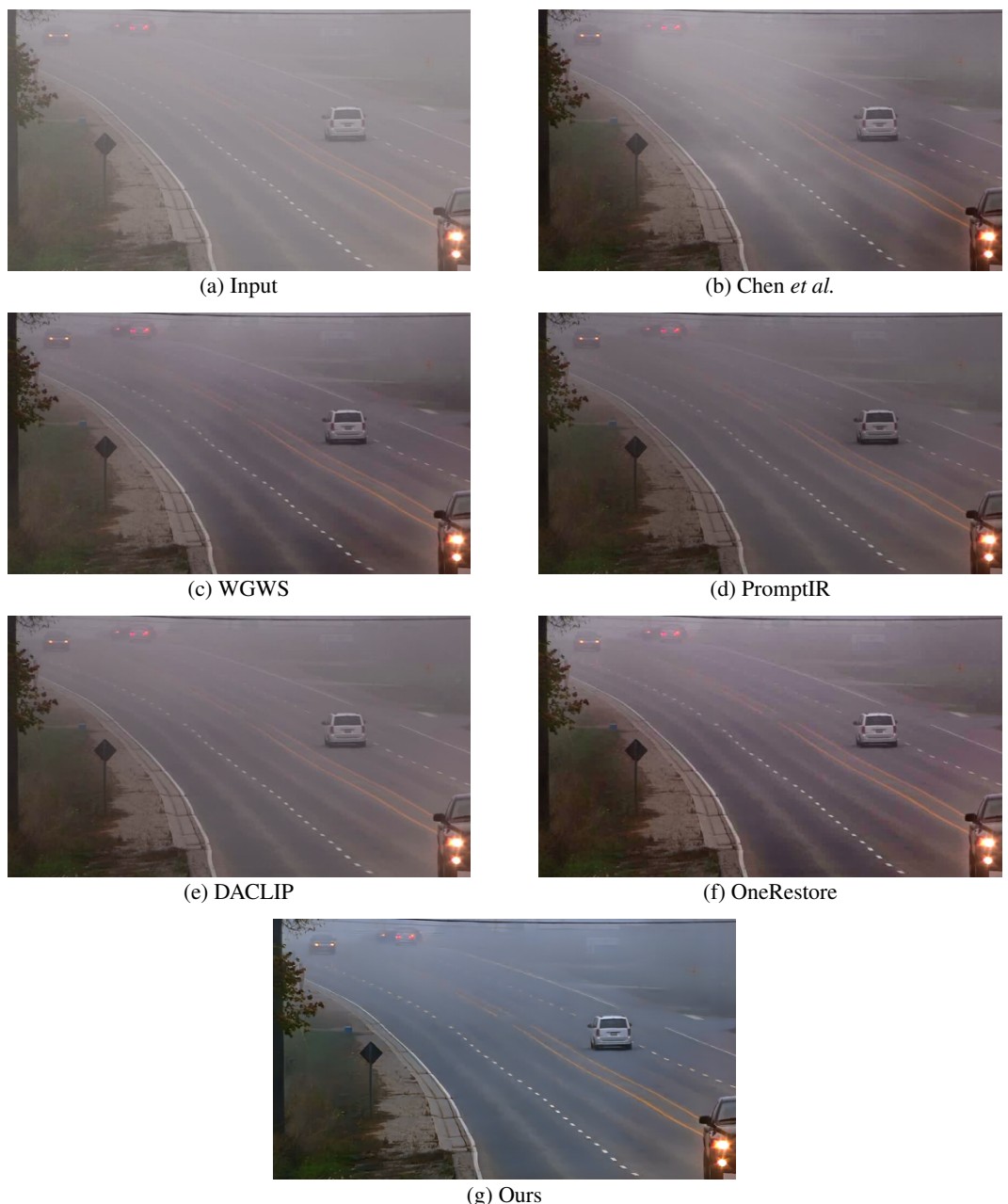

(a) Input

(b) Chen *et al.*

(c) WGWS

(d) PromptIR

(e) DACLIP

(f) OneRestore

(g) Ours

Figure 22: Visual comparison of real-world images under haze with [11, 18, 33, 37, 69].

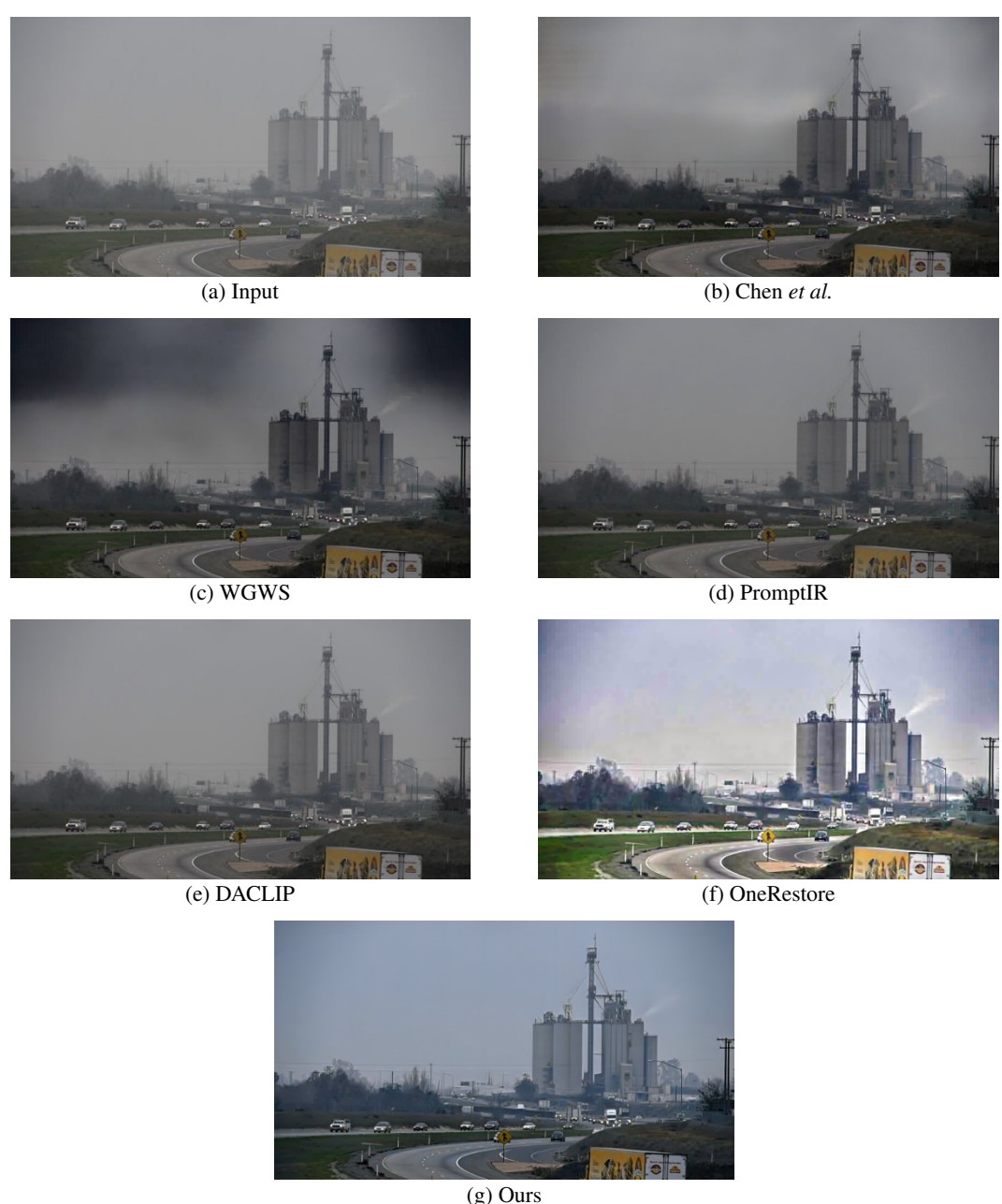

(a) Input

(b) Chen *et al.*

(c) WGWS

(d) PromptIR

(e) DACLIP

(f) OneRestore

(g) Ours

Figure 23: Visual comparison of real-world images under haze with [11, 18, 33, 37, 69].

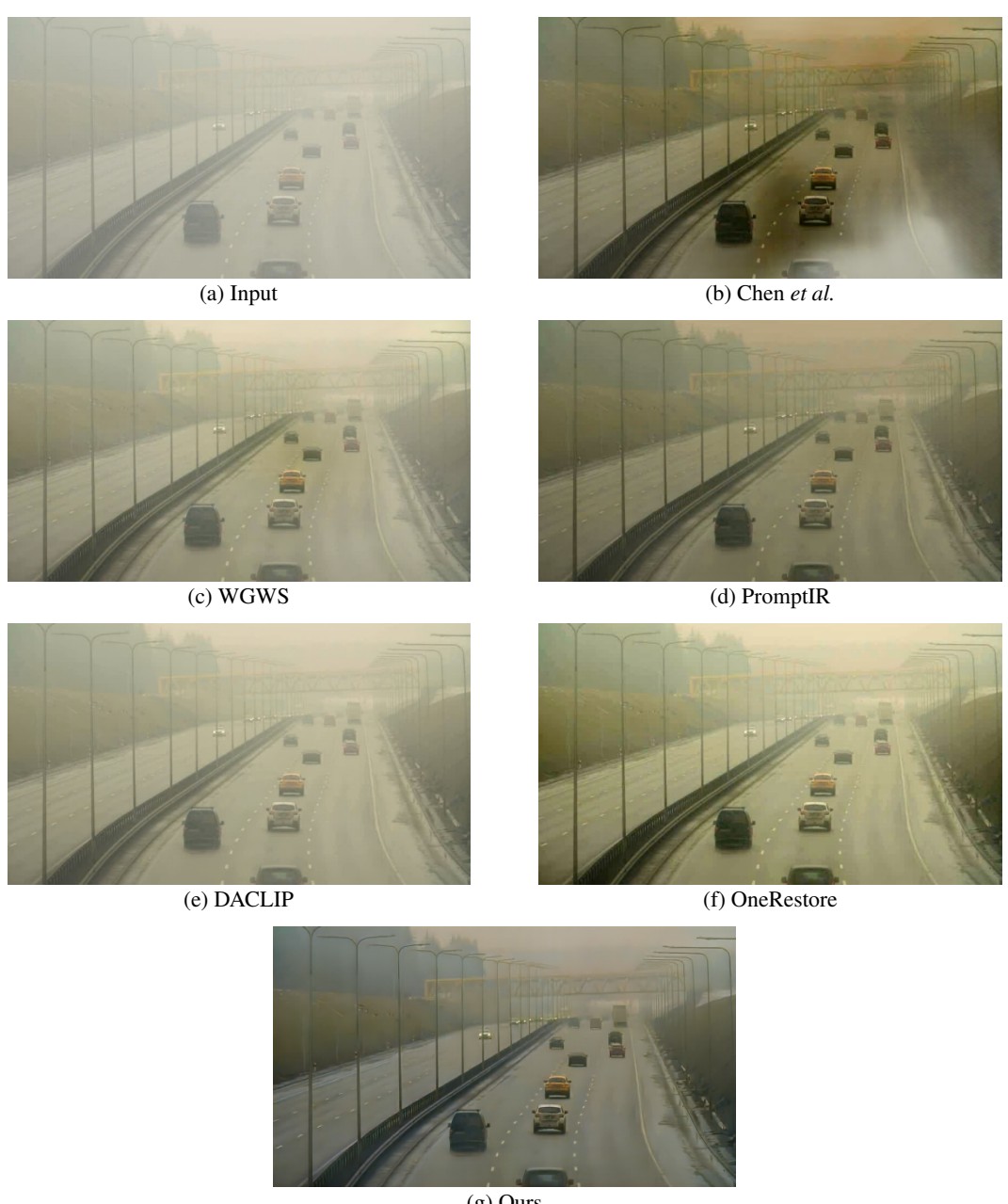

(a) Input

(b) Chen *et al.*

(c) WGWS

(d) PromptIR

(e) DACLIP

(f) OneRestore

(g) Ours

Figure 24: Visual comparison of real-world images under haze with [11, 18, 33, 37, 69].

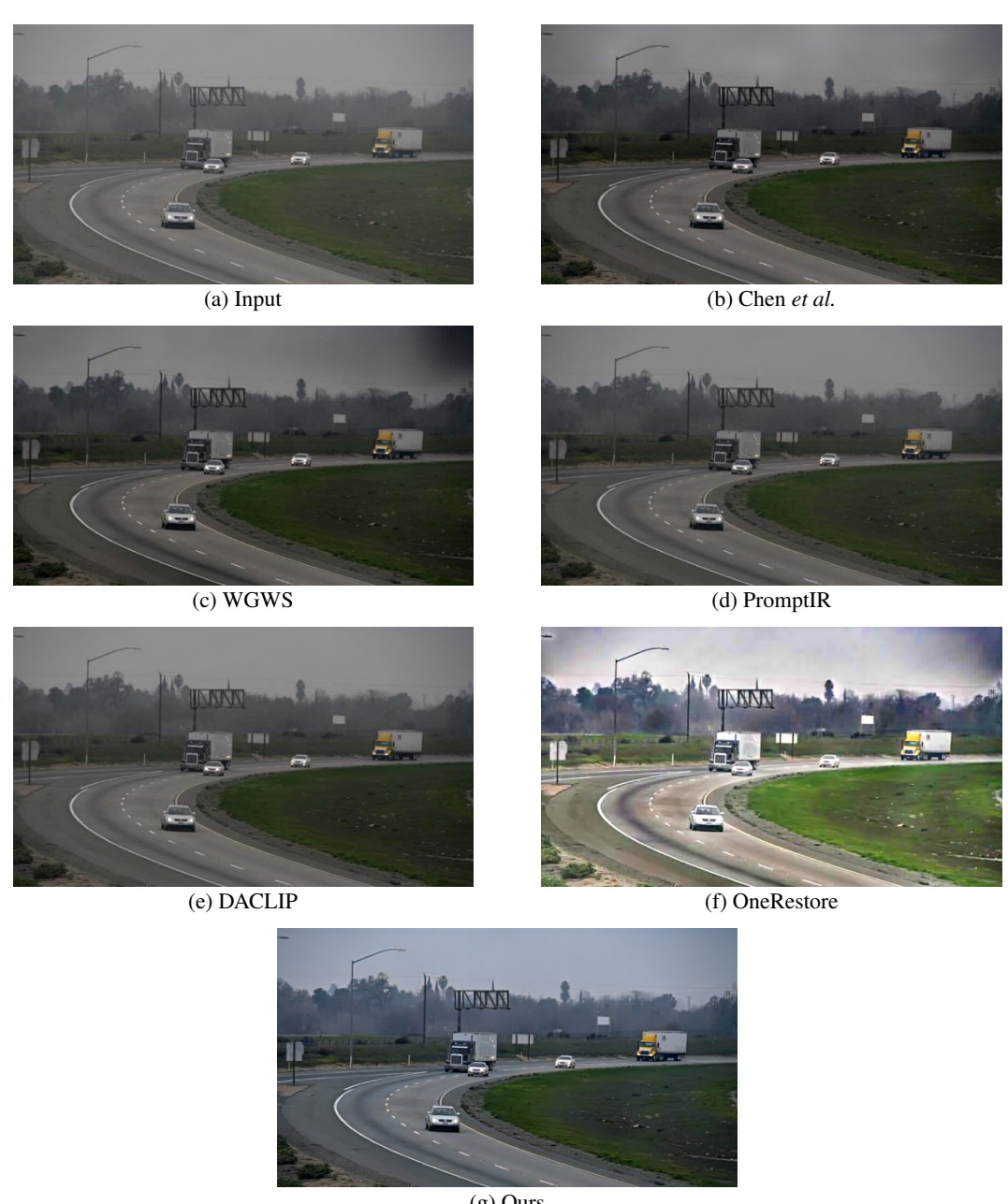

(a) Input

(b) Chen *et al.*

(c) WGWS

(d) PromptIR

(e) DACLIP

(f) OneRestore

(g) Ours

Figure 25: Visual comparison of real-world images under haze with [11, 18, 33, 37, 69].

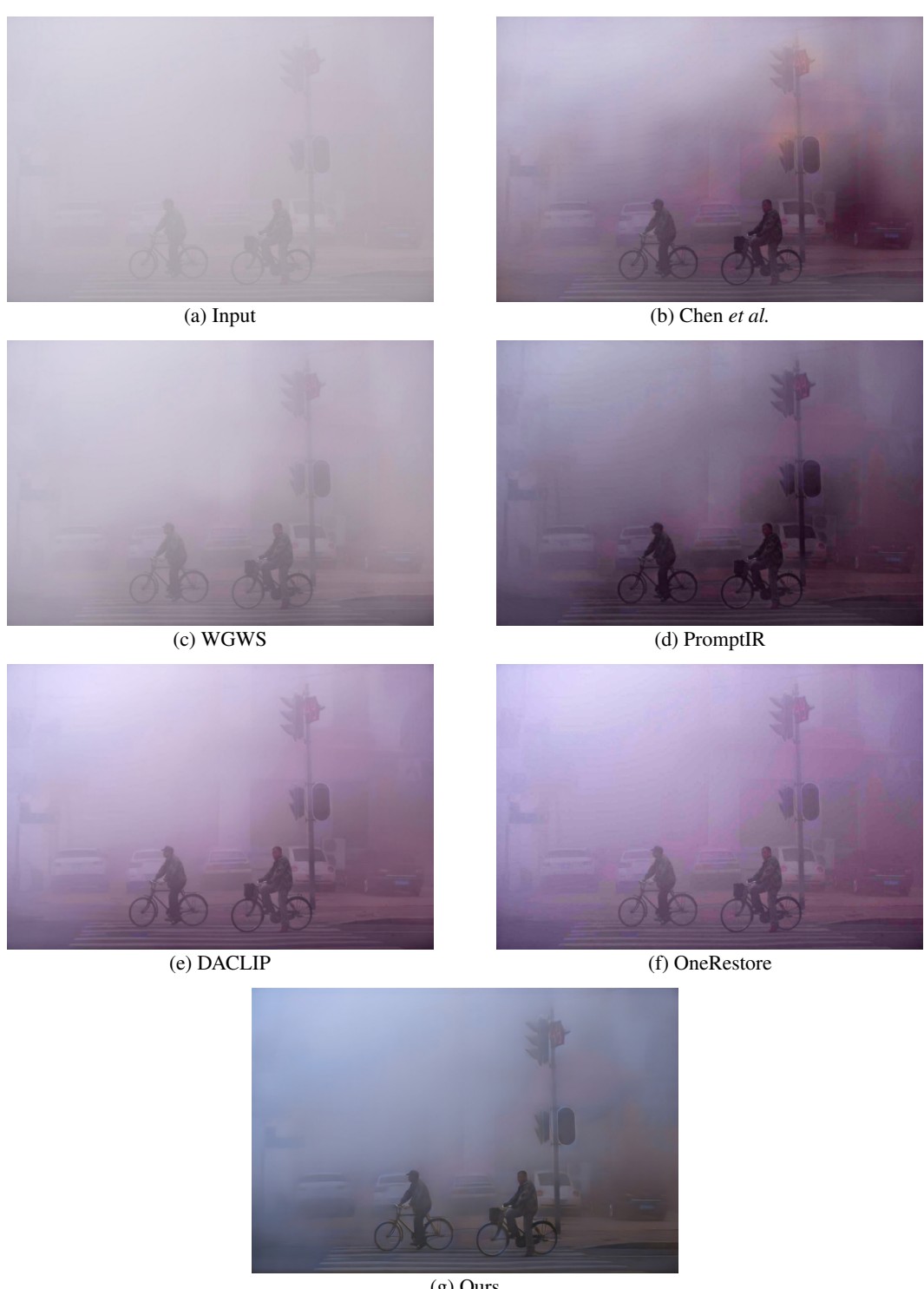

Figure 26: Visual comparison of real-world images under haze with [11, 18, 33, 37, 69].

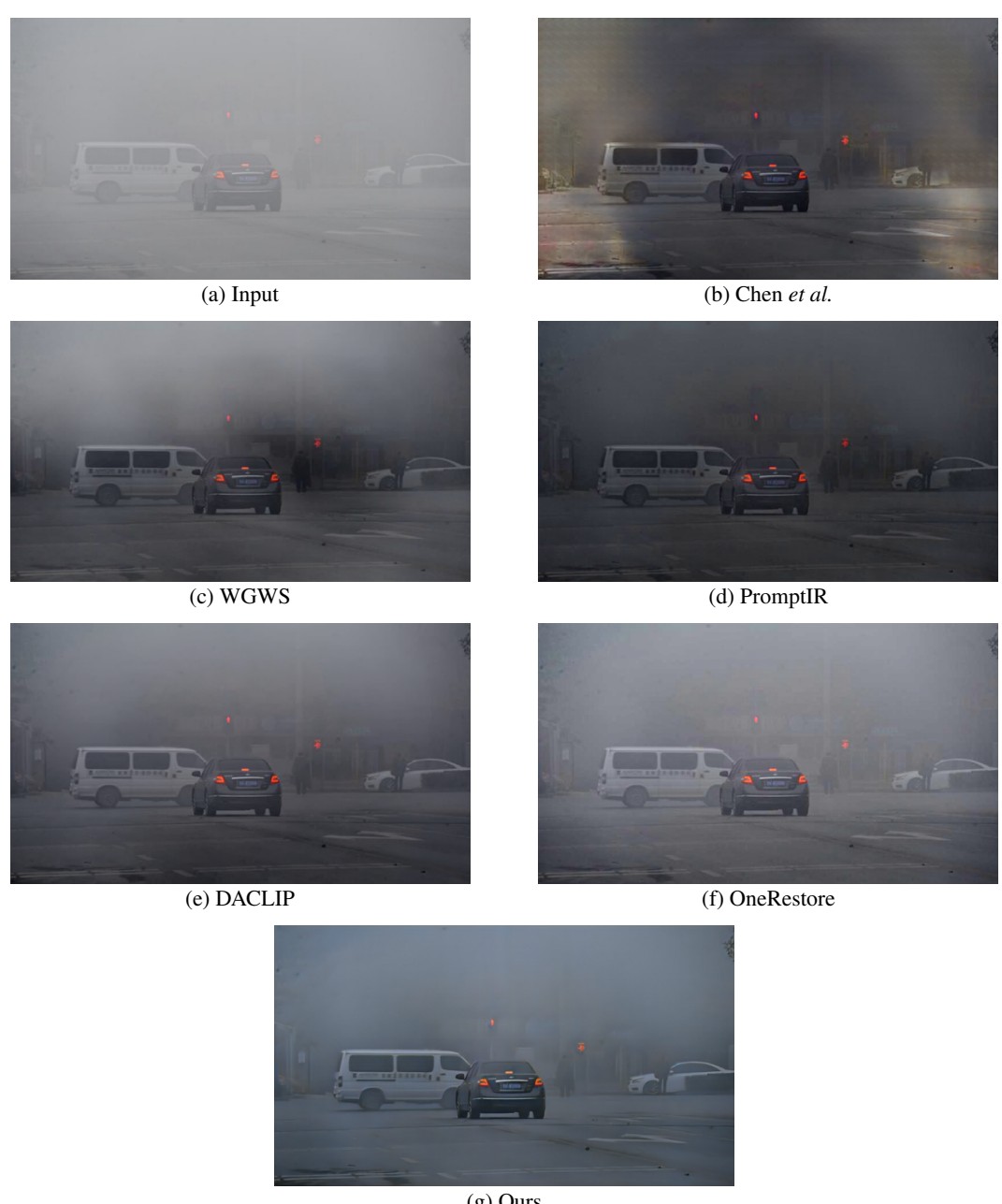

(a) Input

(b) Chen *et al.*

(c) WGWS

(d) PromptIR

(e) DACLIP

(f) OneRestore

(g) Ours

Figure 27: Visual comparison of real-world images under haze with [11, 18, 33, 37, 69].

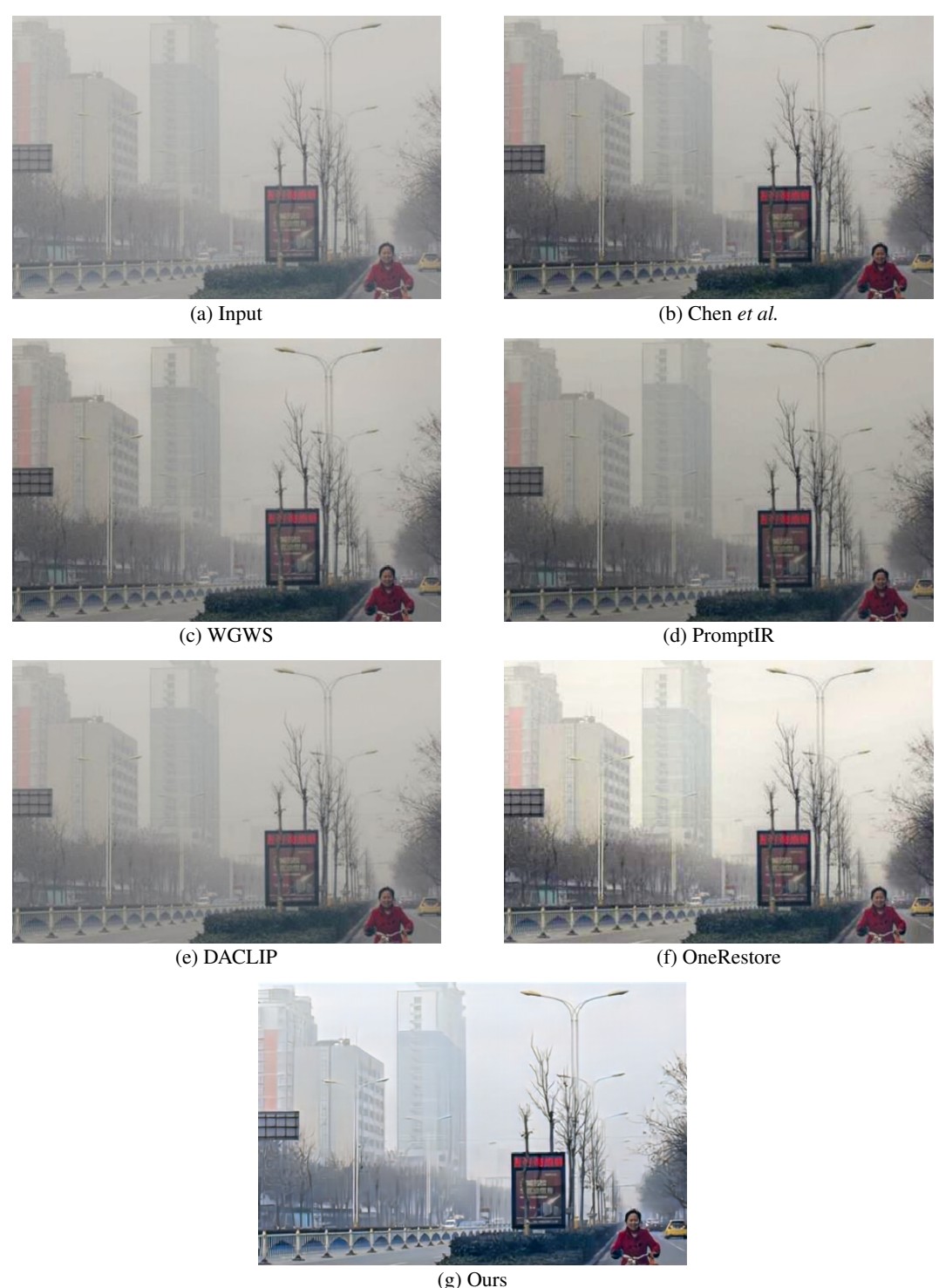

(a) Input

(b) Chen *et al.*

(c) WGWS

(d) PromptIR

(e) DACLIP

(f) OneRestore

(g) Ours

Figure 28: Visual comparison of real-world images under haze with [11, 18, 33, 37, 69].

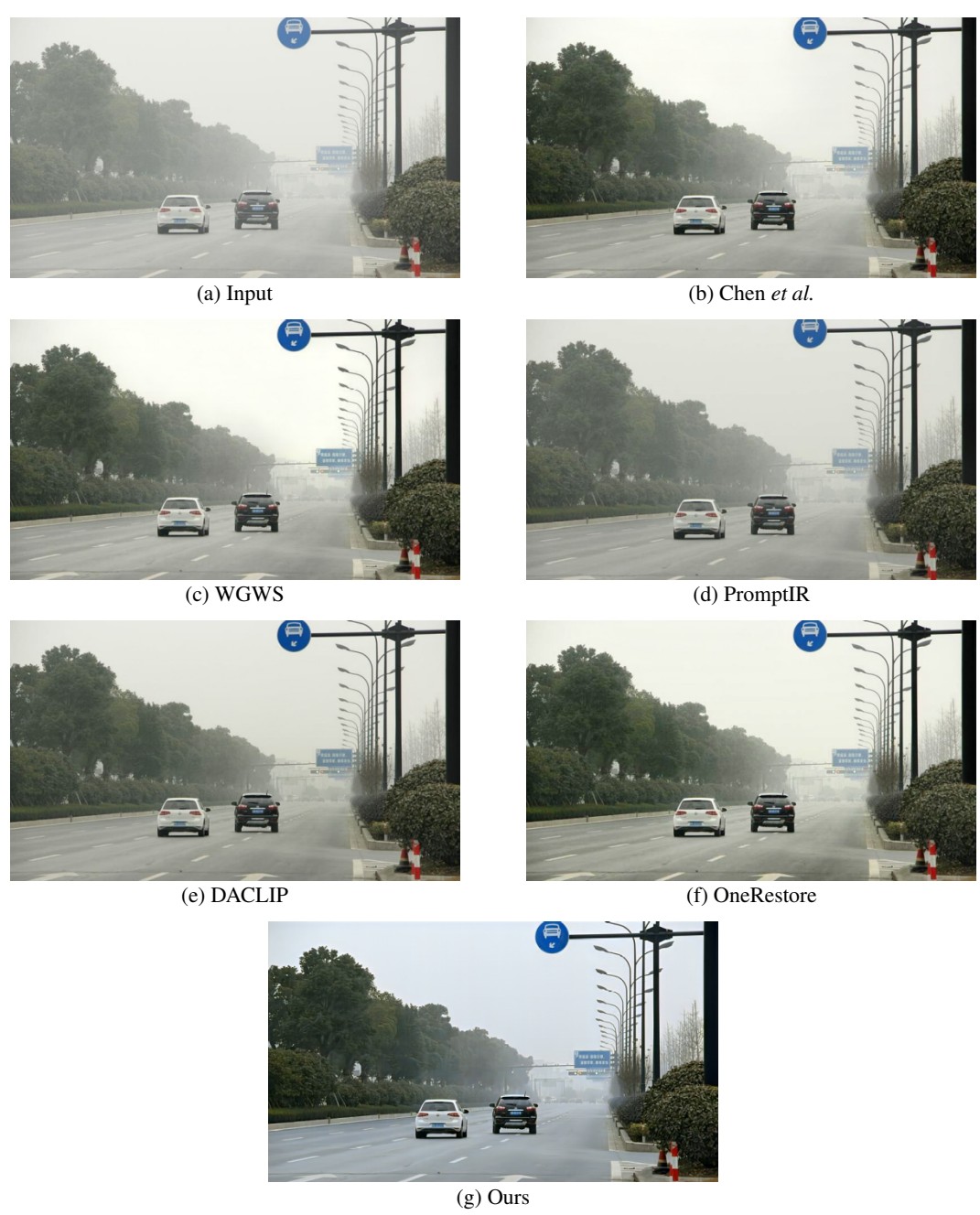

Figure 29: Visual comparison of real-world images under haze with [11, 18, 33, 37, 69].

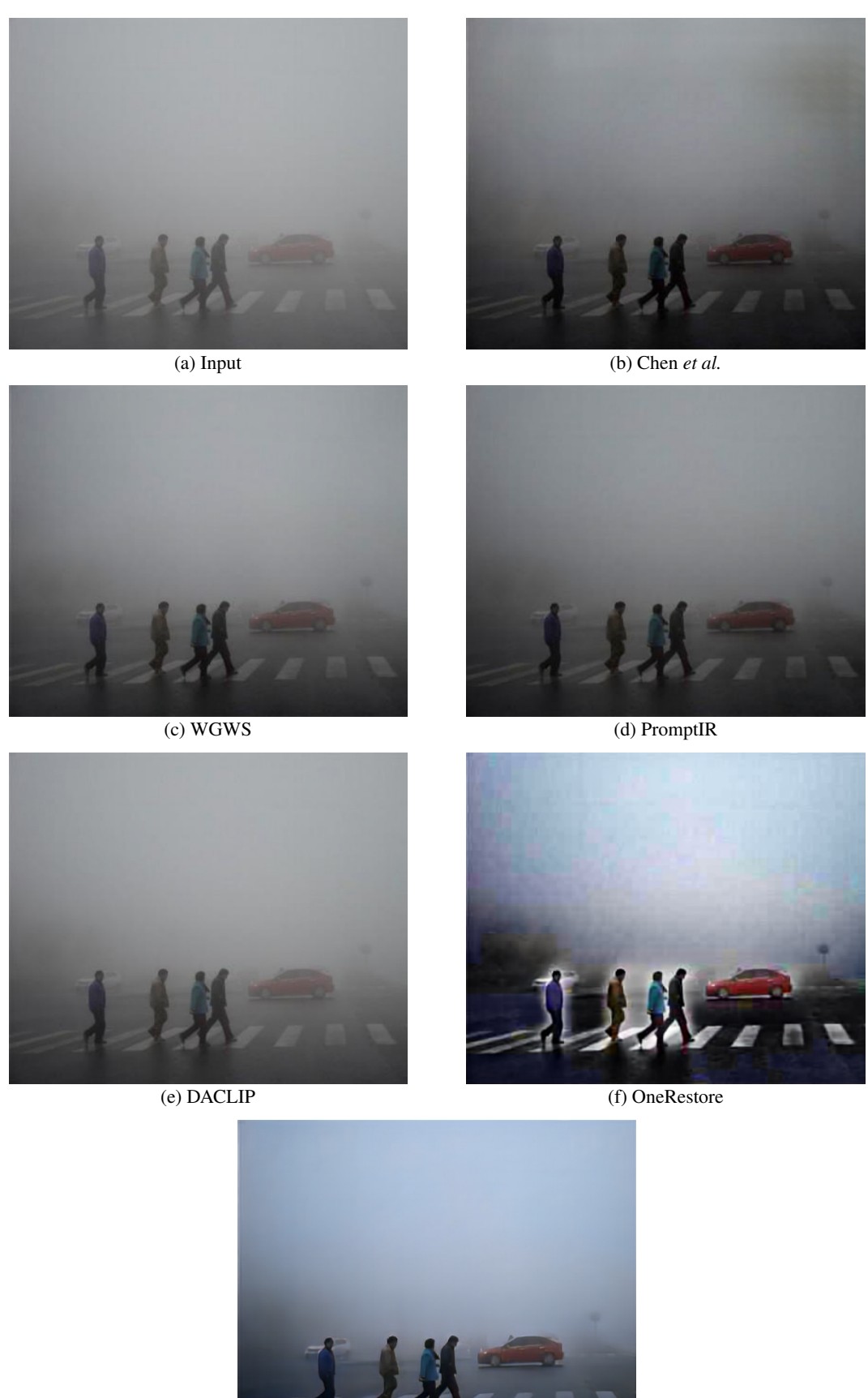

(a) Input

(b) Chen *et al.*

(c) WGWS

(d) PromptIR

(e) DACLIP

(f) OneRestore

(g) Ours

Figure 30: Visual comparison of real-world images under haze with [11, 18, 33, 37, 69].

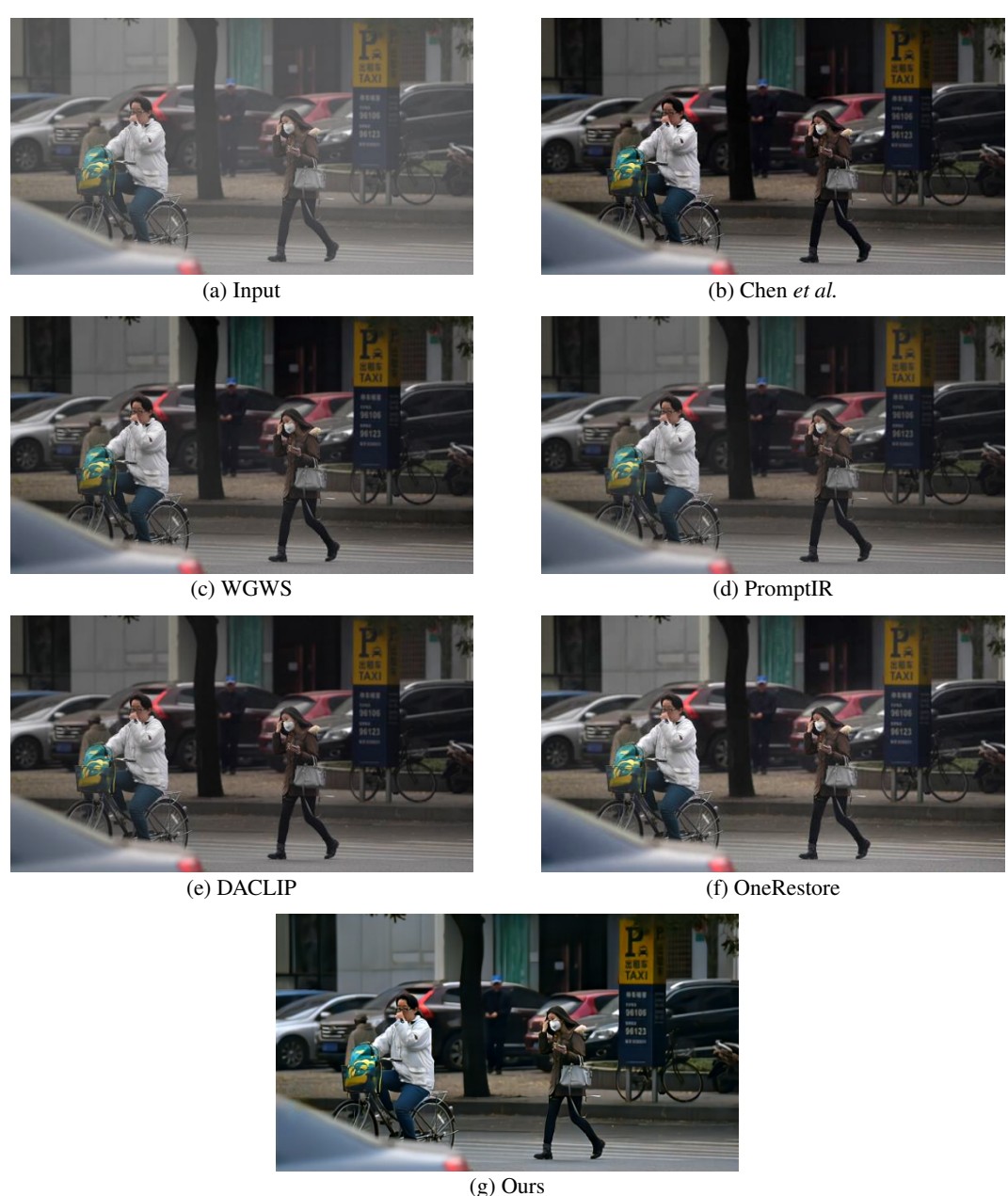

(a) Input

(b) Chen *et al.*

(c) WGWS

(d) PromptIR

(e) DACLIP

(f) OneRestore

(g) Ours

Figure 31: Visual comparison of real-world images under haze with [11, 18, 33, 37, 69].

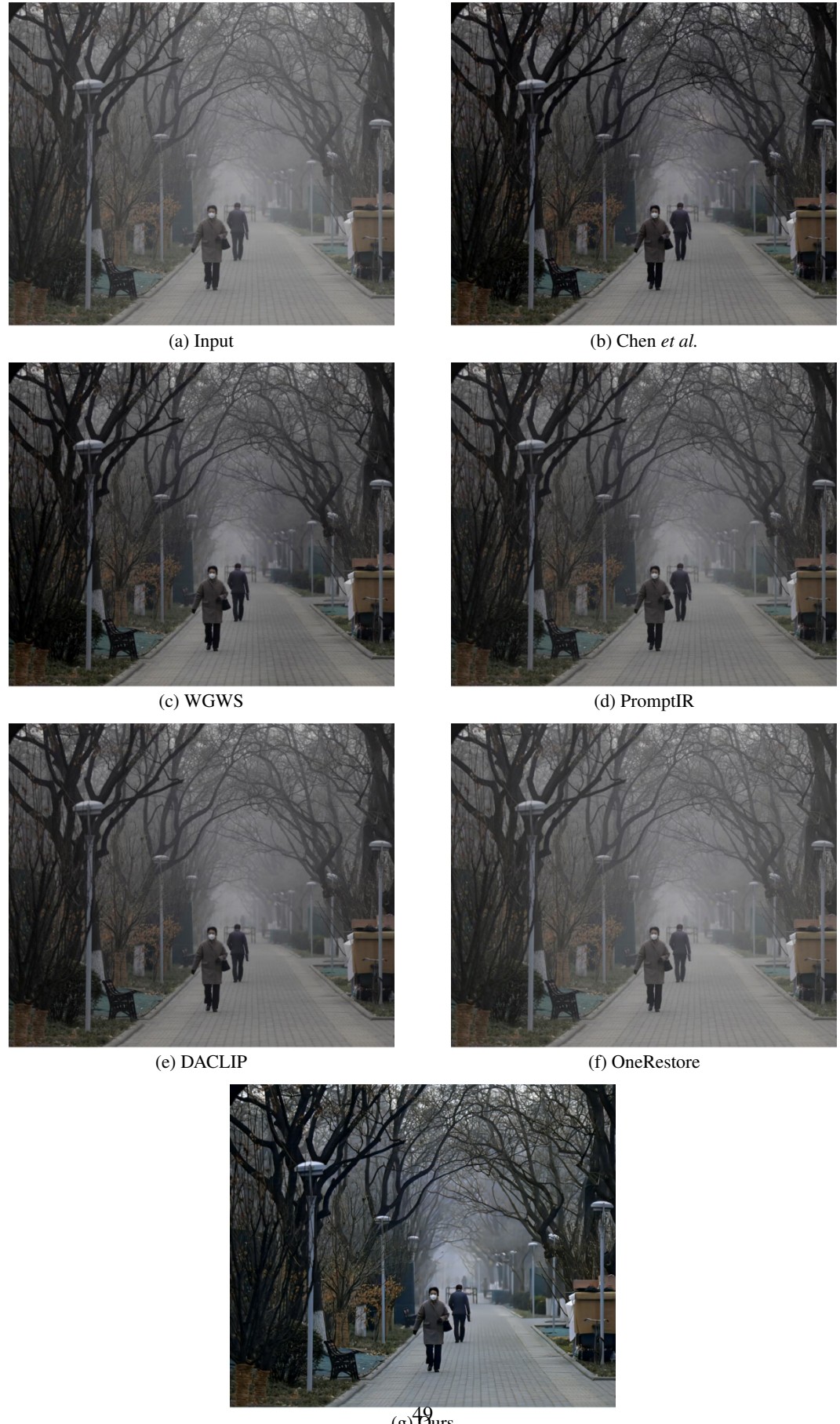

(a) Input

(b) Chen *et al.*

(c) WGWS

(d) PromptIR

(e) DACLIP

(f) OneRestore

(g) Ours

Figure 32: Visual comparison of real-world images under haze with [11, 18, 33, 37, 69].

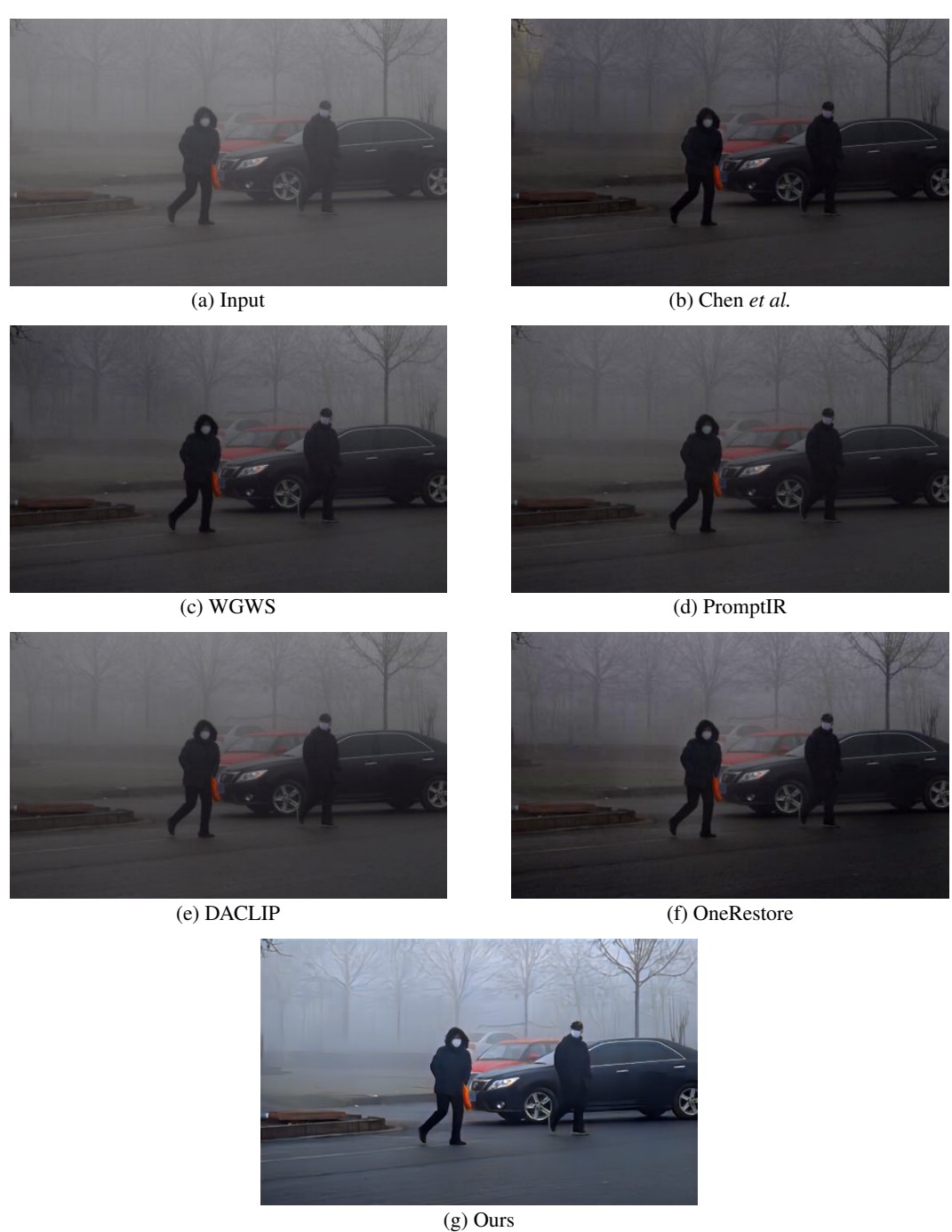

Figure 33: Visual comparison of real-world images under haze with [11, 18, 33, 37, 69].

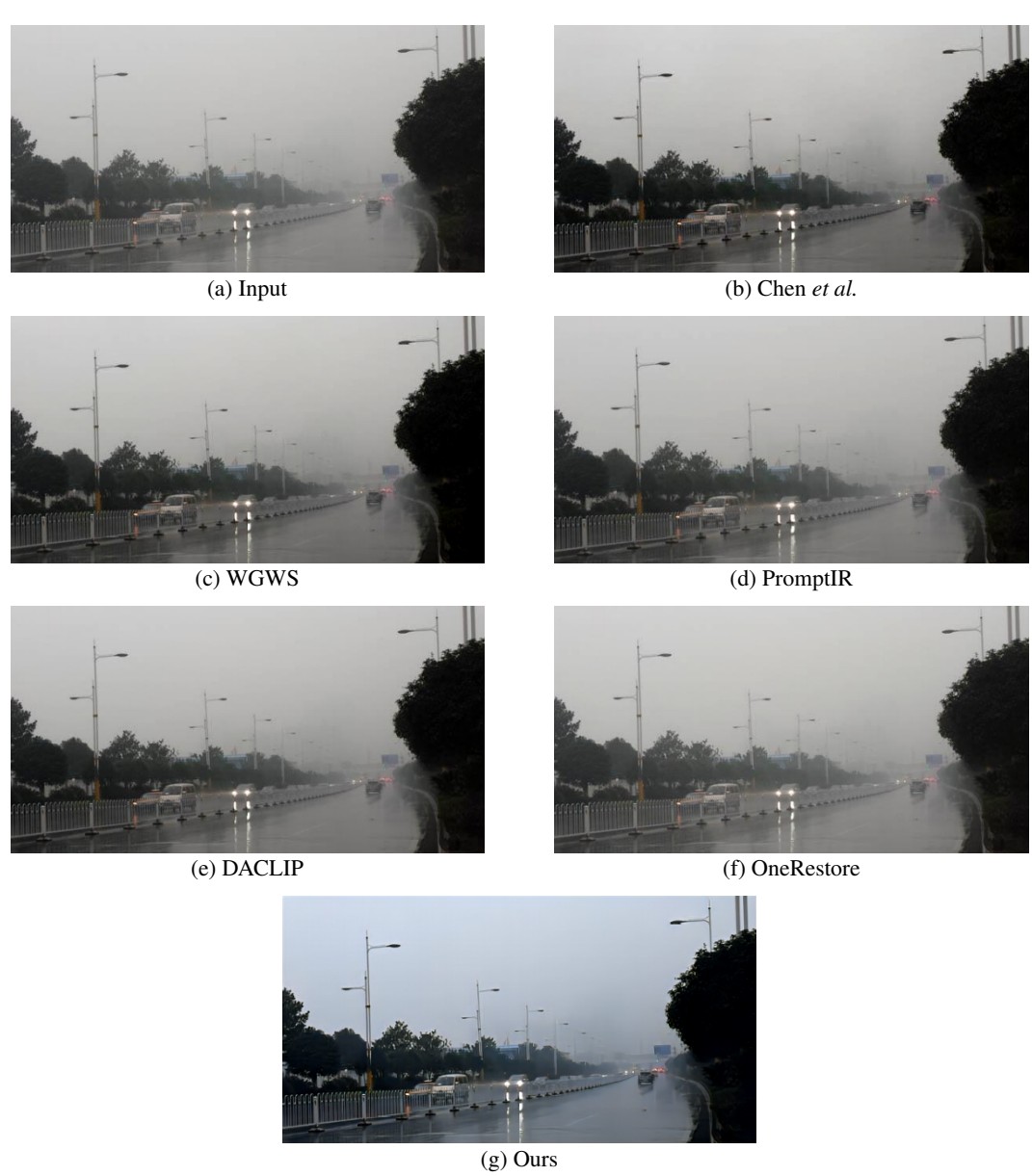

(a) Input

(b) Chen *et al.*

(c) WGWS

(d) PromptIR

(e) DACLIP

(f) OneRestore

(g) Ours

Figure 34: Visual comparison of real-world images under haze with [11, 18, 33, 37, 69].

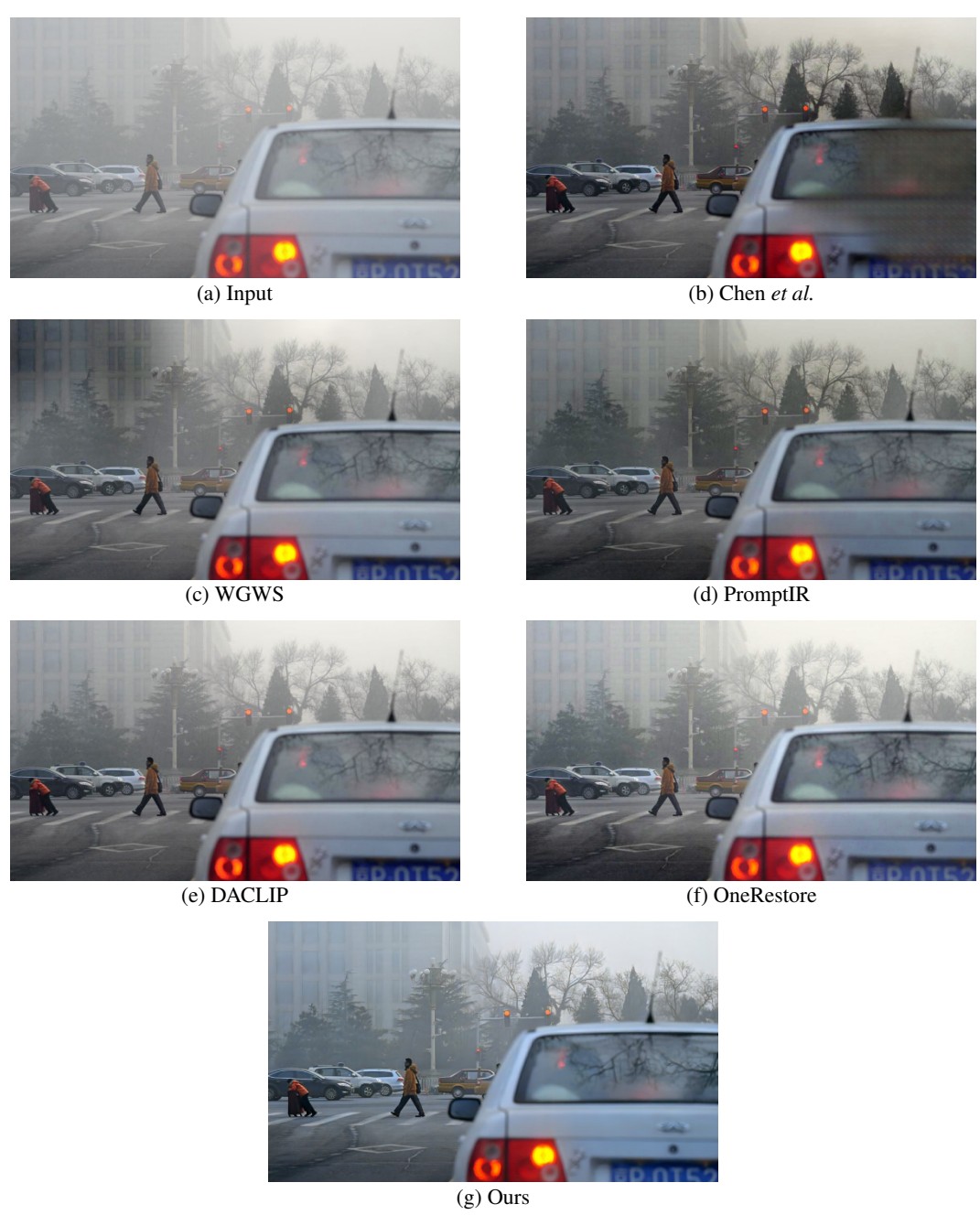

(a) Input

(b) Chen *et al.*

(c) WGWS

(d) PromptIR

(e) DACLIP

(f) OneRestore

(g) Ours

Figure 35: Visual comparison of real-world images under haze with [11, 18, 33, 37, 69].

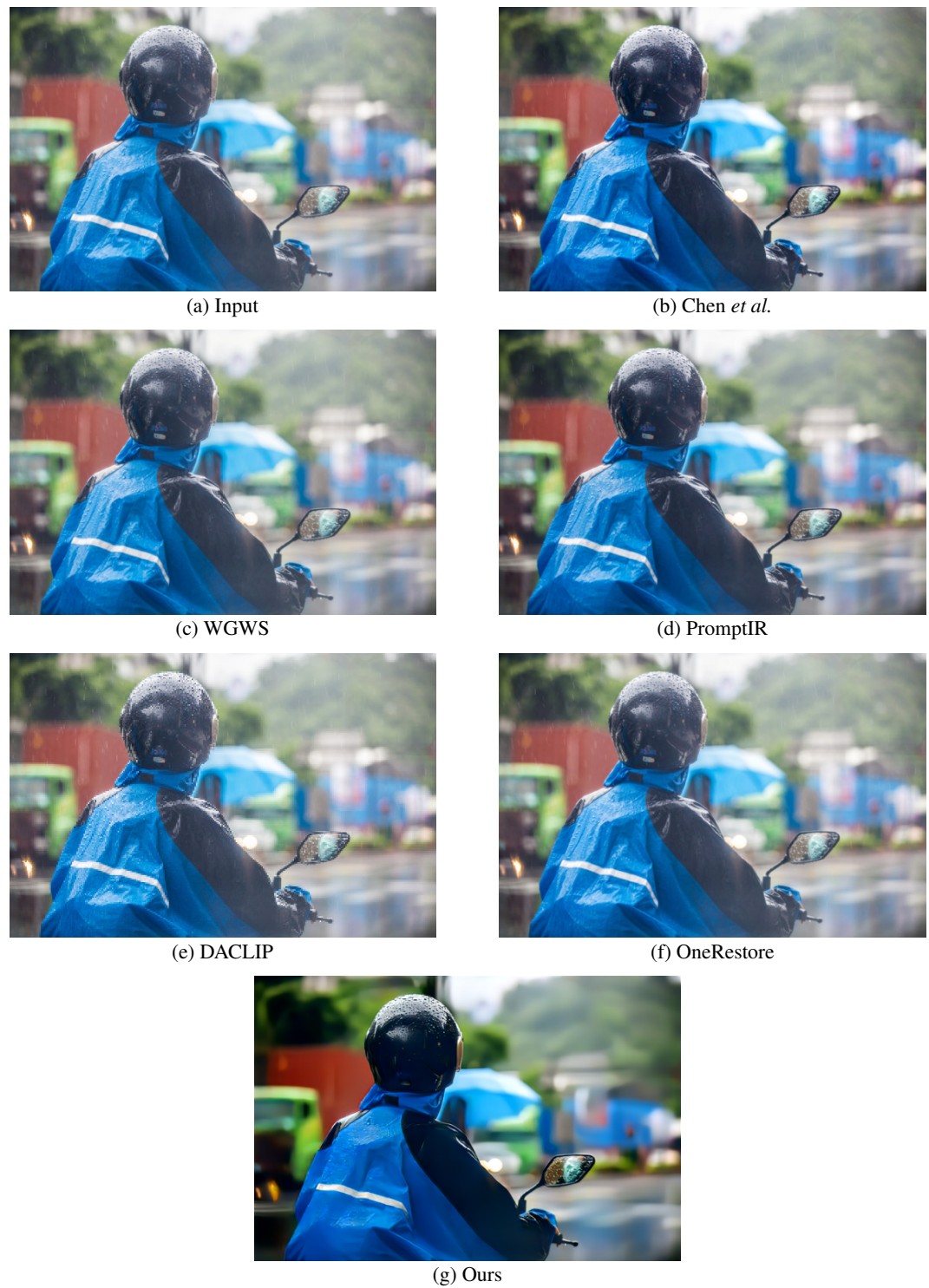

(a) Input

(b) Chen *et al.*

(c) WGWS

(d) PromptIR

(e) DACLIP

(f) OneRestore

(g) Ours

Figure 36: Visual comparison of real-world images under rain with [11, 18, 33, 37, 69].

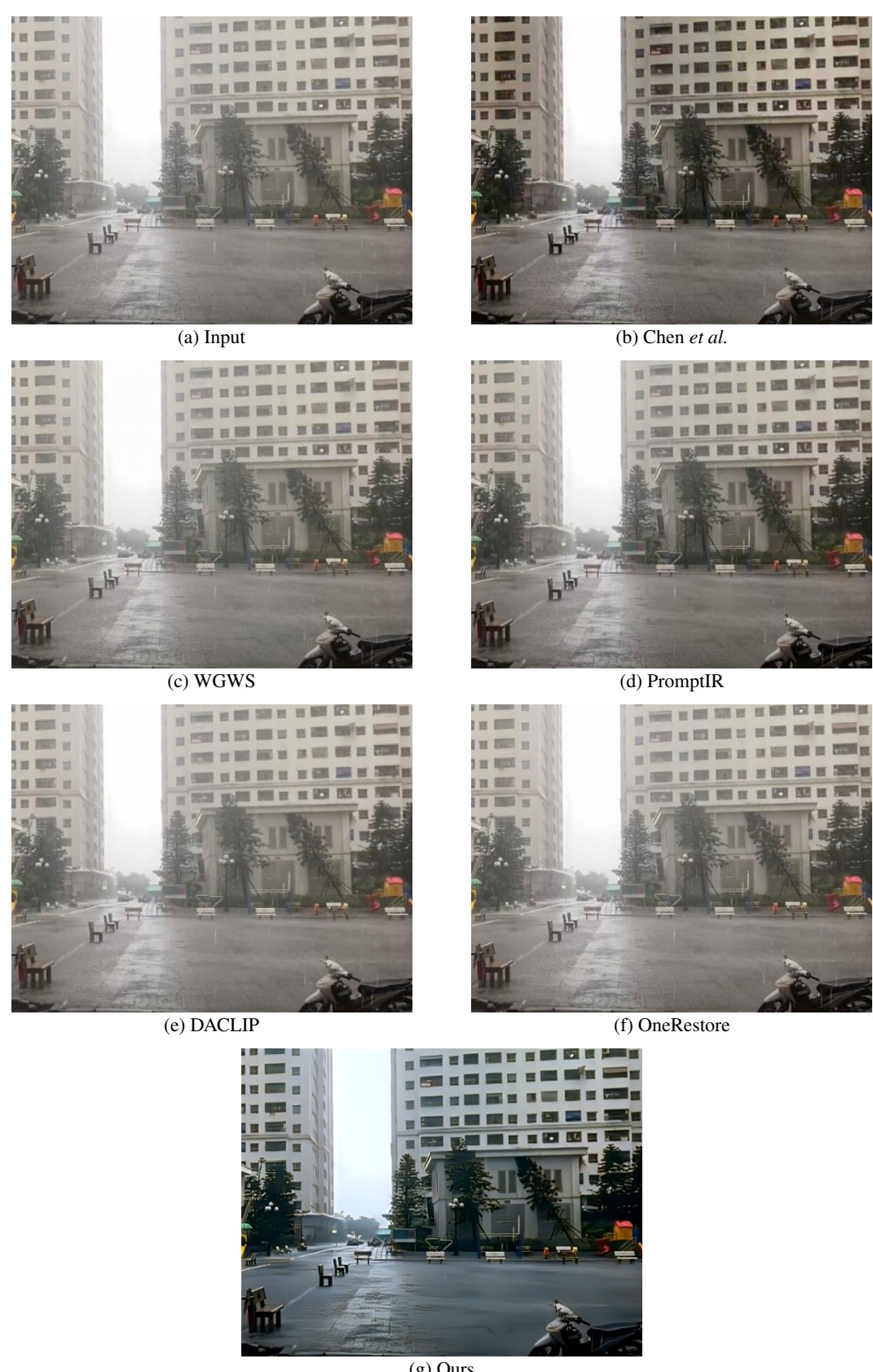

(a) Input

(b) Chen *et al.*

(c) WGWS

(d) PromptIR

(e) DACLIP

(f) OneRestore

(g) Ours

Figure 37: Visual comparison of real-world images under rain with [11, 18, 33, 37, 69].

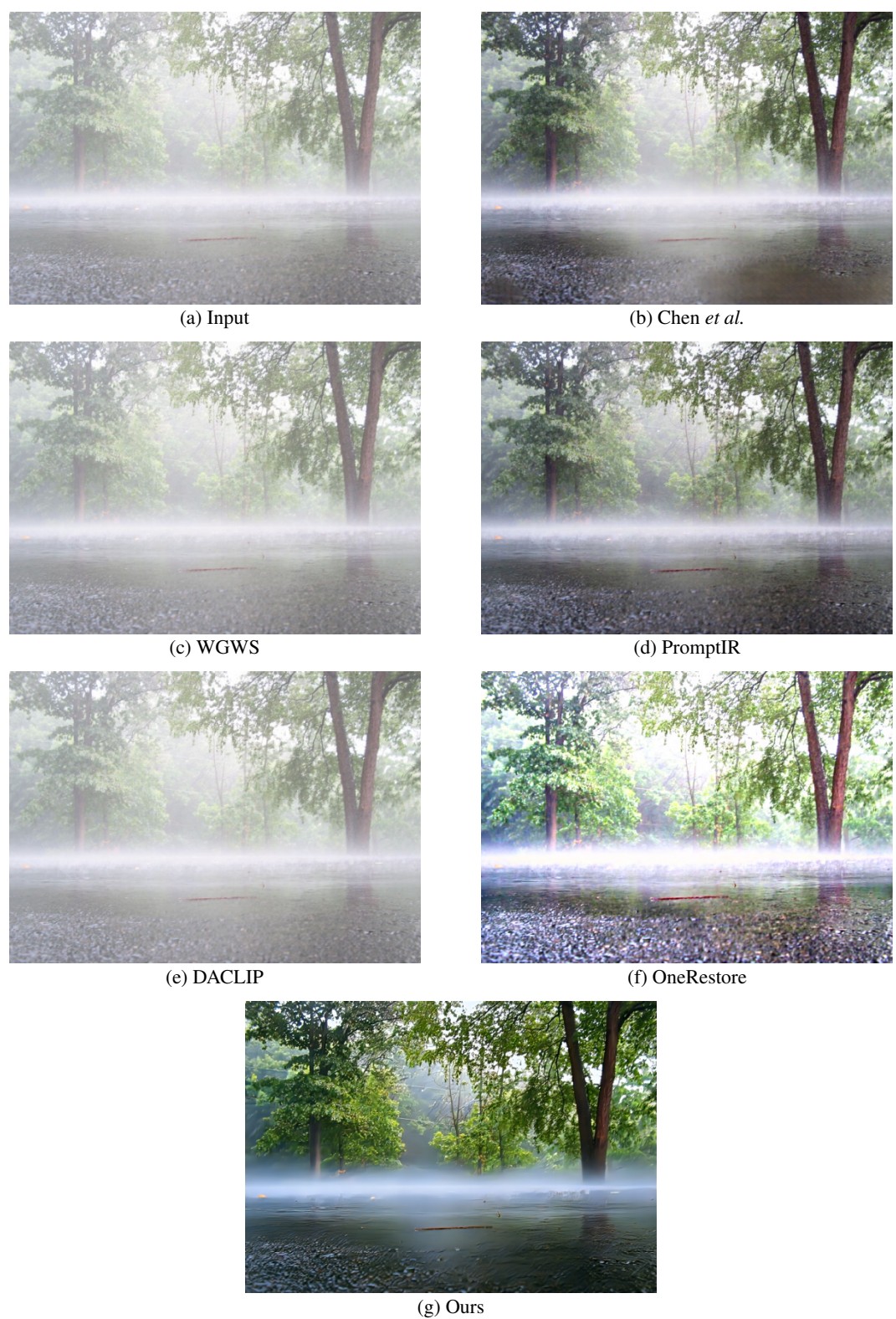

(a) Input

(b) Chen *et al.*

(c) WGWS

(d) PromptIR

(e) DACLIP

(f) OneRestore

(g) Ours

Figure 38: Visual comparison of real-world images under rain with [11, 18, 33, 37, 69].

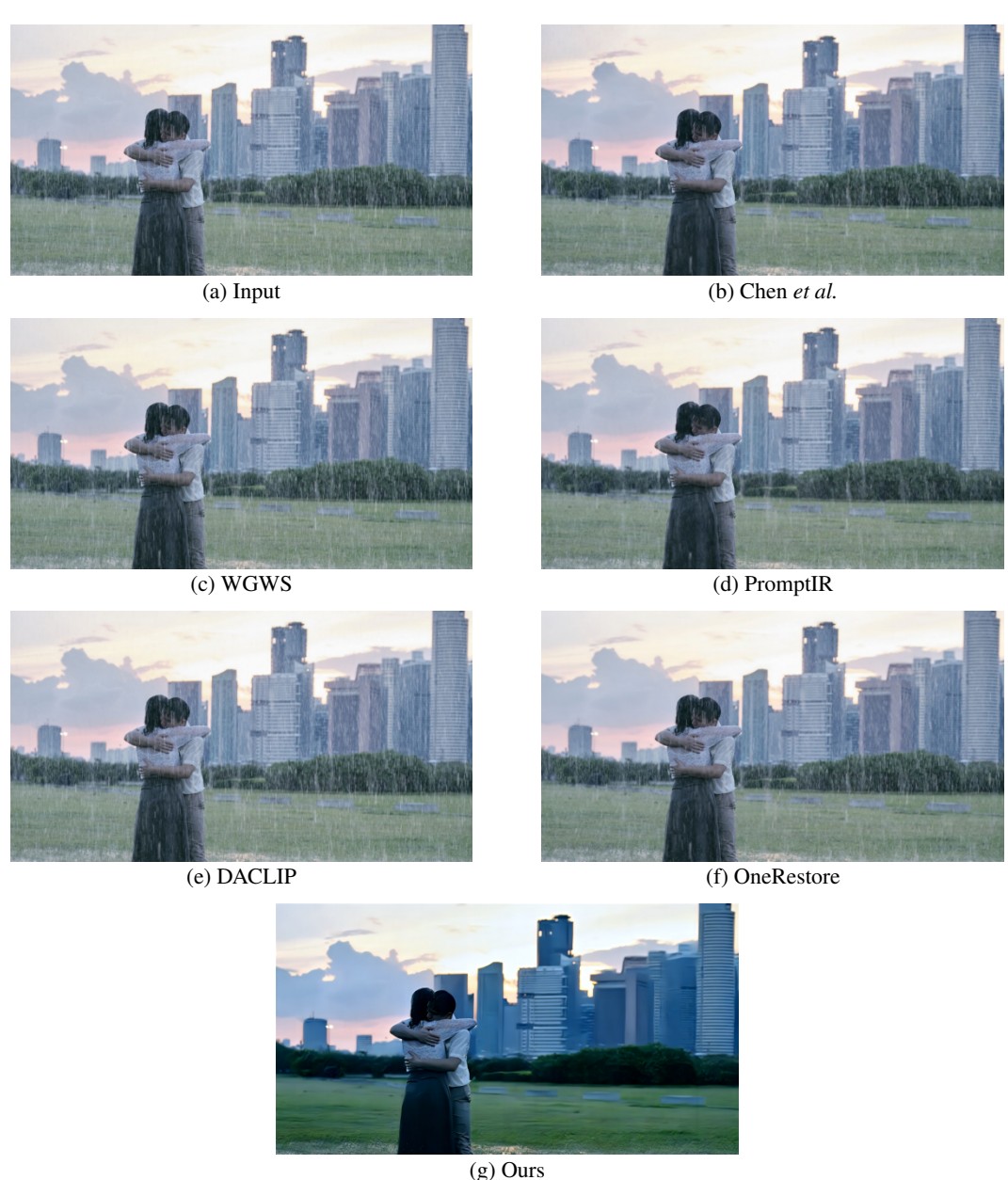

(a) Input

(b) Chen *et al.*

(c) WGWS

(d) PromptIR

(e) DACLIP

(f) OneRestore

(g) Ours

Figure 39: Visual comparison of real-world images under rain with [11, 18, 33, 37, 69].

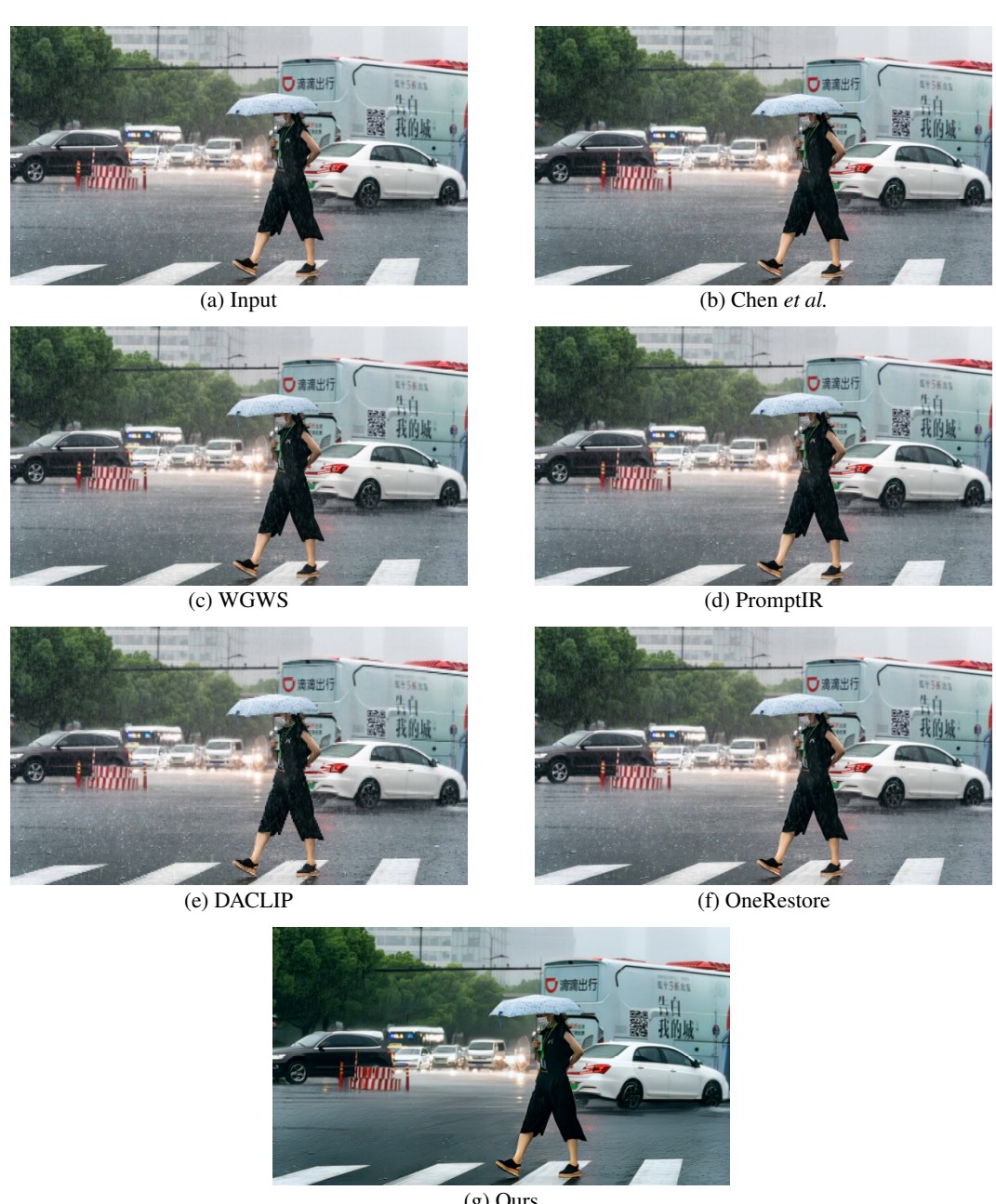

(a) Input

(b) Chen *et al.*

(c) WGWS

(d) PromptIR

(e) DACLIP

(f) OneRestore

(g) Ours

Figure 40: Visual comparison of real-world images under rain with [11, 18, 33, 37, 69].

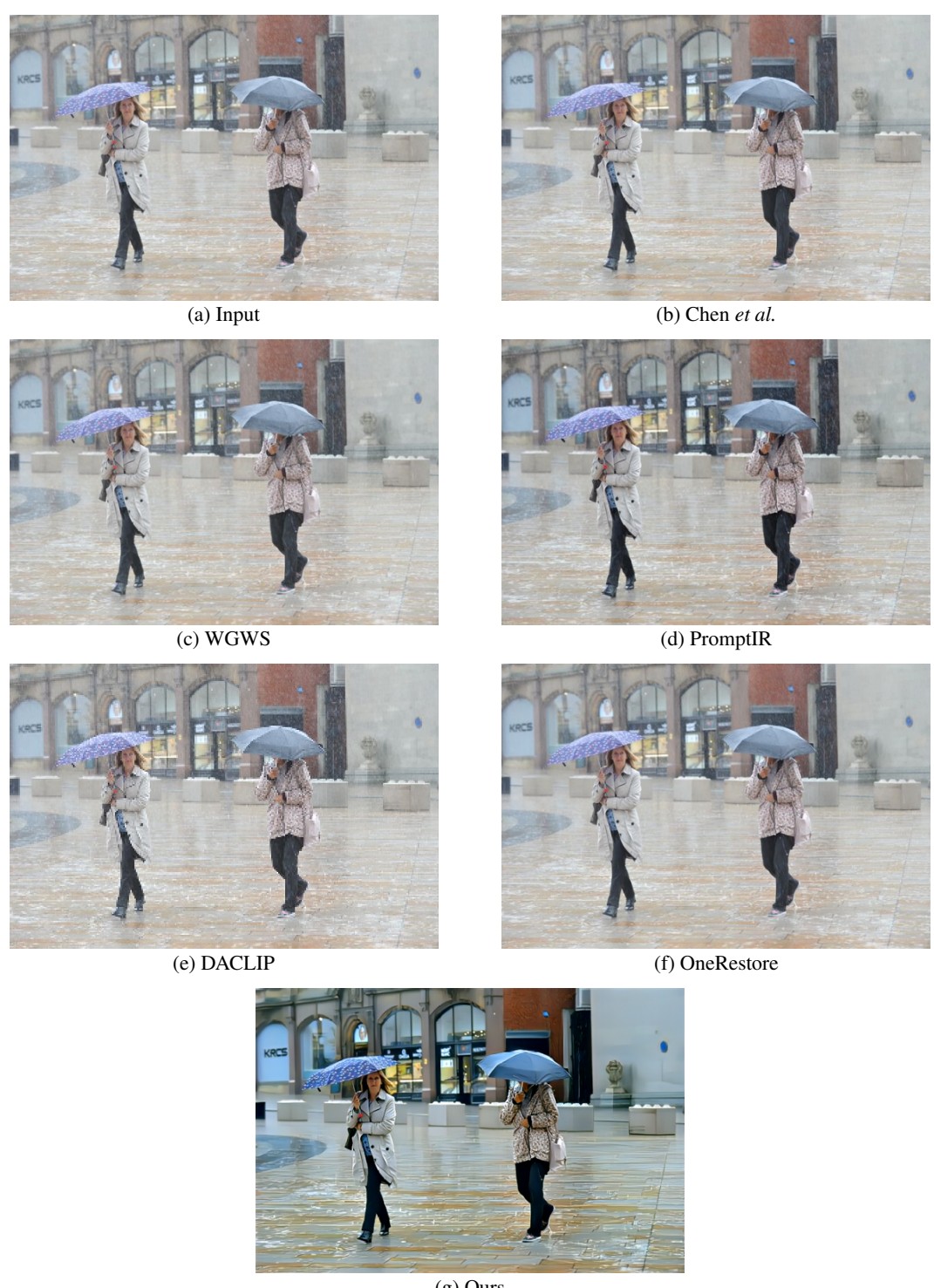

(a) Input         (b) Chen *et al.*

(c) WGWS         (d) PromptIR

(e) DACLIP         (f) OneRestore

(g) Ours

Figure 41: Visual comparison of real-world images under rain with [11, 18, 33, 37, 69].

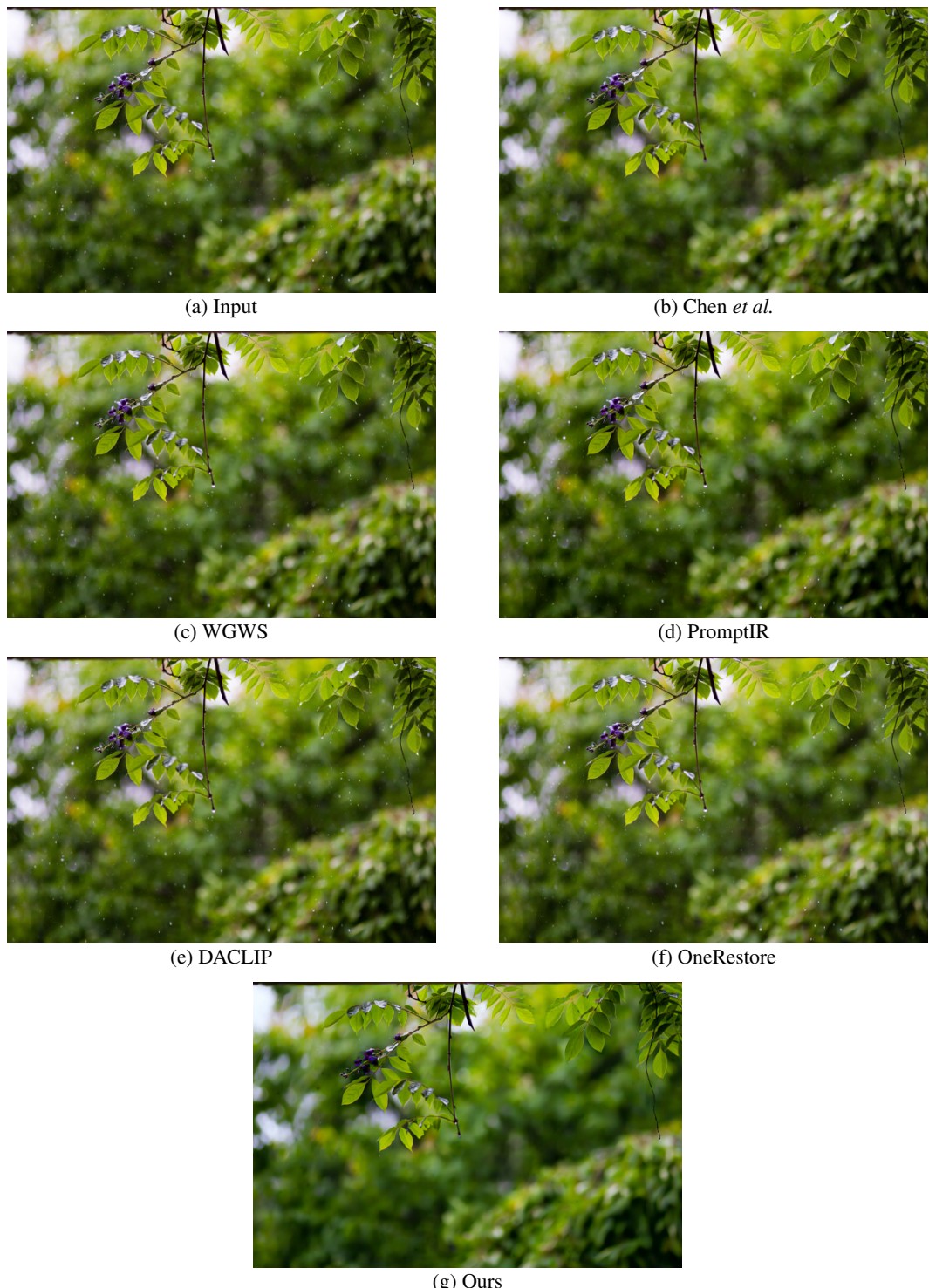

(a) Input

(b) Chen *et al.*

(c) WGWS

(d) PromptIR

(e) DACLIP

(f) OneRestore

(g) Ours

Figure 42: Visual comparison of real-world images under rain with [11, 18, 33, 37, 69].

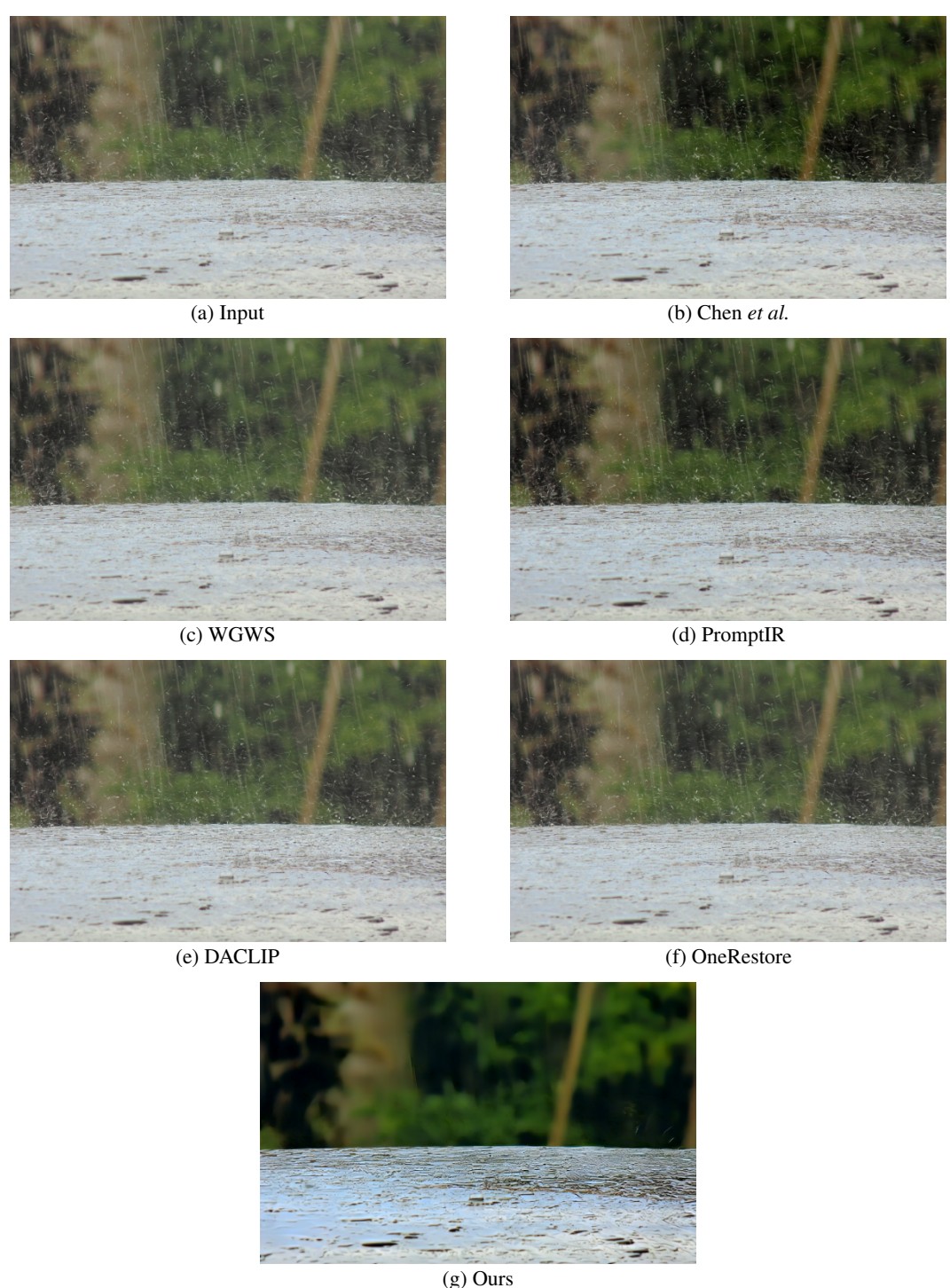

(a) Input

(b) Chen *et al.*

(c) WGWS

(d) PromptIR

(e) DACLIP

(f) OneRestore

(g) Ours

Figure 43: Visual comparison of real-world images under rain with [11, 18, 33, 37, 69].

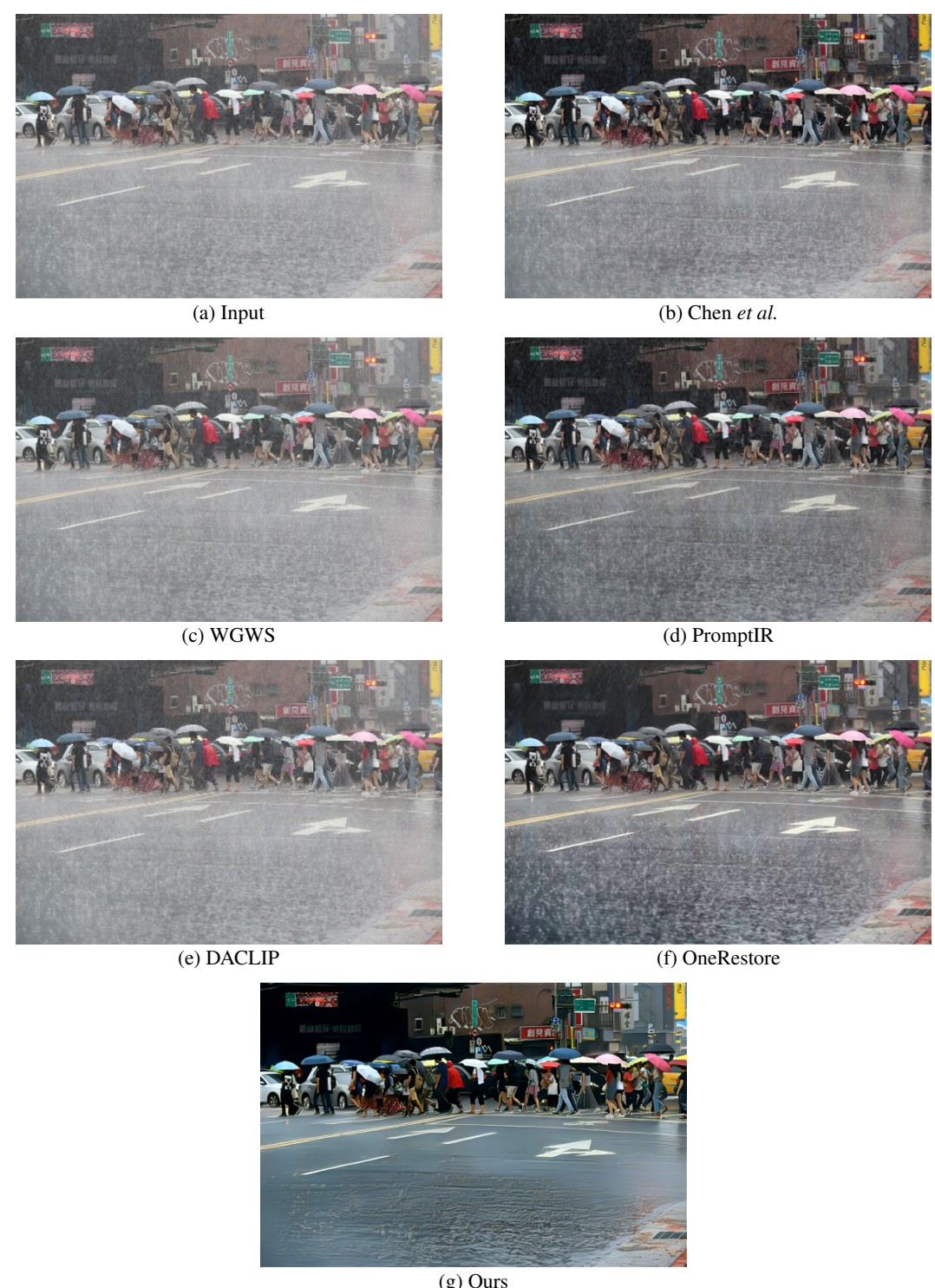

(a) Input

(b) Chen *et al.*

(c) WGWS

(d) PromptIR

(e) DACLIP

(f) OneRestore

(g) Ours

Figure 44: Visual comparison of real-world images under rain with [11, 18, 33, 37, 69].

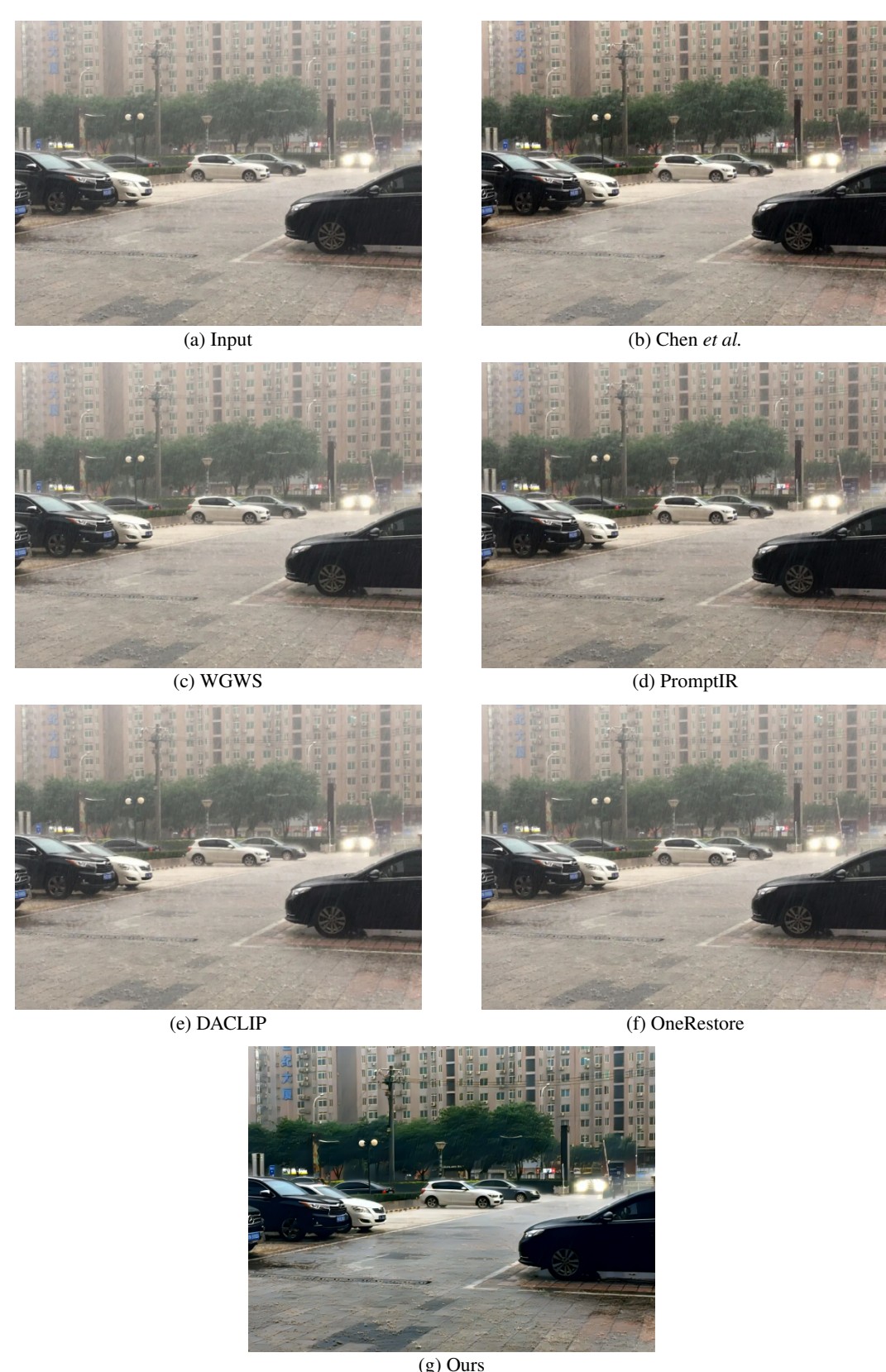

(a) Input

(b) Chen *et al.*

(c) WGWS

(d) PromptIR

(e) DACLIP

(f) OneRestore

(g) Ours

Figure 45: Visual comparison of real-world images under rain with [11, 18, 33, 37, 69].

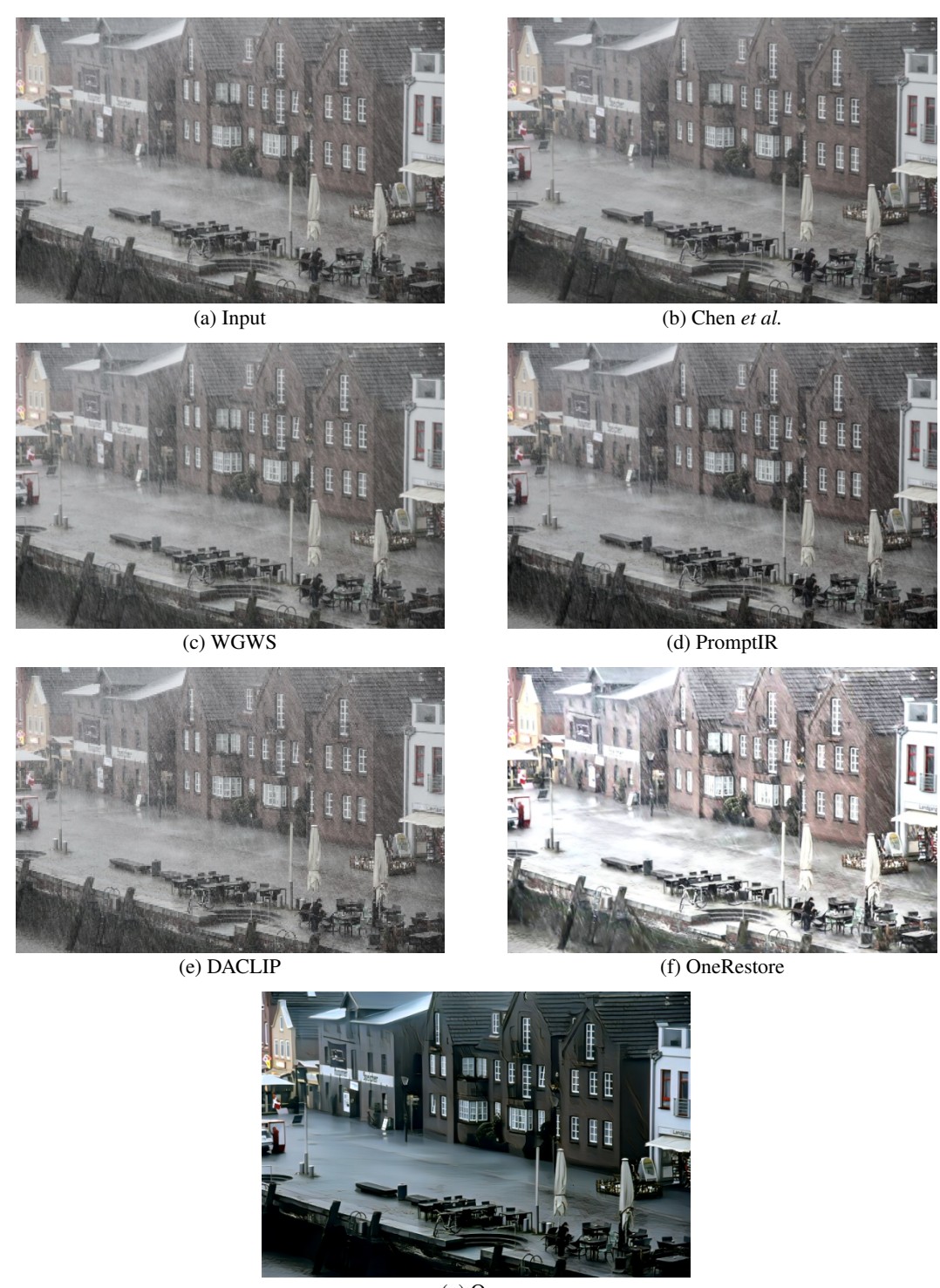

(a) Input

(b) Chen *et al.*

(c) WGWS

(d) PromptIR

(e) DACLIP

(f) OneRestore

(g) Ours

Figure 46: Visual comparison of real-world images under rain with [11, 18, 33, 37, 69].

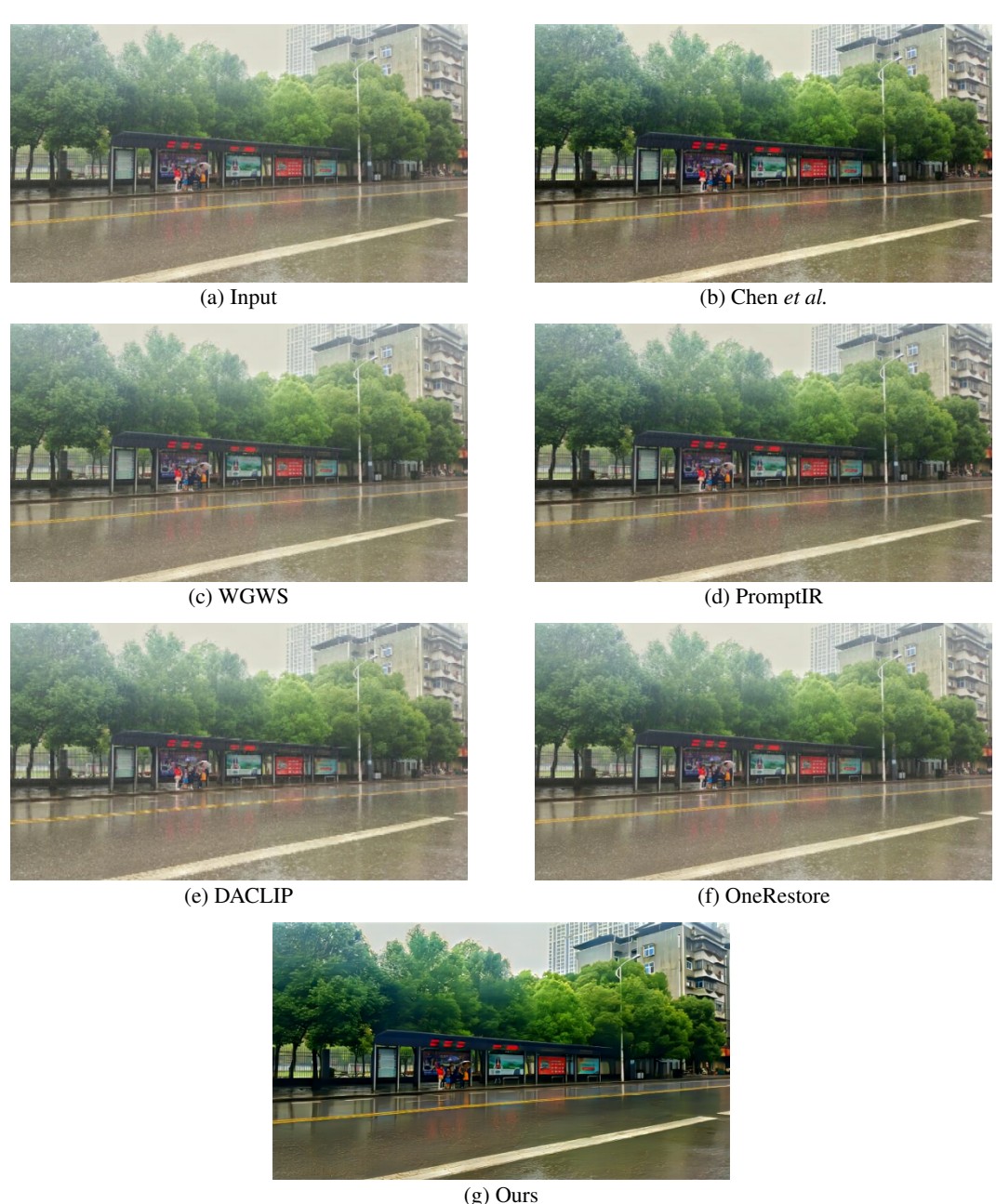

(a) Input

(b) Chen *et al.*

(c) WGWS

(d) PromptIR

(e) DACLIP

(f) OneRestore

(g) Ours

Figure 47: Visual comparison of real-world images under rain with [11, 18, 33, 37, 69].

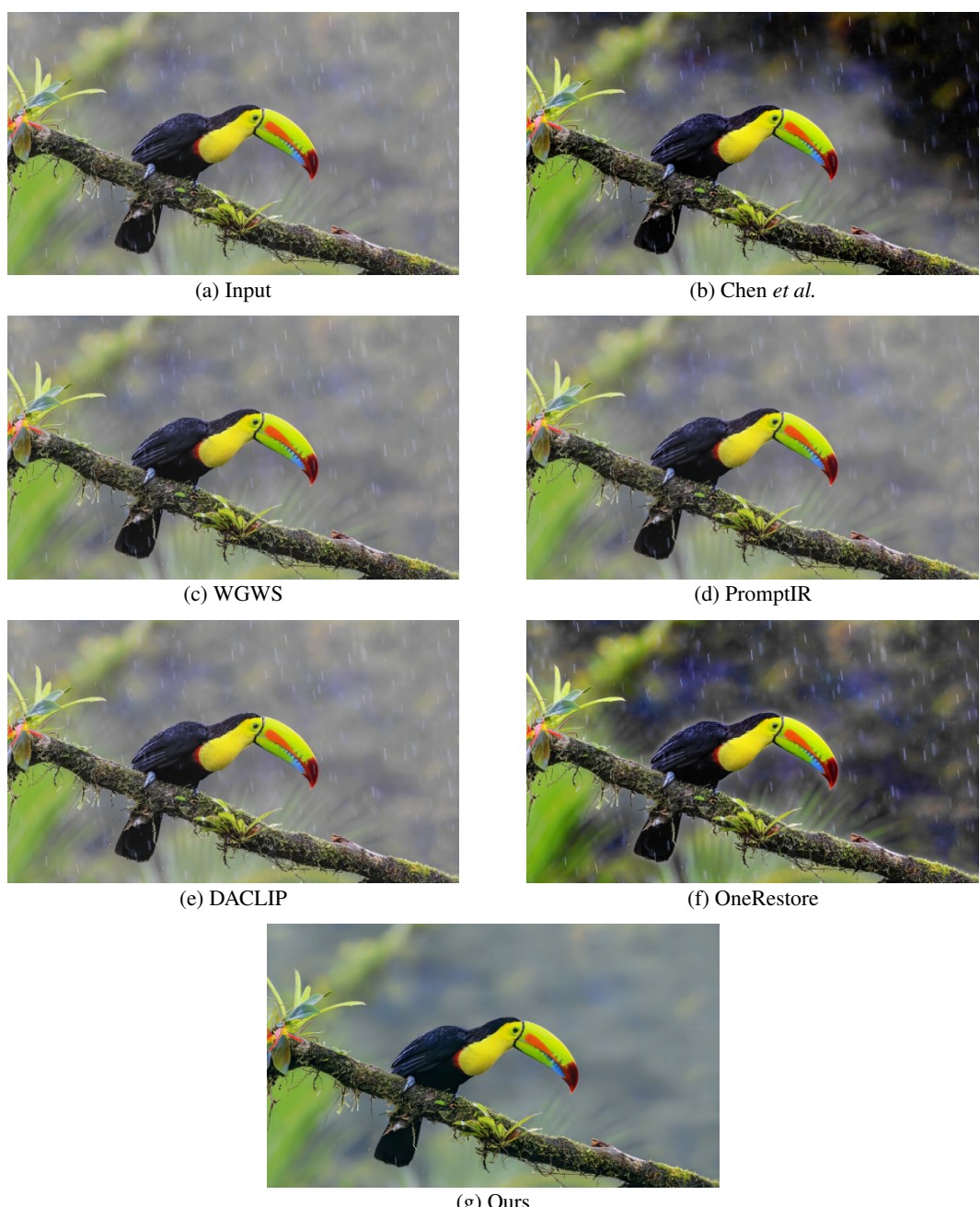

(a) Input

(b) Chen *et al.*

(c) WGWS

(d) PromptIR

(e) DACLIP

(f) OneRestore

(g) Ours

Figure 48: Visual comparison of real-world images under rain with [11, 18, 33, 37, 69].

# C   Additional Results Produced by Our Method

We provide more qualitative results under snow, haze, and rain conditions.

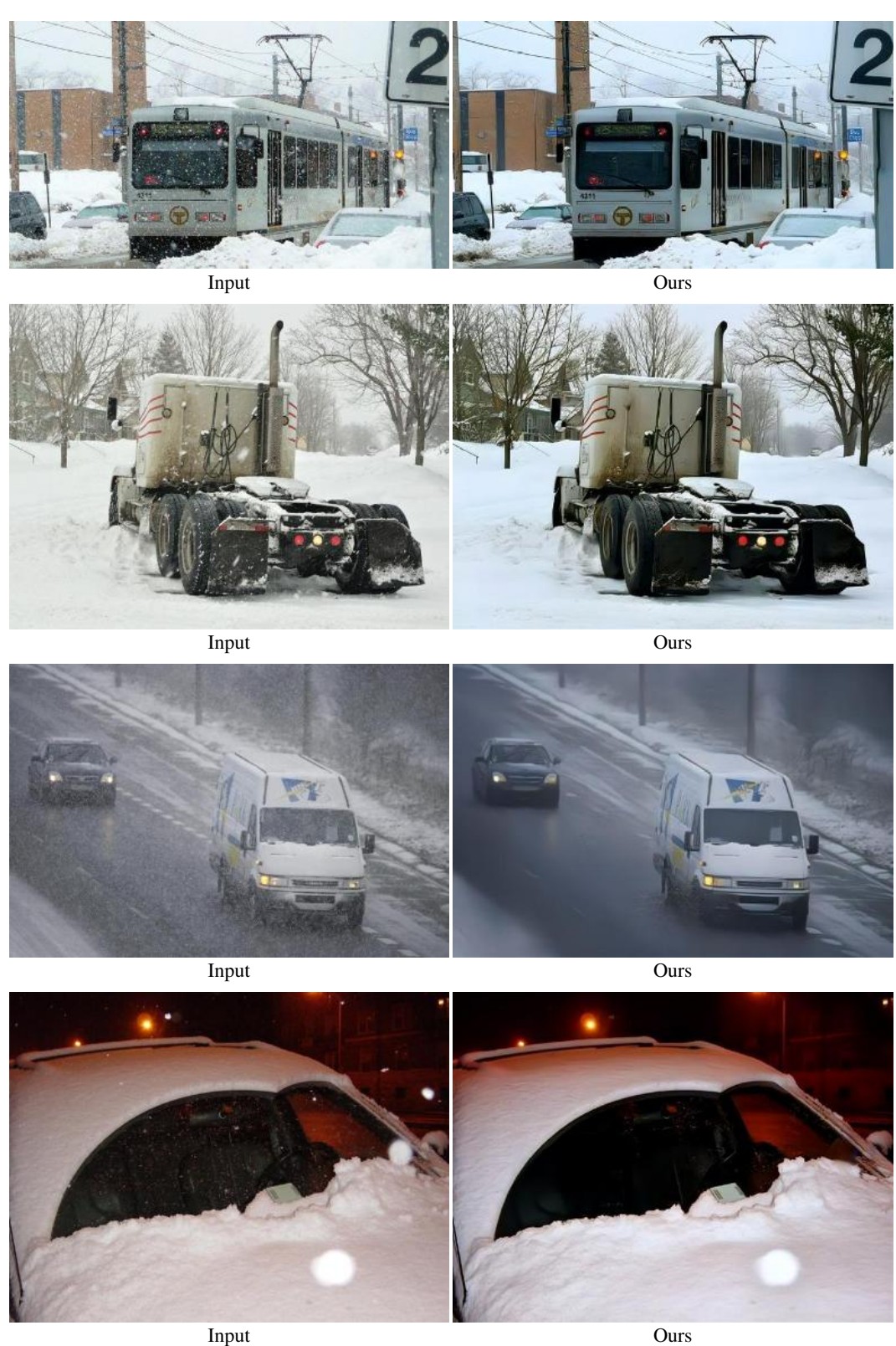

Input          Ours

Figure 49: Additional results produced by our method (snow, set 1).

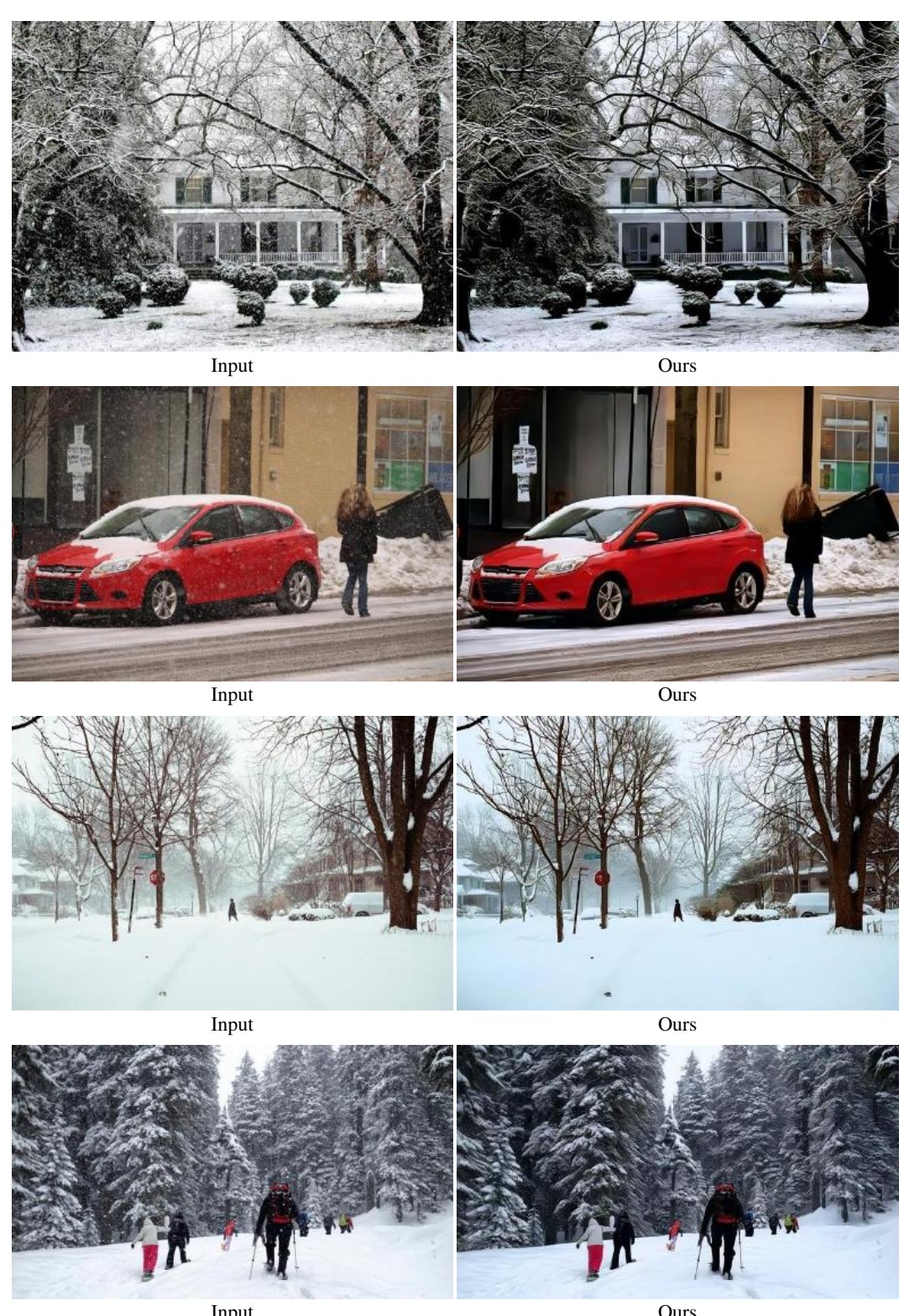

Input            Ours

Input            Ours

Input            Ours

Input            Ours

Figure 50: Additional results produced by our method (snow, set 2).

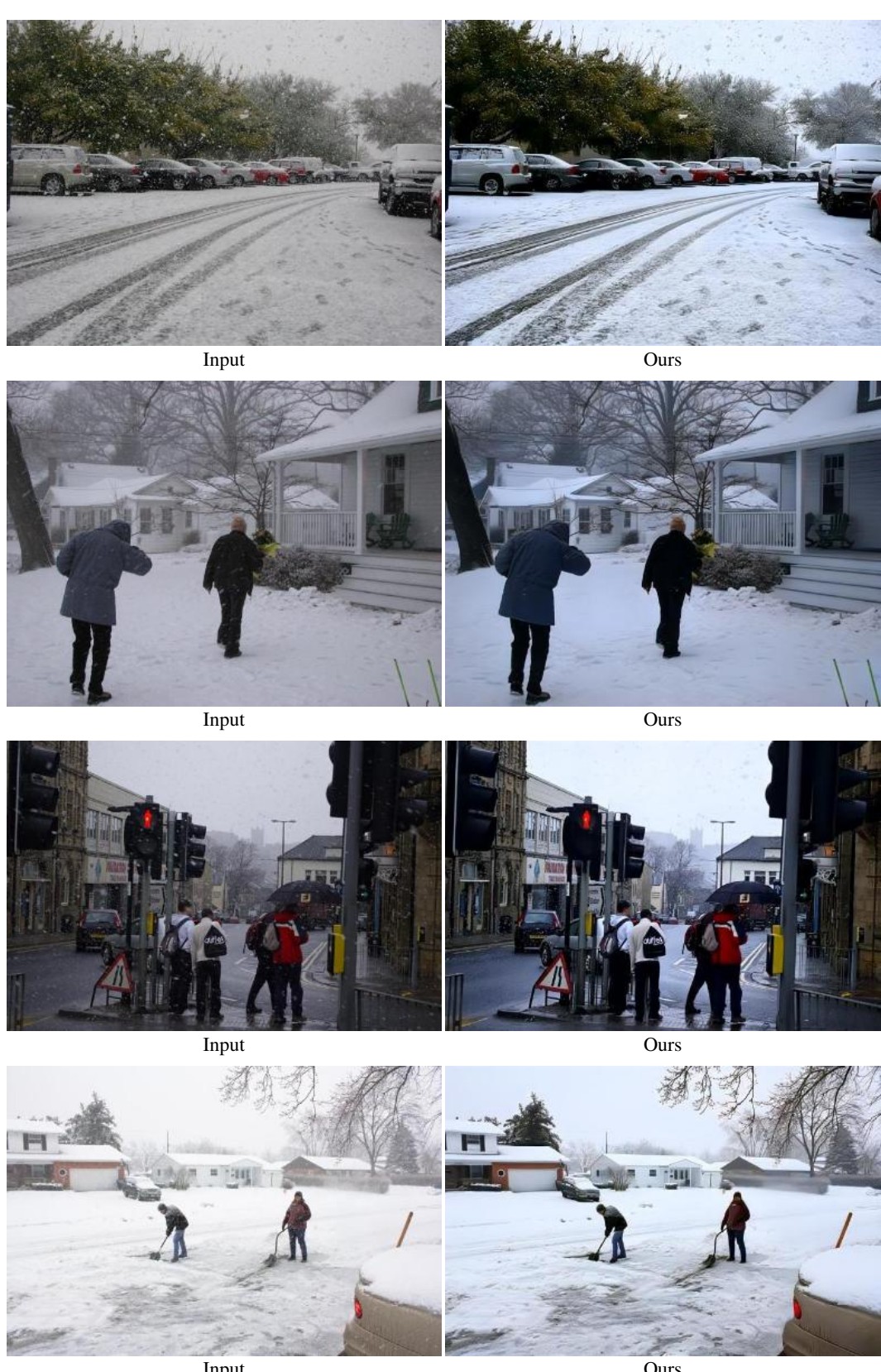

Input                              Ours

Input                              Ours

Input                              Ours

Input                              Ours

Figure 51: Additional results produced by our method (snow, set 3).

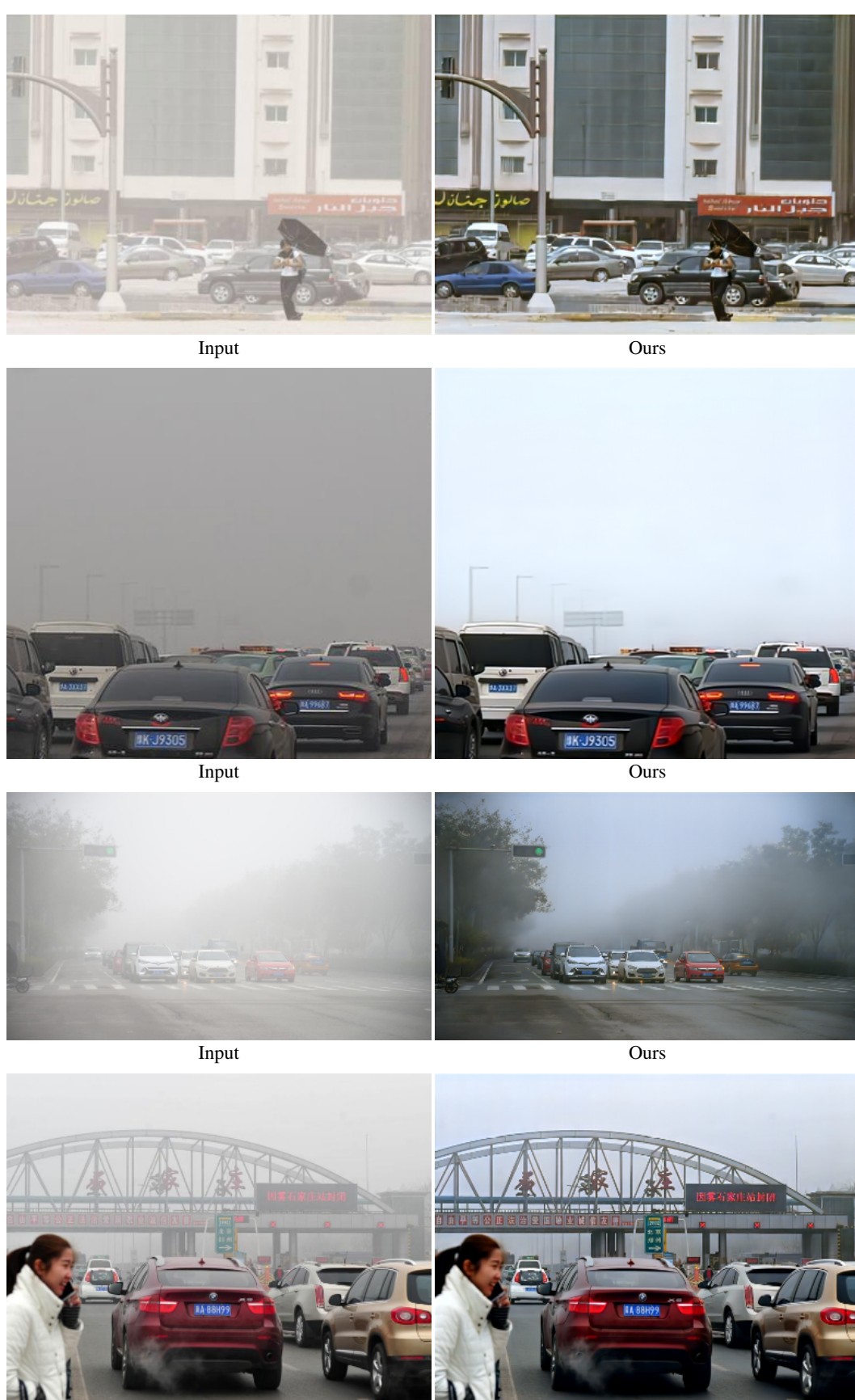

70

Figure 52: Additional results produced by our method (haze, set 1).

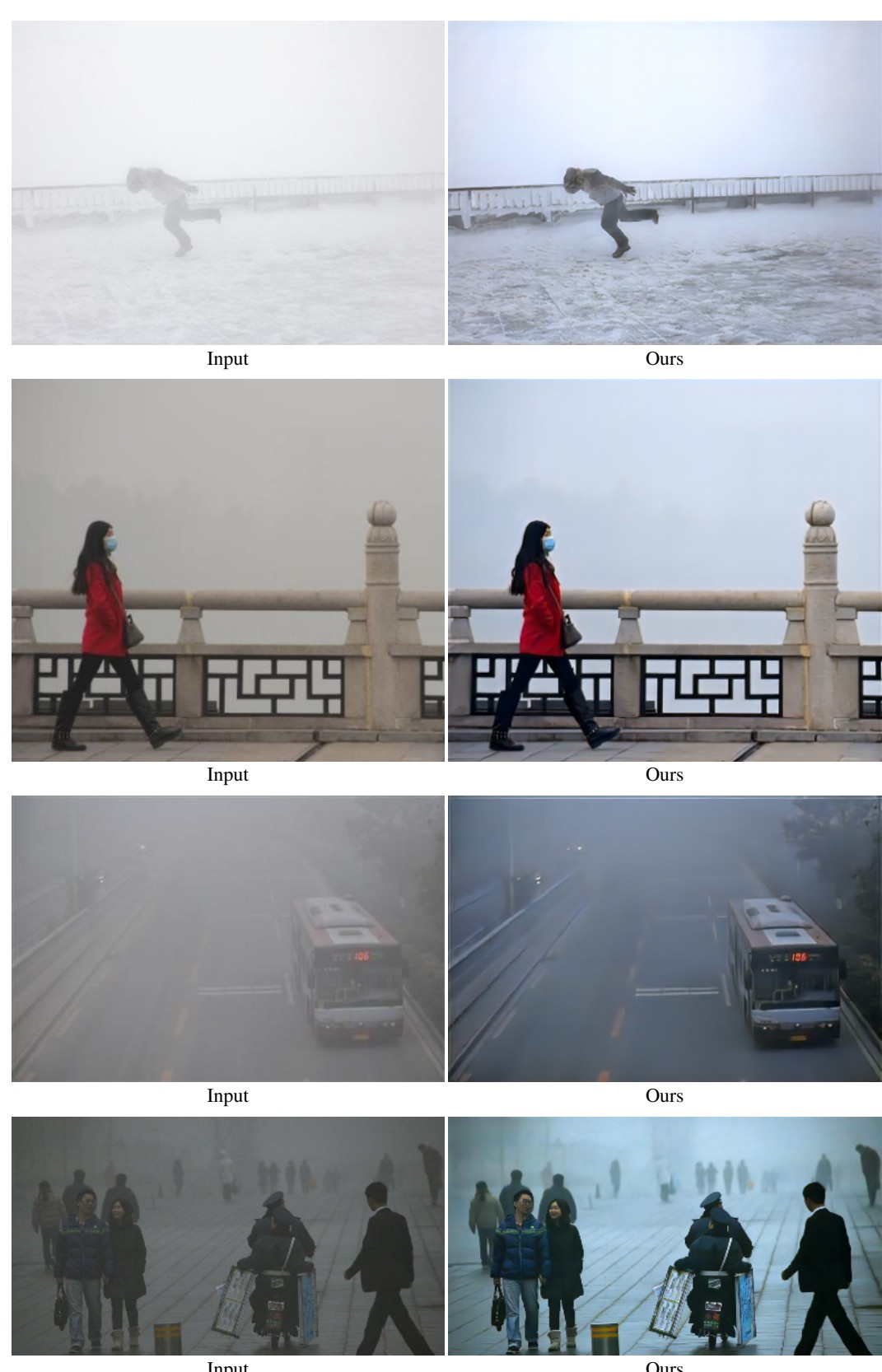

Input        Ours

Input        Ours

Input        Ours

Input        Ours

Figure 53: Additional results produced by our method (haze, set 2).

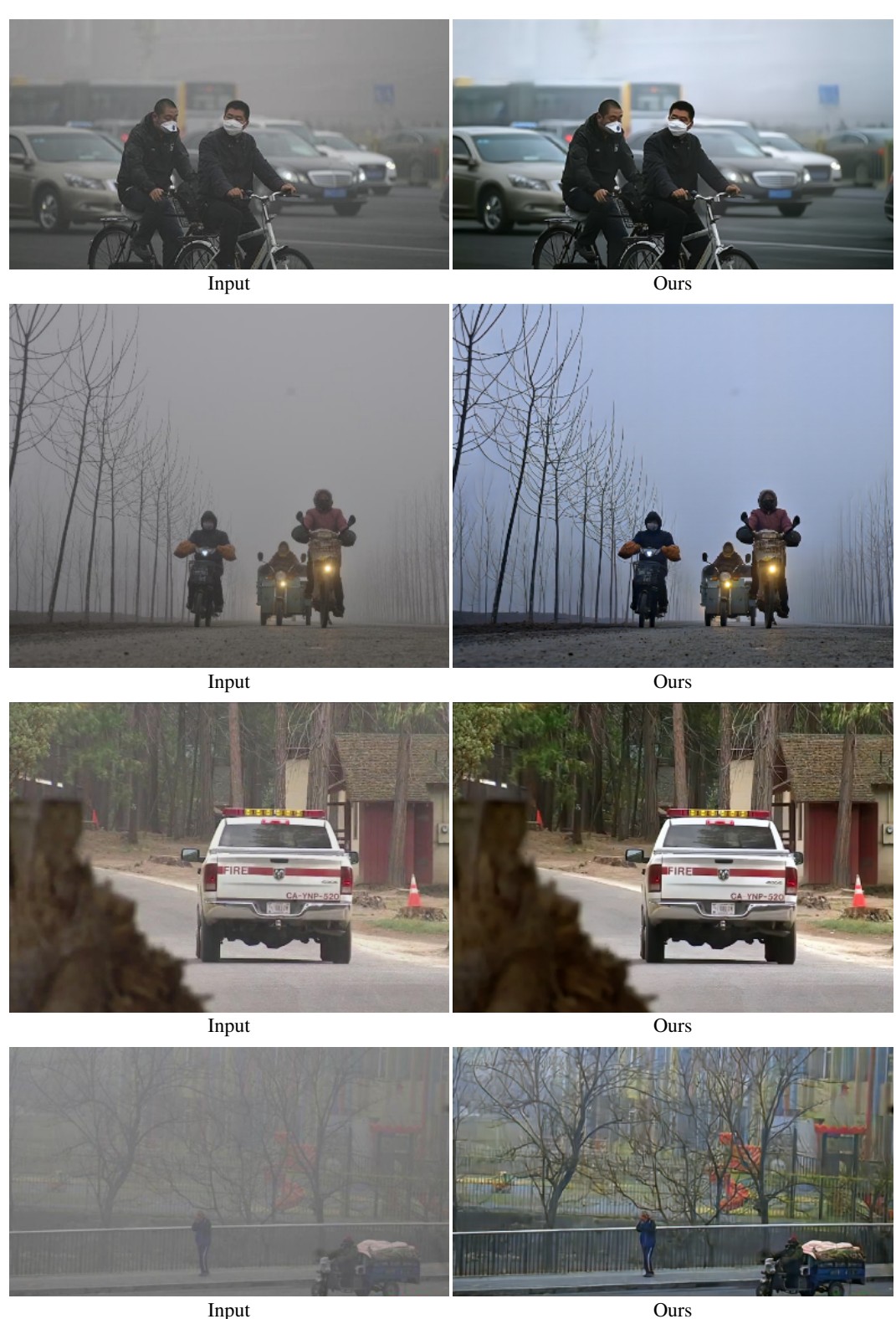

Input Ours

Input Ours

Input Ours

Input Ours

Figure 54: Additional results produced by our method (haze, set 3).

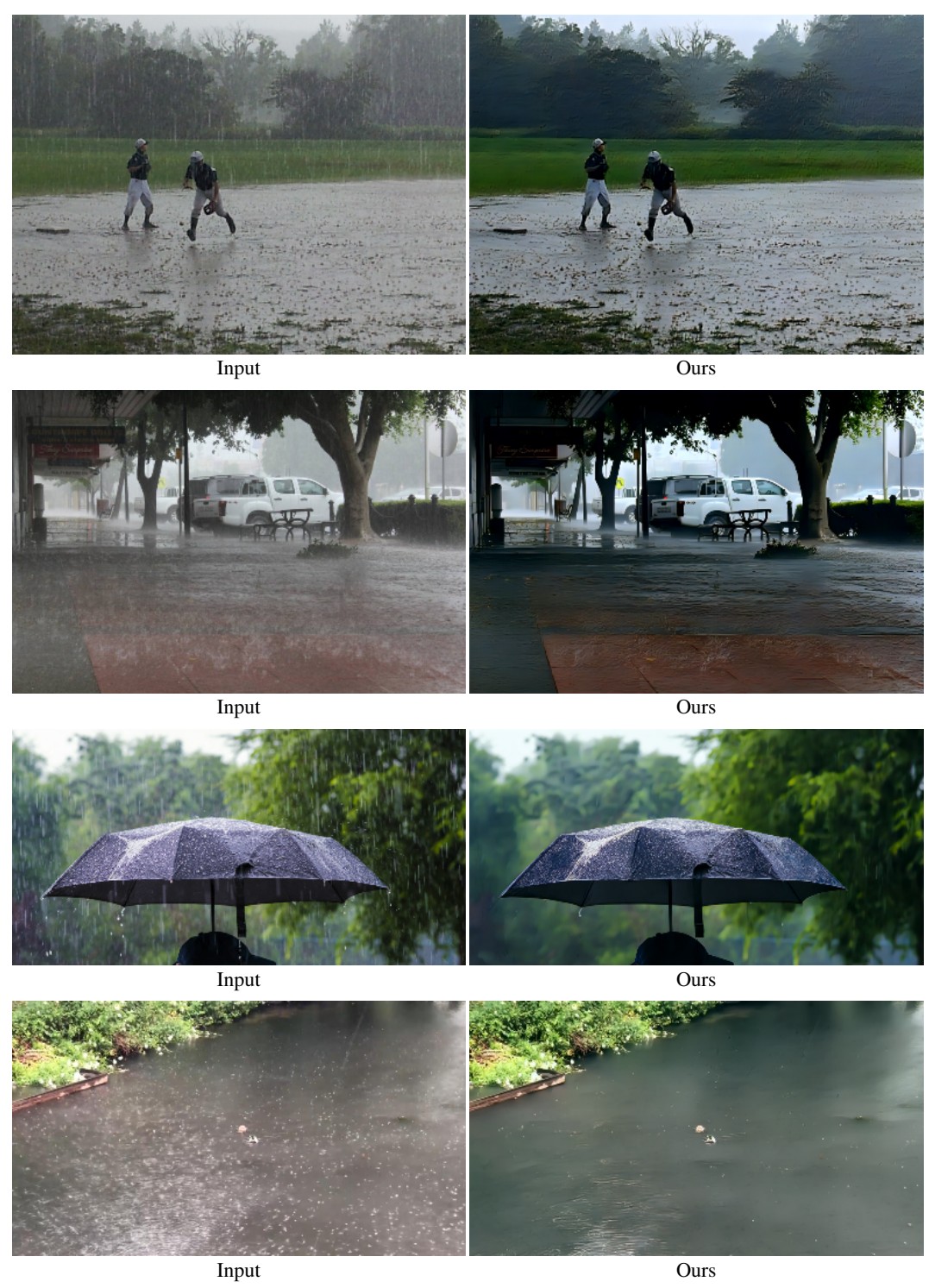

Input           Ours

Input           Ours

Input           Ours

Input           Ours

Figure 55: Additional results produced by our method (rain, set 1).

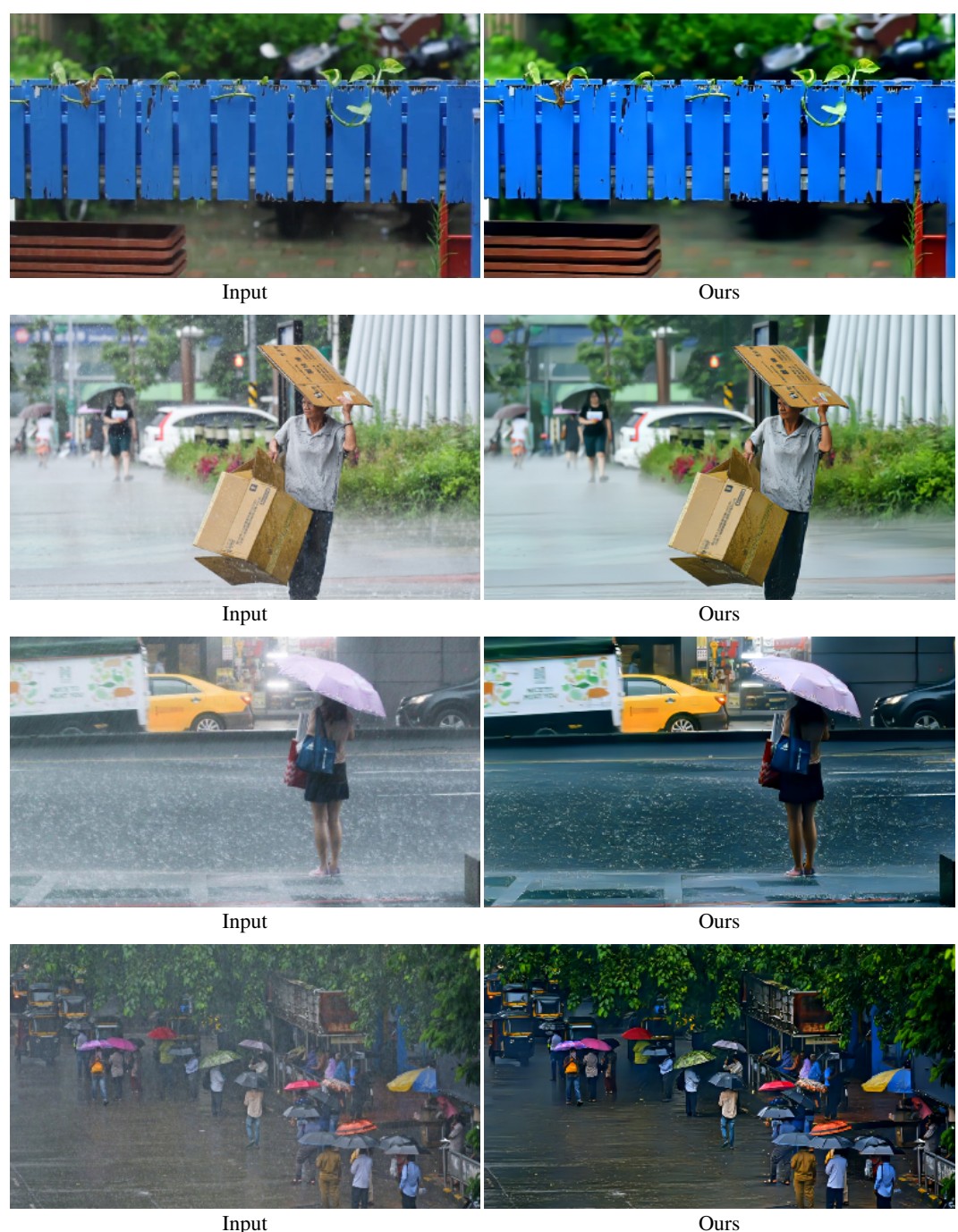

Input           Ours

Figure 56: Additional results produced by our method (rain, set 2).

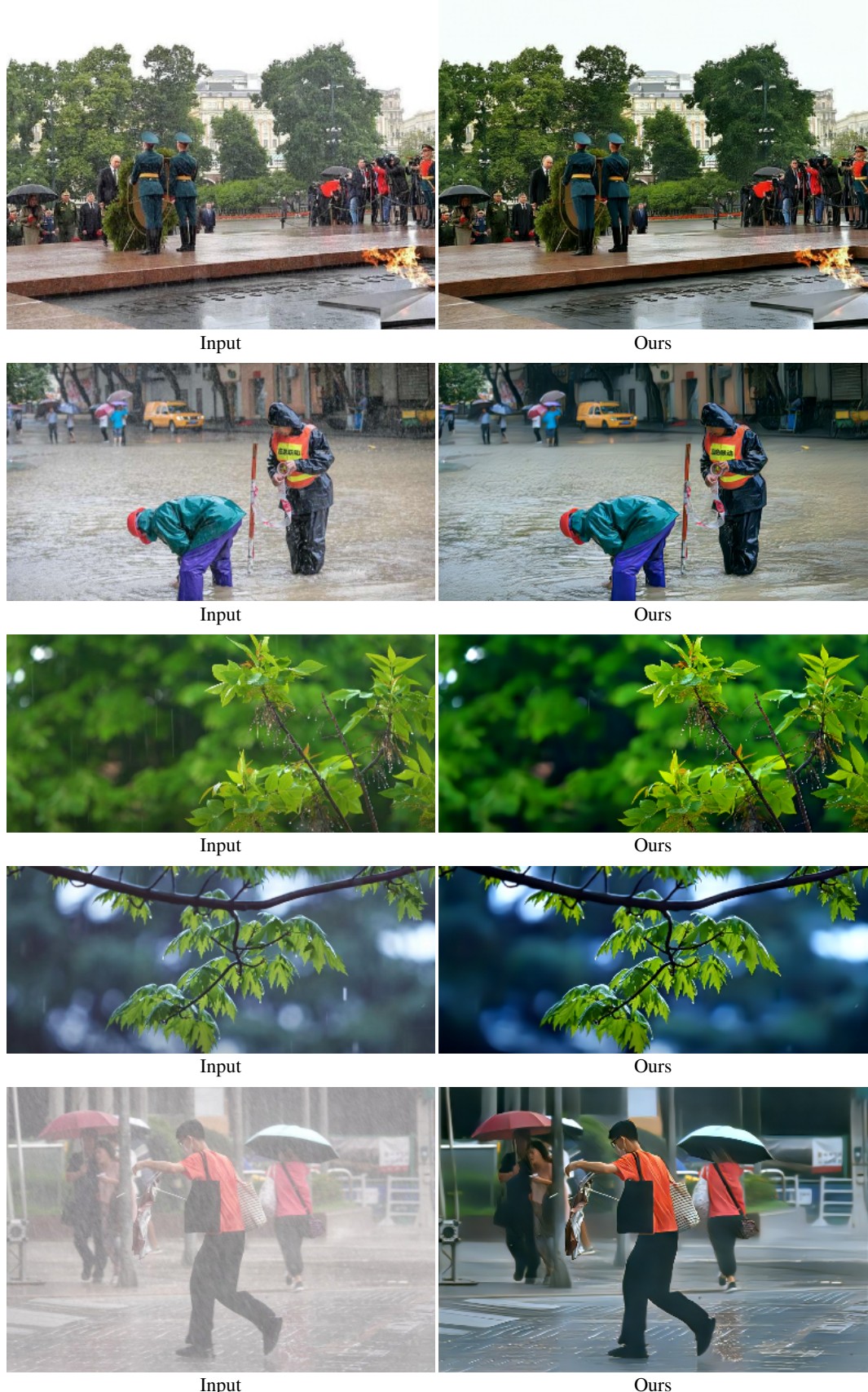

Figure 57: Additional results produced by our method (rain, set 3).

