# OpenReview forum: "Real-World Adverse Weather Image Restoration via Dual-Level Reinforcement Learning with High-Quality Cold Start"
_NeurIPS.cc/2025/Conference — NeurIPS 2025 poster_

### Official Review · Reviewer_jNge · 2025-06-01

**Clarity:** 3
**Significance:** 3
**Originality:** 2
**Rating:** 4
**Confidence:** 5

**Summary:**

The paper introduces HFLS-Weather, a high-fidelity synthetic dataset with one million clean/degraded pairs that couples depth-aware haze, rain, and snow simulation to correct shortcomings in prior rain- or haze-only benchmarks.  Using models pre-trained on this dataset as “cold starts,” the authors build a dual-level reinforcement learning framework: (i) local adaptation via Perturbation-Driven Image Quality Optimization (PIQO), which injects Gaussian parameter perturbations and optimizes a composite no-reference reward without paired supervision ; and (ii) a global meta-controller that selects and orders weather-specific agents according to CLIP-based scene analysis and historical success rates.

**Questions:**

1. Could you clarify how you selected the relative weights assigned to different IQA models in the PIQO reward, and whether training stability changed when those weights were perturbed?

2. The reward signal is a linear combination of no-reference metrics; have you considered a learned reward network trained on small amounts of rated data, and how might that compare with the hand-designed formula?

3. Can you provide more comparison results with the updated methods?

I am very willing to improve my score if the author has well addressed my concerns.

**Ethical Concerns:**

["NO or VERY MINOR ethics concerns only"]

**Final Justification:**

I have read the authors’ response as well as the concerns raised by the other reviewers. The authors’ reply has addressed some of my concerns. I have determined my final score based on this.

**Limitations:**

The author points out that their method introduces additional overhead in terms of computation cost, but the analysis of limitations and failure cases is not comprehensive.

**Paper Formatting Concerns:**

No.

**Quality:**

3

**Strengths And Weaknesses:**

S1. HFLS-Weather supplies depth-consistent multi-weather degradation at a scale far larger than previous datasets, enhancing generalization to real conditions.

S2. PIQO successfully extends GRPO-style reinforcement learning to image restoration, eliminating the need for ground-truth clean images while still providing stable training through KL-regularised updates.

S3. The two-tier design adapts both model parameters and model selection, giving the system flexibility to handle mixed or sequential degradations that defeat single-network baselines.

S4. Across three weather types the method shows consistent, sizeable gains and superior GPT-4o judgements, with detailed ablations clarifying where the improvements originate.

W1. Both training and inference raise resource demands; the authors acknowledge additional latency but do not quantify it.

W2. The composite reward relies on hand-tuned weights for no-reference metrics whose correlation with human perception can vary, yet the impact of these weights is not explored.

W3. The scheduler restricts participation to three agents per image, and there is no analysis of how performance or runtime would change with more degradation types or a larger agent library.

W4. Lack of analysis and verification of extreme degradation scenarios.

W5. The comparative experiment is insufficient. It only compares with some methods in the past, and lacks comparison with the existing latest agent-based methods and the latest All-in-one image restoration methods.

---

> ### Author Rebuttal · Authors · 2025-07-31
>
> We sincerely thank the reviewer for the valuable comments and constructive suggestions, which have played an important role in helping us further improve our work. Your review has been extremely helpful in refining our thinking and enhancing the quality of the paper. We also sincerely appreciate your recognition of the novelty and effectiveness of our work. Below, we provide detailed responses to your comments, with related issues grouped together for clarity. All corresponding discussions and additional experimental results will be included in the final camera-ready version.
>
> ---
> **[W1] Resource Overhead:**
>
> Thank the reviewer for the question. We report that cold-start training of the lightweight single-agent on eight parallel RTX 4090 GPUs required ~72 hours with a 23 GB peak memory footprint. The complete PIQO + Agent framework—trained thereafter on a single RTX 4090—took ~48 hours with a 22 GB peak. During inference, the full framework averages 0.57 s/image (17 GB peak), whereas the lightweight PIQO variant averages 0.027 s/image (4.3 GB peak). It is worth noting that our method demonstrates advantages over agent-based JarvisIR (CVPR 2025) during inference; see [Reviewer YKa4, Limitation 1]. We will include these figures and discuss deployment trade-offs in the revision.
>
> ---
> **[W2&Q2] Reward Design:**
>
> Thank you for the question. We observed that using a single IQA model as the reward could lead to overfitting—while the metric score improved, human-perceived quality often deteriorated. The intuition behind this is that the model can optimize too closely to the reward metric, without capturing the broader perceptual nuances. To mitigate this, we adopted a composite reward combining multiple IQA models with complementary strengths, which helps balance different aspects of perceptual quality and aligns well with human observation. Each score is normalized to [0, 1] to ensure fair weighting across models. Although the weights are hand-tuned, this approach stabilizes training and prevents the model from overfitting to a single metric. Additionally, if training does not converge or exhibits early signs of overfitting (e.g., performance regression despite rising metric scores), we stop early or adjust the weights to restore a better balance. We will include more details on the tuning process and stopping criteria in the revision to clarify how we ensure robust performance.
>
> ---
> **[W3] Expansibility:**
>
> Thank you for your question. To assess the impact of agent library size on performance, we expanded the weather-related agent pool from three to four by adding a Raindrop degradation agent. In real rainy scenarios, this resulted in significant improvements across multiple IQA metrics, suggesting that additional agents can enhance restoration quality. However, the larger agent pool also introduces increased scheduling complexity and inference latency. Future work will explore adaptive agent selection and dynamic scheduling to optimize the trade-off between performance gains and computational cost.
> | Methods          | LIQE  | CLIP-IQA | Q-Align | MUSIQ | Latency |
> |------------------|-------|----------|---------|-------|---------|
> | **Basic**        | 4.02  | 0.56     | 3.29    | 64.11 | 567ms   |
> | **Basic + RainDrop** | 4.14  | 0.59     | 3.37    | 67.42 | 595ms   |
>
> ---
> **[W4] Extreme Degradation Scenarios:**
>
> Thank you for the comment. We agree that extreme degradation scenarios are important to evaluate. In fact, we have included corresponding analysis and visual results in the supplementary material (Sections B & C), where we test our model on severe fog, heavy snow, and dense rain conditions. The results show its effectiveness under extreme weather. We will clarify this in the main paper and explicitly reference the supplementary section.
>
> ---
> **[W5&Q3]  Further Comparisons:**
>
> We perform the experiments with JarvisIR  and DFPIR (CVPR 2025) on WReaL data and show the results below. It is worth noting that our method significantly outperforms JarvisIR in both inference speed and resource efficiency. Our method takes on average 570 ms and 17 GB peak memory per image, while JarvisIR requires 15,250 ms and 29.5 GB peak memory. The performance is evaluated by GPT-4o; see paper for details.
> Notably, our method demonstrates superior performance compared to these strong competitors.
>
> | Weather | Metric             | DFPIR | JarvisIR | Ours  |
> |---------|--------------------|-------|----------|-------|
> | **Snow** | Artifact Removal    | -  | 3.57     | 4.42  |
> |         | Weather Resilience  | -  | 3.61    | 4.35  |
> |         | Overall Visual Quality | - | 3.73     | 4.39  |
> | **Haze** | Artifact Removal    | 3.17  | 3.65     | 4.07  |
> |         | Weather Resilience  | 3.14  | 3.45     | 4.02  |
> |         | Overall Visual Quality | 3.24 | 3.58     | 3.95  |
> | **Rain** | Artifact Removal    | 3.34  | 3.71     | 4.25  |
> |         | Weather Resilience  | 3.15  | 3.69     | 4.07  |
> |         | Overall Visual Quality | 3.18 | 3.87     | 3.90  |
>
> ---
> **Q1:**
> Thank you for the question. The weights assigned to different IQA models in the PIQO reward were chosen based on their relevance to perceptual quality, with CLIP-IQA receiving the highest weight (w₂ = 1) due to its strong alignment with human visual quality judgments, and LIQE and Q-Align given smaller weights (w₁, w₃ = 0.2) to ensure diversity. Perturbing these weights during training showed that increasing CLIP-IQA’s weight sped up convergence but sometimes overfitted on perceptual details, while lowering weights for LIQE and Q-Align slowed convergence but maintained better balance. Large weight changes led to training instability.
>
> ---
> **Limitations:**
> Thank you for the feedback. We acknowledge that while our method introduces additional computational overhead, the analysis of limitations and failure cases could be more comprehensive. The increased computational cost primarily stems from the multi-agent system and the iterative optimization process. Nevertheless, our method demonstrates advantages over JarvisIR. In terms of failure cases, the system may struggle when the agent pool becomes too large, leading to increased inference latency and scheduling complexity. Additionally, when faced with highly complex or unseen degradation types, the system may require more agents, which can affect runtime efficiency. To mitigate these issues, we are exploring adaptive agent selection and dynamic scheduling to optimize performance while managing computational cost.

---

> > ### Comment · Reviewer_jNge · 2025-08-04
> >
> > I have read the authors’ response as well as the concerns raised by the other reviewers. The authors’ reply has addressed some of my concerns. I have determined my final score based on this.

---

> > > ### Author Response · Authors · 2025-08-06
> > >
> > > Thank you very much for your positive response and for deciding to adjust  the final rating. We're pleased to hear that our explanation has helped make the paper much stronger. If you have any further questions or comments, we would be happy to discuss them.

---

### Official Review · Reviewer_oVun · 2025-06-28

**Clarity:** 3
**Significance:** 3
**Originality:** 3
**Rating:** 5
**Confidence:** 3

**Summary:**

This manuscript proposes a dual-level reinforcement-learning framework for restoring real-world images degraded by rain, haze, and snow. It first builds HFLS-Weather, a million-image, depth-aware synthetic dataset that yields strong weather-specific “cold-start” networks. At the local level, each network is refined on unpaired real photos through Perturbation-Driven Image Quality Optimization, which perturbs model weights and selects improvements using a no-reference perceptual reward. At the global level, a meta-controller treats the specialists as agents and learns, from scene cues and past rewards, how to invoke or chain them for mixed degradations. Extensive experiments confirm that this cold-start plus two-stage RL scheme achieves state-of-the-art quality across multiple real-weather benchmarks.

**Questions:**

See weakness. I am also curious about the source of the gains brought by the proposed HFLS-Weather dataset. Do they stem mainly from the more accurate depth estimates obtained with Depth-Anything, or from the larger dataset scale?

**Ethical Concerns:**

["NO or VERY MINOR ethics concerns only"]

**Final Justification:**

I have carefully read the authors’ response as well as the comments from other reviewers. The authors have addressed the main concerns and proposed concrete improvements. I encourage them to include the resolution of the color shift issue and relevant analysis in the revised version. Considering the current contributions and remaining issues, I decide to keep my original score of 5.

**Limitations:**

Yes.

**Paper Formatting Concerns:**

No.

**Quality:**

3

**Strengths And Weaknesses:**

**Strengths**
1. The proposed two-level reinforcement-learning strategy (PIQO and meta-agent) is novel and effective for real-world weather restoration.
2. The manuscript is well organized and written in clear, straightforward language, making the technical ideas easy to grasp.

**Weaknesses**

1. Reward–metric coupling. The RL reward is built from the same no-reference IQA scores that later serve as evaluation metrics. This tight coupling risks over-fitting the policy to those specific measures and weakens claims of fair comparison. Distinct reward and test metrics would provide a more balanced assessment.
2. In Figure 4, the Basic + PIQO + Agent result removes weather effects but shows a visible colour shift, while Basic + PIQO keeps the colours closer to the input. Repeated passes through multiple agents seem to chase higher IQA scores at the expense of faithful colours. Have the authors considered how this issue might be mitigated?
3. The manuscript does not break down how different PIQO settings (e.g., KL threshold, reward mix) affect the outcome. More detailed ablation studies would help show which parts of the method matter most.

---

> ### Author Rebuttal · Authors · 2025-07-31
>
> We sincerely thank the reviewer for the valuable comments and constructive suggestions, which have played an important role in helping us further improve our work. Your review has been extremely helpful in refining our thinking and enhancing the quality of the paper. We also sincerely appreciate your recognition of the novelty and effectiveness of our work. Below, we provide detailed responses to your comments. All corresponding discussions and additional experimental results will be included in the final camera-ready version.
>
> ---
> **[W1]:** Thank the reviewer for this insightful comment. In addition to the IQA metrics used for training, we incorporate MUSIQ exclusively for evaluation and include GPT-4o-based perceptual scores and visual comparisons as independent assessments at test time, helping to mitigate potential reward–metric coupling and ensure fair evaluation. We will clarify this issue in revision.
>
> ---
> **[W2]:** Thank you for the observation. We acknowledge that repeated agent passes may cause color shifts while optimizing IQA scores. To address this, we plan to explore adding global color consistency terms (e.g., ΔE distance) to the reward and applying a lightweight color correction module post-restoration and also plan to investigate learning a reward function that balances perceptual quality with fidelity to input color distributions.
>
> ---
> **[W3]:** Thank you for the suggestion. PIQO settings like the KL threshold and reward weights have a critical impact on convergence. We set $τ = 0.01$ to balance stability and learning speed—lower values may slow or stall convergence, while higher ones can cause instability. If training fails to converge, we recommend reducing $τ$ (e.g., from 0.01 to 0.005 or 0.001) or relaxing the reward filtering criteria to retain more candidate samples. For reward weights, we prioritize CLIP-IQA (w₂ = 1) and assign smaller weights to LIQE and Q-Align (w₁, w₃ = 0.2) to ensure perceptual alignment while maintaining diversity. We will include tuning guidance in the revision.
>
> ---
> **[Q]:** Thank you for the question. The gains from HFLS-Weather come from both Depth-Anything’s accurate depth and the large, diverse dataset scale. Depth improves realism in weather simulation (e.g., fog depth cues), while scale enhances generalization. As shown in Table 4, both contribute complementary benefits, and we will clarify this in the revision.

---

> ### Comment · Reviewer_oVun · 2025-08-05
>
> I have carefully read the authors’ response as well as the comments from other reviewers. The authors have addressed the main concerns and proposed concrete improvements. I encourage them to include the resolution of the color shift issue and relevant analysis in the revised version. Considering the current contributions and remaining issues, I decide to keep my original score of 5.

---

> > ### Author Response · Authors · 2025-08-06
> >
> > Thank you for your positive feedback and for maintaining a strong endorsement of our work. We’re glad that our explanations and proposed improvements have addressed your concerns. We will incorporate the color-shift resolution and relevant analysis into the revised manuscript as you suggested. Please let us know if you have any further questions or comments—we’d be happy to continue the discussion.

---

### Official Review · Reviewer_KNba · 2025-06-29

**Clarity:** 3
**Significance:** 3
**Originality:** 3
**Rating:** 5
**Confidence:** 5

**Summary:**

This paper intent to restore the real-world degraded image using a universal model. They first produce a huge dataset containing 1 million images. Further, they apply a pretrain and finetune strategy, which first pretrain on synthetic data and finetune on real-word data based on reinforcement learning. And a multi-agent system is proposed to restore real-world degraded image.

**Questions:**

1) Could you make the comparison of model efficiency? In my understanding, your proposed Muti-Agent System requires run the single model many times.
2) It is not clear how to train the Muti-Agent System. And more inference details about this agent will be helpful to the reader.
3) In Muti-Agent System, does the restoration is processed step by step? If a degraded image contain rain, haze, and snow, does this agent needs to run three or more single model many times? To my knowledge, previous works could remove rain, haze, and snow just run the model once. What are the advantages of your agent?
4) Overall, I think this is a interesting work, I will reconsider my rate according to your rebuttal for the weaknesses and the questions.

**Ethical Concerns:**

["NO or VERY MINOR ethics concerns only"]

**Final Justification:**

The author has addressed my concerns, so I will raise my rate to 5.

**Limitations:**

yes

**Quality:**

3

**Strengths And Weaknesses:**

Strengths
1) The dataset they produced is beneficial to the image restoration fields if the author could make them public available.
2) The application of Group Relative Policy Optimization to image restoration task is interesting, which the real-world degraded images lacks ground truth to conduct supervision learning.
3) Introducing the agent to the image restoration is interesting.
4) The experiment results show that the proposed method outperforms baseline models.

Weaknesses
1) Perturbing the model parameter to produce multiple results have many previous researches, please add the related reference. And it is better to claim your improvement if you don’t follow their method completely.
2) I am not sure if you introduce a new reinforcement learning method for image restoration or you just connect ‘perturbing model parameter’ and ‘reinforcement learning’ in series.
3) The comparison with baseline models is not fair. The dataset you produce is huger than other dataset, as we all know, the model training on huger data will usually lead to better performance. Do you train baseline models in your dataset? If not, the main contribution may lay on your proposed dataset rather than the model.
4) You state in line 277: ‘To evaluate the effectiveness of our high-fidelity synthetic dataset……’, but in Table 4, the ablation data you used seems not the multi-scene dataset. If the training set only contain singe degraded image, it is impossible to restore mixed degraded image. Anyway, Table 4 can demonstrate the effectiveness to some extend.

---

> ### Author Rebuttal · Authors · 2025-07-31
>
> We sincerely thank the reviewer for the valuable comments and constructive suggestions, which have played an important role in helping us further improve our work. Your review has been extremely helpful in refining our thinking and enhancing the quality of the paper. We also sincerely appreciate your recognition of the novelty and effectiveness of our work. Below, we provide detailed responses to your comments, with related issues grouped together for clarity. All corresponding discussions and additional experimental results will be included in the final camera-ready version.
>
> ---
> **[W1] Related works:**
>
> Thank the reviewer for the suggestion. We will add citations to prior works that use perturbations to generate diverse outputs, including DFPIR (CVPR 2025) for degradation-aware feature perturbation [DFPIR, CVPR’25], and earlier latent-space or parameter-based methods such as Deep Mean‑Shift Priors and Autoencoding Priors for image restoration [Bigdeli et al., ICCV’17; Aittala & Durand, ICCV’17]. Unlike these, our approach applies RL-guided parameter perturbations with IQA-based reward filtering, and introduces a dual-level structure with a global meta-controller for adaptive model coordination, which distinguishes our method.
>
> ---
> **[W2&Q2] Details:**
>
> Thank you for the question. Our contribution is the novel adaptation of RL to image restoration via the proposed Perturbation-driven Image Quality Optimization (PIQO). Unlike standard perturbation or prior RL methods, PIQO injects Gaussian noise into model parameters to generate diverse outputs, filters them using no-reference IQA metrics, and applies a GRPO-inspired update based on group-wise advantages—all without paired supervision. This design enables RL-based training in a domain with deterministic outputs and sparse reward signals, making it, to our knowledge, the first framework to successfully adapt GRPO-style optimization to real-world image restoration. The novelty of our method is acknowledged by *Reviewer oVun-S1* and *Reviewer jNge-S2*, such as “The proposed two-level reinforcement-learning strategy (PIQO and meta-agent) is novel and effective for real-world weather restoration.”
>
> ---
> **[W3] Comparison:**
>
> First, we have reported the performance of the model trained on the proposed dataset without the RL training (PIQO) and agent framework in Table 5 (basic) in paper. The results clearly show the improvement of the proposed PIQO and agent framework. Second, we re-trained WGWS (CVPR,2023) and OneRestore (ECCV,2024) on our dataset, and here are the results. Will add the new results into the paper.
>
> | Methods       | Snow (LIQE) | Snow (CLIP-IQA) | Snow (Q-align) | Snow (MUSIQ) | Haze (LIQE) | Haze (CLIP-IQA) | Haze (Q-align) | Haze (MUSIQ) | Rain (LIQE) | Rain (CLIP-IQA) | Rain (Q-align) | Rain (MUSIQ) |
> |---------------|-------------|-----------------|----------------|--------------|-------------|-----------------|----------------|--------------|-------------|-----------------|----------------|--------------|
> | **WGWS**      | 3.71        | 0.54            | 3.40           | 65.67        | 3.31        | 0.40            | 2.57           | 56.76        | 3.81        | 0.47            | 2.70           | 56.62        |
> | **OneRestore**| 3.73        | 0.55            | 3.56           | 65.32        | 3.38        | 0.41            | 2.67           | 57.13        | 3.83        | 0.48            | 2.68           | 58.67        |
> | **Ours**      | 3.84        | 0.56            | 3.70           | 66.30        | 3.53        | 0.44            | 3.01           | 63.60        | 3.93        | 0.53            | 2.98           | 60.43        |
>
> ---
> **[W4] Effectiveness:**
>
> Thank you for the observation. While some datasets in Table 4 focus on single degradation types, we also include more diverse datasets such as RealSnow and SPA+, which cover complex and mixed weather conditions. Additionally, our HFLS-Weather dataset supports mixed-weather synthesis (e.g., snow+haze, rain+haze), and the “Our Snow+Haze” and “Our Rain+Haze” entries reflect this capability. These results demonstrate that our dataset not only provides high fidelity but also enables robust performance across diverse and dynamic degradation scenarios. We will revise the text in Line 277 to clarify this point.
>
> ---
> **[Q1] Efficiency:**
> Thank you for the question. While our Multi-Agent System involves running multiple specialized models, which increases latency (570ms), it offers better restoration quality for complex or mixed weather conditions compared to single-model methods like OneRestore (13ms). Though slower than single-model systems, it is more efficient than other multi-agent methods like JarvisIR (15250ms). We will include a detailed efficiency-performance comparison in the revision to highlight these trade-offs.
>
> | Methods     | Latency  |
> |-------------|----------|
> | **WGWS**    | 68ms     |
> | **Chen et al** | 33ms  |
> | **PromptIR** | 151ms  |
> | **DACLIP**   | 6543ms  |
> | **OneRestore** | 13ms  |
> | **JarvisIR** | 15250ms  |
> | **Ours**     | 570ms   |
>
> ---
> **[Q3]:**
> Yes, our system restores images step by step by adaptively selecting and sequencing specialized agents based on scene analysis and prior success rates. Unlike one-shot models, it uses IQA feedback to refine execution paths, enabling more accurate restoration under complex, mixed-weather conditions.

---

### Official Review · Reviewer_YKa4 · 2025-07-01

**Clarity:** 2
**Significance:** 3
**Originality:** 2
**Rating:** 4
**Confidence:** 4

**Summary:**

This paper proposes a dual-level reinforcement learning (RL) framework for real-world adverse weather image restoration. It introduces a synthetic dataset (HFLS-Weather) for cold start and learns to adapt to real-world conditions using a two-level control mechanism: a Perturbation-based IQA Optimization (PIQO) module and a global meta-controller for multi-agent scheduling. The method is evaluated on real-world datasets and achieves superior performance compared to several baselines.

**Questions:**

How does the proposed method compare quantitatively and qualitatively with JarvisIR or other recent multi-agent RL-based restoration approaches?

Could you report inference time and computational overhead compared to single-model baselines? This is crucial to judge practical utility.

Have you tested the method on real-world datasets that are not augmented with HFLS-Weather synthetic pretraining? How robust is the model without the cold start?

**Ethical Concerns:**

["NO or VERY MINOR ethics concerns only"]

**Final Justification:**

The authors’ response has addressed most of my concerns. I look forward to seeing a more thorough discussion of related work in the revised version. The contributions of the paper are acceptable, and I am raising my score accordingly.

**Limitations:**

The authors did not clearly address efficiency trade-offs of their method. A framework involving multiple agents and meta-controllers may be difficult to deploy in latency-sensitive applications such as autonomous driving. Moreover, the use of synthetic data for initialization could still introduce bias, and the generalization ability to unseen degradation types should be further evaluated. I suggest the authors add:

A detailed runtime and model size analysis.

A discussion of potential performance degradation in non-synthetic test domains.

Clarification on whether the agent decisions are interpretable or human-understandable.

**Quality:**

3

**Strengths And Weaknesses:**

Strengths:

The proposed dual-level RL design is well-motivated and provides a practical solution to the unpaired restoration challenge under real-world adverse weather.

The integration of no-reference IQA metrics (LIQE, CLIP-IQA, QAlign) for reward modeling enables the framework to work in a self-supervised fashion.

The synthetic dataset (HFLS-Weather) shows strong fidelity and diverse degradation simulation, which helps reduce domain gap.

Experiments are extensive, covering multiple weather conditions, and are supported with both objective metrics and perceptual evaluations.

Weaknesses:

Related work discussion is incomplete: The paper omits prior works that adopt a similar RL-based image restoration paradigm. In particular, JarvisIR (CVPR 2025) also employs a multi-agent RL strategy to improve image quality under adverse weather, guided by perception-aware rewards. While this submission differs in its use of IQA metrics, the high-level structure and adaptive policy scheduling are conceptually similar and should be discussed.

Algorithmic novelty is somewhat limited: The core method builds upon existing mechanisms like parameter perturbation and hierarchical controllers (inspired by GRPO and meta-RL) without introducing fundamentally new learning algorithms.

Efficiency concerns: The proposed framework adds significant inference overhead due to the use of multiple agents and iterative optimization, but runtime and memory profiles are not reported.

Writing quality: The abstract contains a duplicated sentence, and several grammatical errors appear throughout the paper. Some figures (e.g., Fig. 2) are visually dense and lack clear annotation, which hinders readability.

---

> ### Author Rebuttal · Authors · 2025-07-31
>
> We sincerely thank the reviewer for the valuable comments and constructive suggestions, which have played an important role in helping us further improve our work. Your review has been extremely helpful in refining our thinking and enhancing the quality of the paper. We also sincerely appreciate your recognition of the good motivation and effectiveness of our work. Below, we provide detailed responses to your comments, with related issues grouped together for clarity. All corresponding discussions and additional experimental results will be included in the final camera-ready version.
>
> ---
> **[W1] Related works:**
>
> Thank the review for pointing out the work of JarvisIR (CVPR 2025). We will include it in the revised paper. While both JarvisIR and our method adopt multi-agent RL strategies for weather-degraded image restoration, our approach differs in two key aspects: (i) we design a dual-level framework where a global meta-controller dynamically schedules restoration agents, complementing the local PIQO-based refinement, and (ii) we introduce a composite, no-reference reward function based on IQA metrics (CLIP-IQA, LIQE, Q-Align) with reward filtering to improve learning stability.
>
> ---
> **[W2] Novelty:**
>
> We propose PIQO, a novel approach that successfully extends GRPO-style reinforcement learning to image restoration by addressing the challenges of single-image input and non-deterministic rewards. The overall dual-level reinforcement learning framework for adverse weather image restoration is another contribution.
> The novelty of our method is acknowledged by *Reviewer KNba-S2*, *Reviewer oVun-S1*, and *Reviewer jNge-S2*, such as “The proposed two-level reinforcement-learning strategy (PIQO and meta-agent) is novel and effective for real-world weather restoration.”
>
> ---
> **[W3&Q2] Efficiency:**
>
> We report that our full framework (PIQO + Agent) takes on average 0.57 seconds per image on an RTX 4090 GPU with 17 GB peak memory usage. We also provide an ablation with a lightweight single-agent PIQO variant (0.027 seconds, 4.3 GB) that balances performance and efficiency. We will include these details and discuss practical deployment trade-offs.
> It is worth noting that our method significantly outperforms JarvisIR in both inference speed and resource efficiency. Our method takes on average 570 ms and 17 GB peak memory per image, while JarvisIR requires 15,250 ms and 29.5 GB peak memory.
>
> ---
> **[W4] Writing:**
>
> Thanks for the careful review. We have removed the duplicated sentence in the abstract, corrected grammatical issues throughout the paper, and revised Figure 2 with clearer annotations and a simplified layout for better readability in the revised paper.
>
> ---
> **Q1:**
> We perform the experiments with JarvisIR and DFPIR (CVPR 2025) on WReaL data and show the results below. The performance is evaluated by GPT-4o; see paper for details.
> Notably, our method demonstrates superior performance compared to these strong competitors.
>
> | Weather | Metric             | DFPIR | JarvisIR | Ours  |
> |---------|--------------------|-------|----------|-------|
> | **Snow** | Artifact Removal    | -     | 3.57     | 4.42  |
> |         | Weather Resilience  | -     | 3.61     | 4.35  |
> |         | Overall Visual Quality | -   | 3.73     | 4.39  |
> | **Haze** | Artifact Removal    | 3.17  | 3.65     | 4.07  |
> |         | Weather Resilience  | 3.14  | 3.45     | 4.02  |
> |         | Overall Visual Quality | 3.24 | 3.58     | 3.95  |
> | **Rain** | Artifact Removal    | 3.34  | 3.71     | 4.25  |
> |         | Weather Resilience  | 3.15  | 3.69     | 4.07  |
> |         | Overall Visual Quality | 3.18 | 3.87     | 3.90  |
>
> ---
> **Q3:**
> Thank you for the question. Yes, we have tested our method on real-world datasets without the HFLS-Weather synthetic pretraining (i.e., without the cold start). In these tests, we observed that the model struggles to converge when mitigating weather-induced degradations, showing inferior performance compared to the cold-start approach. Our ablation study (Table 4 in paper) highlights the superiority of HFLS-Weather as a cold start, showing substantial improvements in image quality for snow, haze, and rain. Additionally, pretraining with diverse weather types, such as combining snow and haze, enhances the model's generalization ability during fine-tuning with PIQO.
>
> ---
> **Limitation 1:**
> While our Multi-Agent System involves running multiple specialized models, which increases latency (570ms), it offers better restoration quality for complex or mixed weather conditions compared to single-model methods like OneRestore (13ms). Though slower than single-model systems, it is more efficient than other multi-agent methods like JarvisIR (15250ms). We will include a detailed efficiency-performance comparison in the revision to highlight these trade-offs. Please also see W3 for model size.
>
> | Methods     | Latency  |
> |-------------|----------|
> | **WGWS**    | 68ms     |
> | **Chen et al** | 33ms  |
> | **PromptIR** | 151ms  |
> | **DACLIP**   | 6543ms  |
> | **OneRestore** | 13ms  |
> | **JarvisIR** | 15250ms  |
> | **Ours**     | 570ms   |
>
> ---
> **Limitation 2:**
> To address the potential performance degradation when a model trained on synthetic datasets is tested in real, non-synthetic environments, we propose the Perturbation-driven Image Quality Optimization (PIQO) reinforcement learning method. As demonstrated in Table 5 and Figure 4, PIQO facilitates continuous optimization by fine-tuning the model in real-world scenarios. This adaptive process significantly enhances performance, enabling the model to better handle real-world degradations and improving its robustness in actual environments.
>
> ---
> **Limitation 3:**
> Each agent's decision is based on its specialization in a particular degradation type (e.g., snow, haze, rain), with the agents learning to optimize their performance in these specific scenarios. The meta-controller is responsible for selecting which agents to use based on their past performance in similar situations. While the decisions are driven by learned policies and experience, the selection process is determined by the needs of the scenario and the agents' historical success.

---

> > ### Comment · Reviewer_YKa4 · 2025-08-04
> > **Official Comment by Reviewer YKa4**
> >
> > The authors’ response has addressed most of my concerns. I look forward to seeing a more thorough discussion of related work in the revised version. The contributions of the paper are acceptable, and I am raising my score accordingly.

---

> ### Author Response · Authors · 2025-08-06
>
> Thank you very much for your positive response and for deciding to raise the final rating. We’re pleased that our explanations have addressed most of your concerns and, as you suggested, will include a more thorough discussion of related work in the revised version.If you have any further questions or comments, we would be happy to discuss them.

---

### Note · Authors · 2025-08-16

We express our sincere gratitude to the Area Chair (AC) and all four reviewers for their rigorous evaluations, insightful questions, and recognition of the novelty and practical relevance of this work.

Reviewers emphasized a single, unified strength: our proposed two-level reinforcement-learning strategy (PIQO + meta-agent) is viewed as novel, well-motivated, and practically effective for real-world adverse-weather restoration; this includes introducing Group Relative Policy Optimization (GRPO) in a no-ground-truth setting to drive self-supervised optimization (which reviewers regarded as methodologically meaningful), and adopting a multi-agent formulation to enable adaptive scheduling. They also acknowledged the role of HFLS-Weather: as a depth-aware, million-image dataset, it supports high-fidelity cold starts and improves cross-scene generalization.

In response to the reviewers’ questions and concerns, our revision provides coherent additions and clarifications. On related work, we add citations in two directions: studies that use perturbations to generate diverse outputs, and the latest real-world image restoration works. On fairness of comparison, we strengthen the baselines by retraining WGWS/OneRestore on our own data, and we use the WReaL test set to compare with recent methods such as JarvisIR/DFPIR. On efficiency and resources, we report training/inference metrics, including end-to-end latency and peak memory, to inform deployment trade-offs. To address reward–metric coupling and color shifts, we additionally adopt MUSIQ and GPT-4o for perceptual evaluation to decouple from the training reward, and we propose a ΔE-based color-consistency term and a lightweight post-correction. For reproducibility, we provide practical guidance on key PIQO hyperparameters. For scalability, we present ablations on expanding the agent library, showing the trade-off between performance gains and latency overhead.

Finally, we would like to express our sincere appreciation and enthusiasm for the reviewers’ positive endorsements and score increases. We are encouraged that this line of work is seen as promising for further exploration by the community. We believe this paper will stimulate broader discussion and progress in real-world image restoration, and we thank the AC for careful and judicious consideration.

---

### Decision · Program_Chairs · 2025-09-17

**Decision:**

Accept (poster)

**Comment:**

This paper proposes a dual-level reinforcement learning framework for real-world adverse weather image restoration, combining a high-fidelity synthetic dataset (HFLS-Weather), a perturbation-driven PIQO module, and a meta-controller for adaptive agent scheduling. The approach is well-motivated and shows strong performance across rain, haze, and snow, with clear benefits over prior single-model and multi-agent baselines. Strengths include the novel adaptation of GRPO-style RL to unpaired restoration, the scale and fidelity of the synthetic dataset, and extensive ablations clarifying contributions. Weaknesses raised by reviewers concerned fairness of comparisons, efficiency, reward–metric coupling, and color fidelity. The rebuttal addressed these points with retrained baselines, runtime/memory reports, added evaluation metrics, and solutions for color shift, which satisfied most reviewers and improved scores. While algorithmic novelty is incremental, the integration of components is coherent, impactful, and practically relevant. Overall, I recommend acceptance given the strong results, thorough evaluation, and community value of both the framework and dataset.